**Registered Report**

# Systematic assessment of long-read RNA-seq methods for transcript identification and quantification

The Long-read RNA-Seq Genome Annotation Assessment Project Consortium was formed to evaluate the effectiveness of long-read approaches for transcriptome analysis. Using different protocols and sequencing platforms, the consortium generated over 427 million long-read sequences from complementary DNA and direct RNA datasets, encompassing human, mouse and manatee species. Developers utilized these data to address challenges in transcript isoform detection, quantification and de novo transcript detection. The study revealed that libraries with longer, more accurate sequences produce more accurate transcripts than those with increased read depth, whereas greater read depth improved quantification accuracy. In well-annotated genomes, tools based on reference sequences demonstrated the best performance. Incorporating additional orthogonal data and replicate samples is advised when aiming to detect rare and novel transcripts or using reference-free approaches. This collaborative study offers a benchmark for current practices and provides direction for future method development in transcriptome analysis.

The rise of long-read RNA sequencing (lrRNA-seq) technologies demands thorough evaluation. The Long-read RNA-Seq Genome Annotation Assessment Project (LRGASP), an open community effort modeled after successful benchmarking projects[1–4], tackled this by testing tools and platforms across three key areas (Fig. 1a):

- Challenge 1: reconstructing full-length transcripts for well-annotated genomes.
- Challenge 2: quantifying transcript abundance.
- Challenge 3: de novo transcript reconstruction for genomes lacking high-quality references.

Long-read sequencing showed its potential for capturing full-length and novel transcripts, even in well-known genomes; however, moderate agreement among bioinformatics tools highlighted variations in analytical goals. Quantifying transcripts effectively remains challenging, with long-read tools lagging behind short-read tools due to throughput and error limitations. The project also validated many lowly expressed, single-sample transcripts, suggesting further exploration of long-read data for reference transcriptome creation.

## Results

### LRGASP data and study design

The LRGASP Consortium Organizers produced long-read and short-read RNA-seq data from aliquots of the same RNA samples using a variety of library protocols and sequencing platforms (Fig. 1a, Supplementary Table 1 and Supplementary Data 1). The Challenge 1 and 2 samples consisted of human and mouse ENCODE biosamples, including the human WTC11 induced pluripotent stem (iPS) cell line and a mouse embryonic stem (ES) cell line for Challenge 1 and a mix (H1-mix) of H1 human ES (H1-hES) cells and definitive endoderm derived from H1 (H1-DE) for

✉e-mail: vollmers@ucsc.edu; frankish@ebi.ac.uk; kinfai@med.umich.edu; gs9yr@virginia.edu; ali.mortazavi@uci.edu; ana.conesa@csic.es; anbrooks@ucsc.edu.

**Fig. 1 | Overview of the LRGASP. a**, Data produced for LRGASP. **b**, Distribution of read lengths, identify Q score and sequencing depth (per biological replicate) for the WTC11 sample. **c**, The collaborative design of the LRGASP organizers and participants. **d**, Number of isoforms reported by each tool on different data types for the human WTC11 sample for Challenge 1. Number of submissions per tool, in order, n = 6, 6, 4, 1, 6, 1, 6, 3, 1 and 12. **e**, Median TPM value reported by each tool on different data types for the human WTC11 sample for Challenge 2. Number of submissions per tool, in order, n = 11, 3, 4, 6, 1, 6 and 1. **f**, Number of isoforms reported by each tool on different data types for the mouse ES data for Challenge 3. Number of submissions per tool, in order, n = 6, 5, 2 and 4. **g**, Pairwise relative overlap of unique junction chains (UJCs) reported by each submission. The UJCs reported by a submission are used as a reference set for each row. The fraction of overlap of UJCs from the column submission is shown as a heatmap. For example, a submission that has a small subset of many other UJCs from other submissions will have a high fraction shown in the rows but a low fraction by column for that submission. Data are only shown for WTC11 submissions. **h**, Spearman correlation of TPM values between submissions to Challenge 2. **i**, Pairwise relative overlap of UJCs reported by each submission. The UJCs reported by a submission are used as a reference set for each row. The fraction of overlap of UJCs from the column submission is shown as a heatmap. Ba, Bambu; Bl, RNA-Bloom; FM, FLAMES; FR, FLAIR; IB, Iso_IB; IQ, IsoQuant; IT, IsoTools; Ly, LyRic; Ma, Mandalorion; rS, rnaSPAdes; Sp, Spectra; ST, StringTie2; TL, TALON-LAPA. The figure was partially created with BioRender.com.

**BOX 1**

# Metrics used for evaluation of Challenges 1 and 2

| Challenge | Metric | Description |
|---|---|---|
| 1 | FSM | Transcripts matching a reference transcript at all splice junctions. |
| 1 | ISM | Transcripts matching consecutive, but not all, splice junctions of the reference transcripts. |
| 1 | NIC | Transcripts containing new combinations of (1) already annotated splice junctions, (2) novel splice junctions formed from already annotated donors and acceptors or (3) unannotated intron retention. |
| 1 | NNC | Transcripts using novel donors and/or acceptors. |
| 1 | Reference match | FSM transcript with 5′ and 3′ ends within 50 nt of the TSS/TTS annotation. |
| 1 | 3′ poly(A) supported | Transcript with poly(A) signal sequence support or short-read 3′ end sequencing (for example QuantSeq) support at the 3′ end. |
| 1 | 5′ CAGE supported | Transcript with CAGE support at the 5′ end. |
| 1 | 3′ reference supported | Transcript with 3′ end within 50 nt from a reference transcript TTS. |
| 1 | 5′ reference supported | Transcript with 5′ end within 50 nt from a reference transcript TSS. |
| 1 | SRTM | FSM/ISM transcript with 5′ end within 50 nt of the TSS or has CAGE support AND 3′ end within 50 nt of the TTS or has poly(A) signal sequence support or short-read 3′ end sequencing support. |
| 1 | SNTM | NIC/NNC transcript with 5′ end within 50 nt of the TSS or CAGE support AND 3′ end within 50 nt of the TTS or has poly(A) signal sequence support or short-read 3′ end sequencing support AND Illumina read support at novel junctions. |
| 1 | %LRC | Fraction of the transcript model sequence length mapped by one or more long reads. |
| 1 | Read multiplicity | Number of assigned transcripts per read. |
| 1 | Redundancy | No. LR transcript models/reference model. |
| 1 | Longest junction chain ISM NIC/NNC | No. junctions in ISM/no. junctions reference no. reference junctions/no. junctions in NIC/NNC. |
| 1 | Intron retention (IR) level | Number of IR within the NIC category. |
| 1 | Ilumina splice junction support | Percentage of splice junction in transcript model with Illumina support. |
| 1 | Full Illumina splice junction support | Percentage of transcripts in category with all splice junction supported. |
| 1 | Percentage of novel junctions | No. of new junctions/total no. junctions. |
| 1 | Percentage of non-canonical junctions | No. of non-canonical junctions/total no. junctions. |
| 1 | Percentage of non-canonical transcripts | Percentage of transcripts with at least one non-canonical junction. |
| 1 | Intra-priming | Evidence of intra-priming (described elsewhere[8]). |
| 1 | Reverse transcriptase (RT) switching | Evidence of RT switching (described elsewhere[8]). |
| 2 | IM and ACVC | IM and ACVC characterize the CV of abundance estimates among multiple replicates. |
| 2 | CM and ACC | CM and ACC characterize the similarity of abundance profiles between pairs of replicates. |
| 2 | RE | RE characterizes the resolution of abundance estimation. |
| 2 | SCC | SCC evaluates the monotonic relationship between the estimation and the ground truth. |
| 2 | MRD | MRD is the median of the relative difference of abundance estimates among all transcripts. |
| 2 | NRMSE | NRMSE measures the normalized root mean square error between the estimation and the ground truth, which characterizes the variability of the quantification accuracy. |
| 2 | PET | PET characterizes the percentage of truly expressed transcripts in SIRV-Set 4 data. |

Challenge 2. All samples were grown as biological triplicates with the RNA extracted at one site, spiked with 5′-capped spike-in RNA variants[5] (Lexogen SIRV-Set 4) and distributed to all production groups. A single pooled sample of manatee whole-blood transcriptome was generated for Challenge 3. We performed different cDNA preparation methods for each sample, including an early-access Oxford Nanopore Technologies (ONT) cDNA kit (PCS110), ENCODE Pacific Biosciences (PacBio)

cDNA and R2C2 (ref. 6) for increased sequence accuracy with the ONT platform and CapTrap[5] to enrich for 5′-capped RNAs[7] (Supplementary Methods). We also performed direct RNA sequencing (dRNA) with ONT.

The quality of the LRGASP datasets was extensively assessed (Supplementary Tables 2–6). cDNA-PacBio and R2C2-ONT datasets contained the longest read-length distributions, whereas sequence quality was higher for CapTrap-PacBio, cDNA-PacBio and R2C2-ONT

than other experimental approaches. We obtained approximately ten times more reads from CapTrap-ONT and cDNA-ONT than with other methods (Fig. 1b).

LRGASP invited tool developers to submit predictions for all three challenges employing the consortium datasets for which ground-truth data were or were not available. Moreover, the consortium provided evaluation metrics[8] and scripts to participants. (Fig. 1c and Box 1). To avoid conflict of interest, the evaluations and validations were performed by a subgroup of the LRGASP organizers who did not submit predictions. The entire LRGASP study design and evaluation benchmarks were published as a Registered Report[9]. This open design aimed for a fair and transparent benchmarking effort.

A total of 14 tools and laboratories submitted predictions (Supplementary Table 7). While submitters could choose the type of experimental procedure (a combination of library preparation and sequencing platform) that they wished to participate in, predictions were required for all biological samples in the chosen experimental procedure to assess pipeline consistency. We received 141, 143 and 25 submissions for Challenges 1, 2 and 3, respectively.

We observed a large variability in the number and quantification of transcript models predicted by each submission, with differences of up to tenfold in each challenge (Fig. 1d–f). Moreover, there was little overlap in transcripts identified by any two pipelines in Challenges 1 and 3 and low pairwise correlations were detected for Challenge 2 quantification results (Fig. 1g–i and Supplementary Data 2). These results highlight the importance of the comprehensive benchmarking presented here.

## Evaluation of transcript detection with a high-quality genome

In Challenge 1, we assessed transcript model predictions using various datasets to gauge different aspects of performance. Experimental methods and tools were evaluated for their ability to detect transcripts and genes using SQANTI3 (ref. 8) categories and orthogonal datasets detailed in Supplementary Table 8, Fig. 2a and Extended Data Fig. 1. Results were consistent across WTC11, H1-mix and mouse ES samples (Fig. 2a, Supplementary Figs. 1 and 2 and Supplementary Data 3–5). We observed considerable variation in the detection of known genes (399–23,647) and transcripts (524–329,131) (Extended Data Fig. 1a), by the different methods, with an average of 3–4 transcripts per gene reported by most pipelines, except for Spectra[10] and Iso_IB[11] that reported a huge number of transcripts (~170,000 and 330,000K, respectively); however, the relationship between read metrics and detected transcript numbers was unclear due to pipeline variations (Supplementary Figs. 3–7). The analysis tool mostly dictated the number of detected features (Extended Data Fig. 1b–d).

Pipelines also greatly varied in detecting GENCODE-annotated transcripts (full splice match; FSM), transcripts missing 3′ or 5′ end exons (incomplete splice match; ISM), containing novel junctions of GENCODE-annotated donor and acceptor sites (novel in catalog; NIC) or containing novel donor or acceptor sites with respect to GENCODE (novel not in catalog; NNC). Bambu[12], FLAIR[13], FLAMES[14] and IsoQuant[15] consistently detected a high percentage of FSM and a low proportion of ISM transcripts. In contrast, TALON[16,17], IsoTools[18] and LyRic detected a relatively high number of ISMs (Extended Data Fig. 1b). The LyRic submission group noted that they did not use existing annotations to guide analysis, which can explain their results. As for novel transcripts, Bambu

reported the lowest values for NNC and NIC, followed by IsoQuant and TALON. FLAIR and Mandalorion[19] pipelines typically returned around 20% NIC and low NNC percentages. LyRic and FLAMES were among the pipelines with the highest percentages of novel transcript detections. Iso_IB and Spectra generally returned many isoforms and only a small fraction were FSMs (Extended Data Fig. 1c). Results stratified by library preparation and sequencing platform followed similar patterns (Supplementary Figs. 8–15).

We compared support for transcript models against reference annotations and short-read sequencing data, including cDNA sequencing, CAGE and QuantSeq. Our analysis revealed that many pipelines achieved a high percentage of known transcripts with full support at transcription start sites (TSSs), transcription termination sites (TTSs) and junctions (referred to as supported reference transcript models (SRTMs); Methods) but showed lower full support for novel transcript models (SNTMs) (Fig. 2a, Extended Data Fig. 1d and Supplementary Figs. 1 and 2). Generally, tools analyzing cDNA-PacBio and cDNA-ONT data demonstrated high values of full support for both novel and known transcripts; however, many TALON pipelines exhibited only moderate full support for known transcripts, possibly due to a high number of ISMs. Nonetheless, TALON consistently provided full support for novel transcripts in most cases. In contrast, LyRic, IsoQuant, FLAMES and Bambu, which exhibited high full support values for novel transcripts using cDNA-PacBio data, yielded novel transcript models with lower support when processing ONT libraries. Additionally, we observed that, in general, pipelines were more successful in reporting experimentally supported 3′ ends than 5′ ends. Transcript models generally aligned with reference TSSs and TTSs, although variations among pipelines were observed. Bambu and IsoQuant reported a high percentage of transcripts matching reference TSSs and TTSs but exhibited comparatively lower support from CAGE and QuantSeq data. Conversely, certain submissions from LyRic and FLAMES produced transcript models with experimentally validated transcript ends, with Mandalorion achieving the most consistent high CAGE support rates. This result suggested that lrRNA-seq pipelines are highly guided by reference annotations to complete transcript sequences. We tested this by measuring long-read coverage (LRC) of the transcript predictions from our read alignments. FLAMES, Iso_IB, IsoTools, LyRic and Mandalorion showed nearly complete LRC for their transcript models (>98% coverage for all transcripts). In contrast, FLAIR, Spectra, TALON, IsoQuant, Bambu and StringTie2 had lower coverage rates (~90, 90, 85, 75, 60 and 45%, respectively) (Extended Data Fig. 2), suggesting that they may use different alignment strategies or additional information (for example, reference annotation or short reads) to finalize transcript models. Finally, we looked at the percentage of junctions with Illumina reads support and canonical splice sites. We found these values were generally very high for all pipelines except Spectra, Iso_IB and FLAMES using cDNA-ONT and CapTrap-ONT data, with LyRic on PacBio showing the highest percentage of splice junctions supported by Illumina reads (Fig. 2a). Gene biotype detection was uniform across methods (Supplementary Figs. 16 and 17).

We assessed the consistency of detecting known and novel unique intron chains (UICs) by various pipelines across multiple sequencing setups. When considering all 47 WTC11 submissions, detection by only one pipeline was the most frequent transcript class (Fig. 2b and Supplementary Fig. 18). Moreover, frequency in transcript detection

---

**Fig. 2 | Evaluation of transcript identification with a reference annotation for Challenge 1. a**, Percentage of transcript models fully supported at 5′ ends either by reference annotation or same-sample CAGE data (left), 3′ end either by reference annotation or same-sample QuantSeq data (middle) and splice junctions (SJ) by short-read coverage or a canonical site (right). **b**, Agreement in transcript detection as a function of the number of detecting pipelines, **c**, Performance of tools based on spliced-short (top) and unspliced long SIRVs (bottom). **d**, Performance of tools based on simulated data. **e**, Performance

of tools on known and novel transcripts of 50 genes manually annotated by GENCODE. **f**, Summary of performance metrics of tools for the cDNA-PacBio and cDNA-ONT benchmarking datasets. The color scale represents the performance value ranging from worse (dark blue) to better (light yellow). The graphic symbol indicates the ranking position of the tool for the metric represented in each row. LO, long (reads) only; LS, long and short (reads); Sen_kn, sensitivity for known transcripts; Pre_kn, precision for known transcripts; Sen_no, sensitivity for novel transcripts; Pre_no, precision for novel transcripts; 1/Red, inverse of redundancy.

depended on the SQANTI3 structural category. Novel transcripts were less consistently detected, whereas FSMs were nearly the only transcript type found by more than 40 pipelines. Overall, the overlap in

detection between any two pipelines was higher for genes and junctions than for UICs, even when we only considered dominant UICs accounting for over 50% of the gene expression (Supplementary Figs. 19–21),

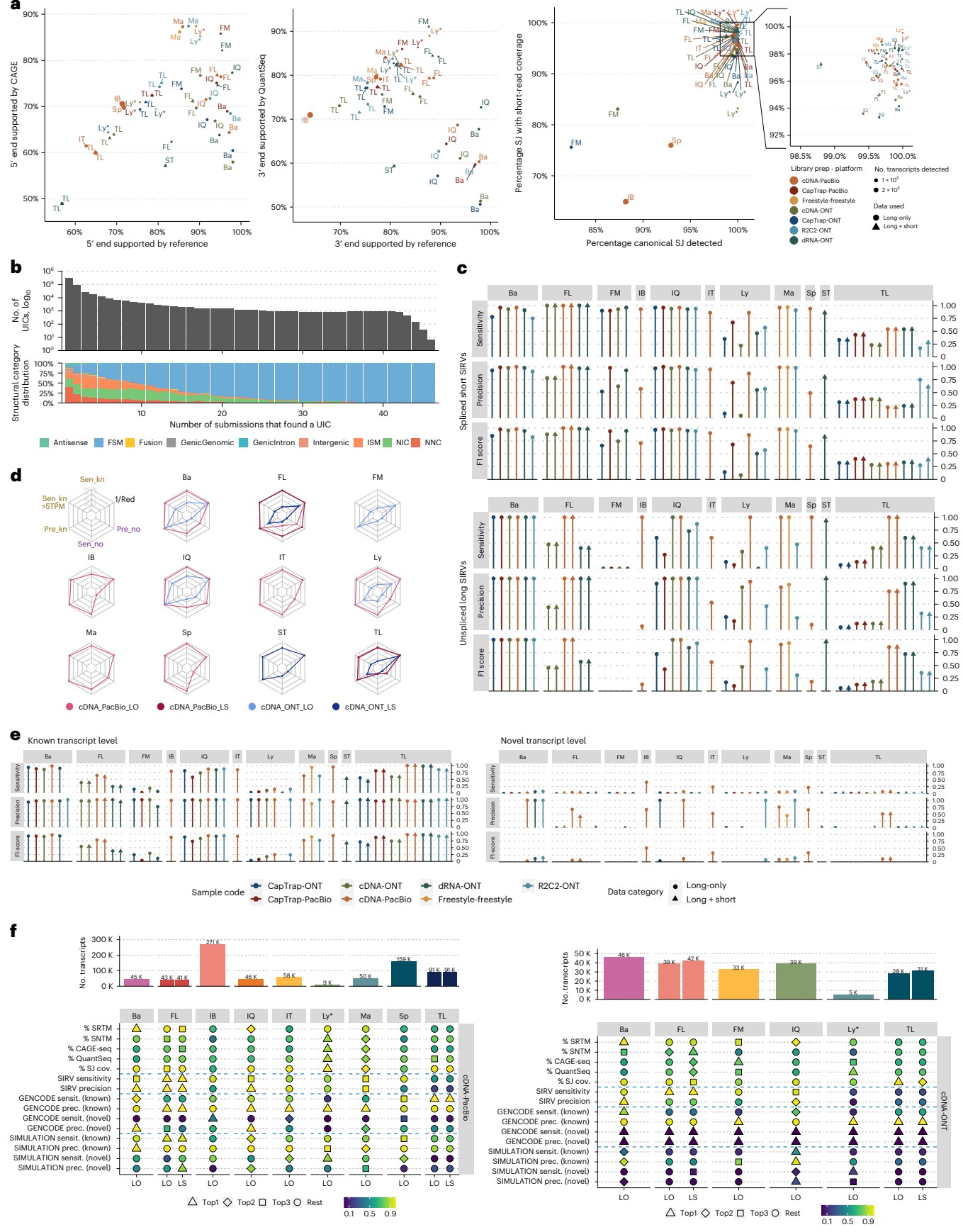

highlighting the disparity in the identification of transcript models across methodologies. We then re-evaluated the agreement in UIC detection by looking at the overlap of each analysis method with the rest and discarding tools with a large number of detections (Spectra and Iso_IB). Overall, many UICs were detected by all analysis pipelines when considering each library preparation/sequencing platform combination separately, although discrepancies persisted among SQANTI3 categories and experimental methods (Supplementary Figs. 22–27).

Without a ground truth in cell-line data, we gauged method accuracy using spike-ins, simulated data and GENCODE manual curation. Using the spliced SIRV-Set 4 dataset, most tools showed high sensitivity, except for TALON and LyRic, which did not use the SIRV reference annotation, whereas LyRic and Bambu's sensitivity varied with the library preparation method (Fig. 2c and Supplementary Data 3–5). LyRic only had a sensitivity above 0.8 for the cDNA-PacBio sample, and Bambu showed lower sensitivity with R2C2-ONT and CapTrap-ONT. Precision was generally high for Bambu, IsoQuant, IsoTools and Mandalorion methods and low for TALON, Iso_IB and Spectra. FLAMES, FLAIR and LyRic showed variable results. F1 scores generally matched precision, with IsoQuant, Mandalorion, FLAIR, Bambu and IsoTools performing best.

The SIRV dataset's long, non-spliced transcripts were separately assessed, yielding very different results (Fig. 2c and Supplementary Data 3–5). FLAMES reported no such transcripts, whereas Bambu excelled, likely aided by reference data. Sensitivity was generally lower, except with cDNA-PacBio, where most tools, including TALON and LyRic, achieved 100% sensitivity. CapTrap data typically resulted in lower sensitivity, suggesting limitations in capturing long transcripts. Precision varied with the tool and protocol used. As these SIRVs do not contain splice sites, low precision values indicate false variability at TTSs and TSSs. As results obtained with the long non-spliced dataset may be the combination of the ability of the analysis tool to process non-spliced data and accurately define TSSs and TTSs and the capacity of the experimental protocol to capture long molecules, we looked at the aligned coverage for long SIRVs. We found that cDNA-PacBio provided the most uniform LRC for long SIRVs despite a drop at the 5' ends of the longest transcripts (Extended Data Fig. 3).

SIRV annotation was available to participants. This benchmark, while useful for testing experimental protocols, might be biased and cannot evaluate new transcript predictions—a key advantage of long-read sequencing. We used simulated datasets, including undisclosed novel transcripts, for a broader assessment (Supplementary Data 6 and 7). Mandalorion performed well with PacBio data for both known and novel transcripts, as did IsoTools and LyRic, though their precision for novel transcripts was lower (Fig. 2d and Supplementary Fig. 28). Bambu and FLAIR were sensitive and precise for known transcripts but less so for novel ones, especially FLAIR without short-read data was unable to discover new transcripts accurately. Spectra and Iso_IB had sensitive but imprecise detection of novel transcripts. For all tools, sensitivity increased on highly expressed transcripts and

redundancy values were close to 1, except for Iso IB and Spectra, which returned a higher number of redundant predictions. Nanopore simulations generally showed low performance across tools, possibly due to lower NanoSim[20] read coverage in the simulated transcript models (Supplementary Fig. 29). Exceptions were Bambu and IsoQuant, which had good precision for ONT-known simulated transcripts and StringTie at metrics other than those related to novel transcript discovery. In summary, simulated data indicated lower sensitivity and precision for novel compared to known transcripts (Fig. 2d).

While useful for large-scale and novel transcript analysis, simulated data are limited by the simulation algorithms' properties that may not replicate complexities such as library preparation and biological noise. To address this, 50 undisclosed genes were rigorously annotated by GENCODE experts using LRGASP sequencing data for evaluation. Manually annotated loci were chosen for having mapped reads in all six library preparation/sequencing platform combinations and average to moderately high expression levels (Extended Data Fig. 4). GENCODE annotators evaluated the long-read data for each experimental procedure independently and called transcript models in each case (Supplementary Methods). Globally, 271 models, mostly novel, were accepted as true transcripts in the WTC11 sample, with NNC as the primary category. FSMs, though fewer, were more consistently detected across multiple conditions. Most novel transcripts appeared in just one dataset (Extended Data Fig. 4) and a trend was also observed in the mouse ES sample (Extended Data Fig. 5).

Assessment of pipelines on selected loci revealed performance variations driven by the analysis method. While all showed high gene-level precision, sensitivity was generally lower than in previous datasets. FLAMES, LyRic, FLAIR (on dRNA data) and TALON (on CapTrap and cDNA-ONT datasets) exhibited lower sensitivity. Bambu, IsoTools, IsoQuant and Spectra showed the highest sensitivity at the gene detection level (Supplementary Fig. 30a), followed by TALON and Mandalorion, but were more dependent on the data type. A similar pattern of sensitivity and precision was observed when considering transcripts already present in the reference annotation (Fig. 2e); however, for novel transcript detection, sensitivity was surprisingly low in all cases and precision greatly varied, ranging from 1 to 0 to non-computable even within the same tool, due to a low number of novel discoveries (<4) by most pipelines (Fig. 2e and Supplementary Fig. 30b). Results were similar for the mouse ES annotated dataset (Supplementary Figs. 31 and 32).

In summary, differences in library preparation, sequencing platforms and analysis tools significantly affected the transcriptome definition (Fig. 2f and Supplementary Figs. 33–36). Notably, the number of transcripts detected was not associated with the number of reads (Supplementary Fig. 3). Some tools (Bambu, FLAMES, FLAIR and IsoQuant) heavily relied on annotation for transcript modeling, while other methods (Iso_IB, IsoTools, Mandalorion, TALON and LyRic) allowed more novelty based on the actual data. For all methods, accurate prediction of novel transcripts was challenging.

**Fig. 3 | Evaluation of transcript isoform quantification for Challenge 2. a**, Cartoon diagrams to explain evaluation metrics without or with a ground truth. **b**–**e**, Overall evaluation results of eight quantification tools and seven protocols-platforms on real data with multiple replicates (**b**), cell mixing experiment (**c**), SIRV-Set 4 data (**d**) and simulation data (**e**). Box plots of evaluation metrics across various datasets, depicting the minimum, lower quartile, median, upper quartile and maximum values. Bar plots represent the mean values of evaluation metrics across diverse datasets, with error bars indicating the s.d. **b**, Number of submissions per tool or protocol-platform, in order, n = 36, 12, 16, 24, 4, 24, 6, and 4 per tool or n = 22, 24, 26, 18, 18, 14 and 4 per protocol-platform. **c**, Number of submissions per tool or per protocol-platform, in order, n = 6, 3, 4, 6, 1, 6, 1 and 1 per tool or n = 5, 5, 6, 4, 4, 3 and 1 per protocol-platform. **d**, Number of submissions per tool or per protocol-platform, in order, n = 36, 12, 16, 24, 4, 24, 6 and 4 per tool or n = 22, 24, 26, 18, 18, 14 and 4 per protocol-platform. **e**, Number of submissions per tool or per protocol-platform, in order, n = 8, 4, 2, 4, 2, 4, 1 and

2 per tool or n = 12, 6, 7, 0, 0, 0 and 2 per protocol-platform. **f**, Quantification tool scores under common cDNA-ONT and cDNA-PacBio platforms across various evaluation metrics, with the top three performers highlighted for each metric. **g**, Based on the average values of each metric across all quantification tools, scores for protocols-platforms are displayed, along with the top three performers for each metric. Blank spaces denote instances where the tool or protocols-platforms did not have participants submitting the corresponding quantitative results. **h**, Evaluation of quantification tools with respect to multiple transcript features, including the number of isoforms, number of exons, isoform length and a customized statistic K-value representing the complexity of exon-isoform structures. Here, the normalized MRD metric is used to evaluate the performance of quantification tools on human cDNA-PacBio simulation data. Additionally, RSEM evaluation results with respect to transcript features based on human short-read simulation data are shown as a control.

## Evaluation of transcript quantification

We assessed transcript quantification performance using 84 RNA sequencing datasets (including SIRV-Set 4) from four human cell lines (H1-hES cells, H1-DE, H1-mix and WTC11) and six simulation datasets for Nanopore (NanoSim), PacBio (IsoSeqSim[21]) and Illumina (RSEM[22]) reads (Fig. 1a). Seven quantification tools (IsoQuant,

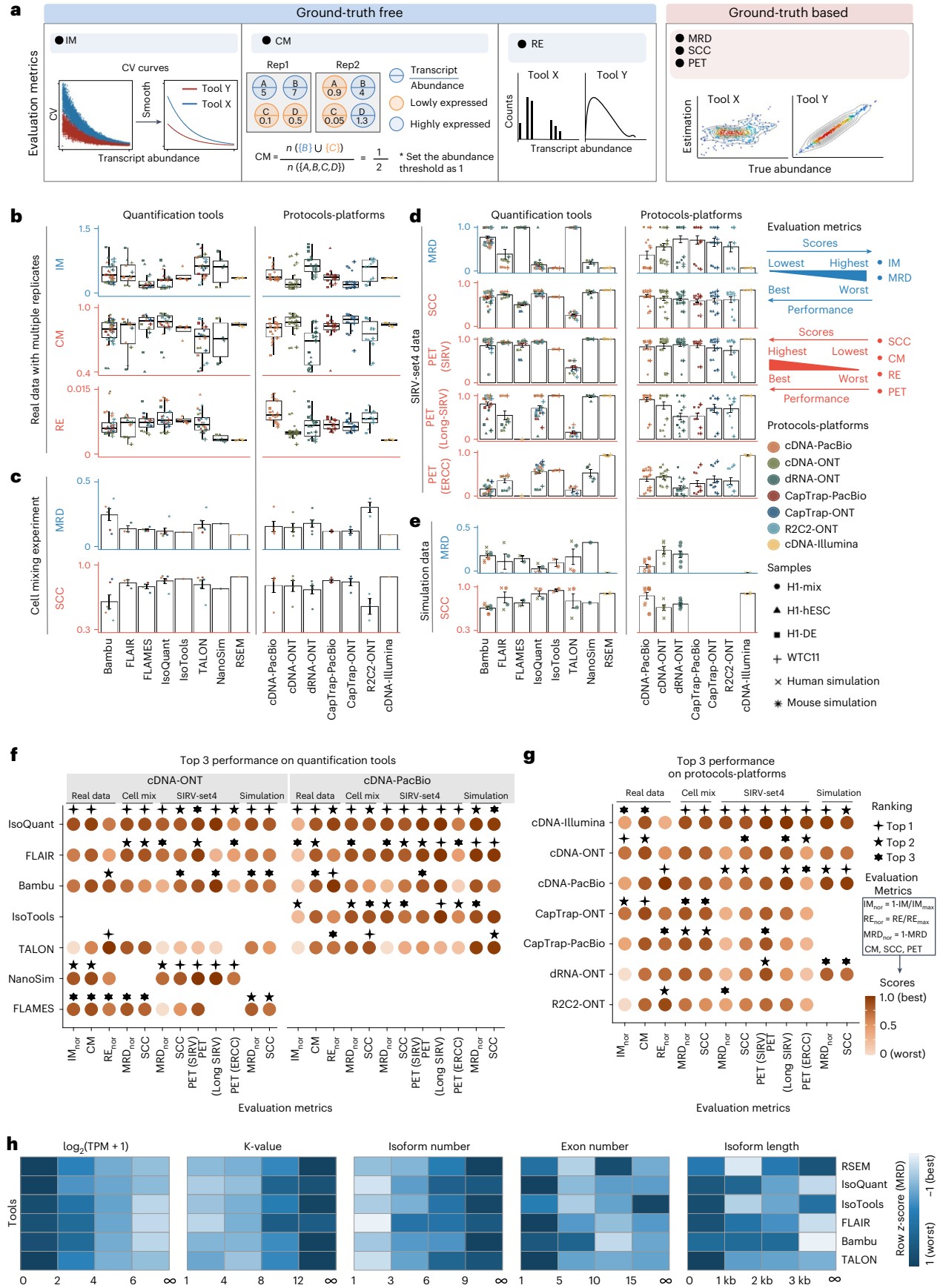

Bambu, TALON, FLAIR, FLAMES, NanoSim and IsoTools) were tested on six combinations of protocols-platforms (cDNA-PacBio, cDNA-ONT, dRNA-ONT, CapTrap-PacBio, CapTrap-ONT and R2C2-ONT), yielding 143 submitted datasets. As a control, we quantified short-read datasets (cDNA-Illumina) using the RSEM tool with the GENCODE reference annotation. We employed nine metrics for performance assessment in various data scenarios (Fig. 3a and Box 1). A benchmarking web application[23] allowed users to upload their results, generating interactive evaluation reports in HTML and PDF formats.

First, we assessed eight quantification tools across diverse protocols-platforms in four data scenarios (Fig. 3b–f and Extended Data Figs. 6 and 7). For real data with multiple replicates, four metrics were designed to evaluate the reproducibility and consistency of transcript abundance estimates among multiple replicates: irreproducibility measure (IM), area under the coefficient of variation curve (ACVC), consistency measure (CM) and area under the consistency curve (ACC) (Fig. 3b, Supplementary Figs. 37 and 38 and Supplementary Data 4). FLAMES, IsoQuant and IsoTools performed comparably to RSEM, showing low IM and ACVC and high CM and ACC (Fig. 3b and Supplementary Figs. 37b and 38c). FLAIR and Bambu slightly underperformed RSEM but surpassed other quantification tools. Specifically, IsoQuant on cDNA-ONT and CapTrap-ONT, and FLAMES on CapTrap-ONT ranked in the top three across all datasets (<0.15 and <0.53 for IM and ACVC, >0.89 and >9.53 for CM and ACC). Notably, all tools showed poor performance on dRNA-ONT (mean IM = 0.66, ACVC = 2.62, CM = 0.64 and ACC = 8.31), likely due to its low throughput (<1 million reads per replicate). In addition, the resolution entropy (RE) metric characterized the resolution of transcript abundance estimates among multiple replicates in real data (Fig. 3b, Supplementary Fig. 39 and Supplementary Data 8). The top six tools (IsoQuant, IsoTools, FLAMES, FLAIR, TALON and Bambu) achieved comparable resolution, at least 2.7-fold higher than NanoSim and RSEM. This disparity may be due to NanoSim and RSEM utilizing the GENCODE reference annotation, including numerous transcripts not expressed in specific samples, leading to many low-expression transcripts in the quantification results (79.02% and 58.04% of transcripts with transcripts per million (TPM) ≤ 1 in H1-hES cell samples).

Due to the challenges of transcript-level quantification and the lack of a gold standard in real data, we designed an evaluation strategy using a cell mixing experiment (Supplementary Fig. 40a). In this experiment, an undisclosed ratio of H1-hES cells and H1-DE samples was mixed before sequencing and participants estimated transcript abundance in the mixed sample initially. Subsequently, data from individual H1-hES cell and H1-DE samples were released and participants submitted separate quantifications for these datasets. The quantification of mixed samples should be equivalent to the expected ratios from the quantification of individual cell lines. Three metrics evaluated quantification accuracy by comparing expected and observed abundance: Spearman correlation coefficient (SCC), median relative difference (MRD) and normalized root mean square error (NRMSE) (Fig. 3c, Supplementary Fig. 40b,c and Supplementary Data 9). Most tools showed good correlation (0.74–0.87 for mean SCC) between expected and observed abundance, except Bambu (0.53), with RSEM showing superior performance in cell mixing experiments with the highest SCC (0.87), lowest MRD (0.13) and NRMSE (0.38) values (Fig. 3c and Supplementary Fig. 40b,c). Among long-read-based tools, IsoQuant on cDNA-ONT performed best in MRD (0.14) and SCC (0.85), whereas FLAIR on cDNA-ONT recorded the lowest NRMSE (0.43).

SIRV-Set 4 and the simulation data assessed the proximity of estimations to ground-truth values using four metrics: percentage of expressed transcripts (PET), SCC, MRD and NRMSE (Fig. 3d, Supplementary Figs. 41 and 42 and Supplementary Data 10 and 11). For SIRV-Set 4, tools exhibited substantial variation in quantifying SIRV transcripts with TPM > 0, ranging from 28 to 136. RSEM outperformed

other long-read-based tools with higher average SCC (0.84 versus 0.29–0.78), lower MRD (0.12 versus 0.13–1.00) and NRMSE (0.45 versus 0.89–2.19). NanoSim (SCC = 0.78, MRD = 0.23 and NRMSE = 0.89) and IsoQuant (0.76, 0.19 and 0.89) led long-read-based tools, followed by IsoTools (0.69, 0.13 and 1.02), FLAIR (0.73, 0.42 and 1.13) and Bambu (0.68, 0.79 and 1.55). Except for TALON and FLAMES, all tools excelled in quantifying regular and long SIRV transcripts with TPM > 0 (PET > 80%). Conversely, most struggled with quantifying ERCC transcripts with TPM > 0 (PET < 50%), likely due to the low expression levels of many ERCC transcripts[24,25].

For simulation data, tools performed markedly better on PacBio data than ONT data (Fig. 3e). Notably, FLAIR, IsoQuant, IsoTools and TALON on cDNA-PacBio exhibited the highest correlation (SCC > 0.97) between estimation and ground truth, slightly surpassing RSEM (SCC = 0.90) and outperforming other long-read pipelines (SCC < 0.83). Moreover, transcript annotation accuracy notably influenced quantification accuracy. With inaccurate annotation, RSEM yielded mean NRMSE values of 2.74- and 3.27-times higher than long-read-based tools and RSEM with accurate annotation, respectively (Supplementary Fig. 43). This emphasizes the critical importance of accurate sample-specific annotation for transcript quantification.

Next, we evaluated seven combinations of protocols-platforms across diverse quantification tools (Fig. 3b–e,g and Extended Data Fig. 6). Based on reproducibility and consistency metrics on real data (Fig. 3a and Supplementary Figs. 37b and 38c), CapTrap-ONT, CapTrap-PacBio, cDNA-PacBio and cDNA-ONT demonstrated similar performance: low IM and ACVC and high CM and ACC, outperforming dRNA-ONT and R2C2-ONT likely due to their lower sequencing depths (Fig. 1b). In particular, CapTrap-ONT and cDNA-ONT exhibited the lowest irreproducibility (mean IM = 0.19 and 0.20 and ACVC = 0.50 and 0.51) and highest consistency (mean CM = 0.89 and 0.86 and ACC = 9.49 and 9.51). For abundance resolution, cDNA-PacBio and R2C2-ONT outperformed others, with at least a twofold higher RE than cDNA-ONT (Fig. 3b). Notably, there were bimodal distributions of read length for some protocols-platforms (cDNA-PacBio, CapTrap-PacBio and definitely dRNA-ONT for R2C2-ONT (Supplementary Fig. 44). Varying sequencing error rates across platforms (Fig. 1b) suggest that tools may have specific advantages in handling certain data types (Fig. 3f and Extended Data Fig. 7).

For cell mixing experiments (Fig. 3c and Supplementary Fig. 40b), CapTrap-PacBio, CapTrap-ONT, cDNA-PacBio, cDNA-ONT and dRNA-ONT showed similar performances (mean SCC scores, 0.73 to 0.83), whereas the remaining R2C2-ONT scored below 0.60 in mean SCC. In particular, CapTrap-PacBio exhibited the best quantification accuracy, surpassing other long-read-based protocols-platforms and comparable to cDNA-Illumina.

For SIRV-Set 4 data, cDNA-PacBio outperformed other long-read-based protocols-platforms (Fig. 3d and Supplementary Fig. 41a), with the highest SCC (0.70 versus 0.60–0.66) and the lowest MRD (0.40 versus 0.58–0.75) and NRMSE (1.14 versus 1.38–1.52). cDNA-ONT followed and outperformed the other protocols-platforms. Notably, all protocols-platforms struggled to quantify ERCC transcripts with TPM > 0 (mean PET = 33.01%) compared to regular SIRV (mean PET = 82.17%) and long SIRV transcripts (mean PET = 69.75%). Particularly, cDNA-ONT, cDNA-PacBio, CapTrap-ONT and R2C2-ONT showed similar PET performance (34.39–43.27%) in ERCC quantification, surpassing dRNA-ONT (18.35%) and CapTrap-PacBio (27.99%). For long SIRV transcripts, except CapTrap-PacBio and dRNA-ONT, all could quantify over 70% of transcripts with TPM > 0. All performed well for regular SIRV transcripts, with dRNA-ONT, CapTrap-PacBio, cDNA-PacBio and cDNA-ONT being the most prominent (PET > 82.00%). Similar to SIRV-Set 4 data, the simulation study revealed cDNA-PacBio's superior performance in SCC, MRD and NRMSE compared to cDNA-ONT and dRNA-ONT (Fig. 3e and Supplementary Fig. 42b).

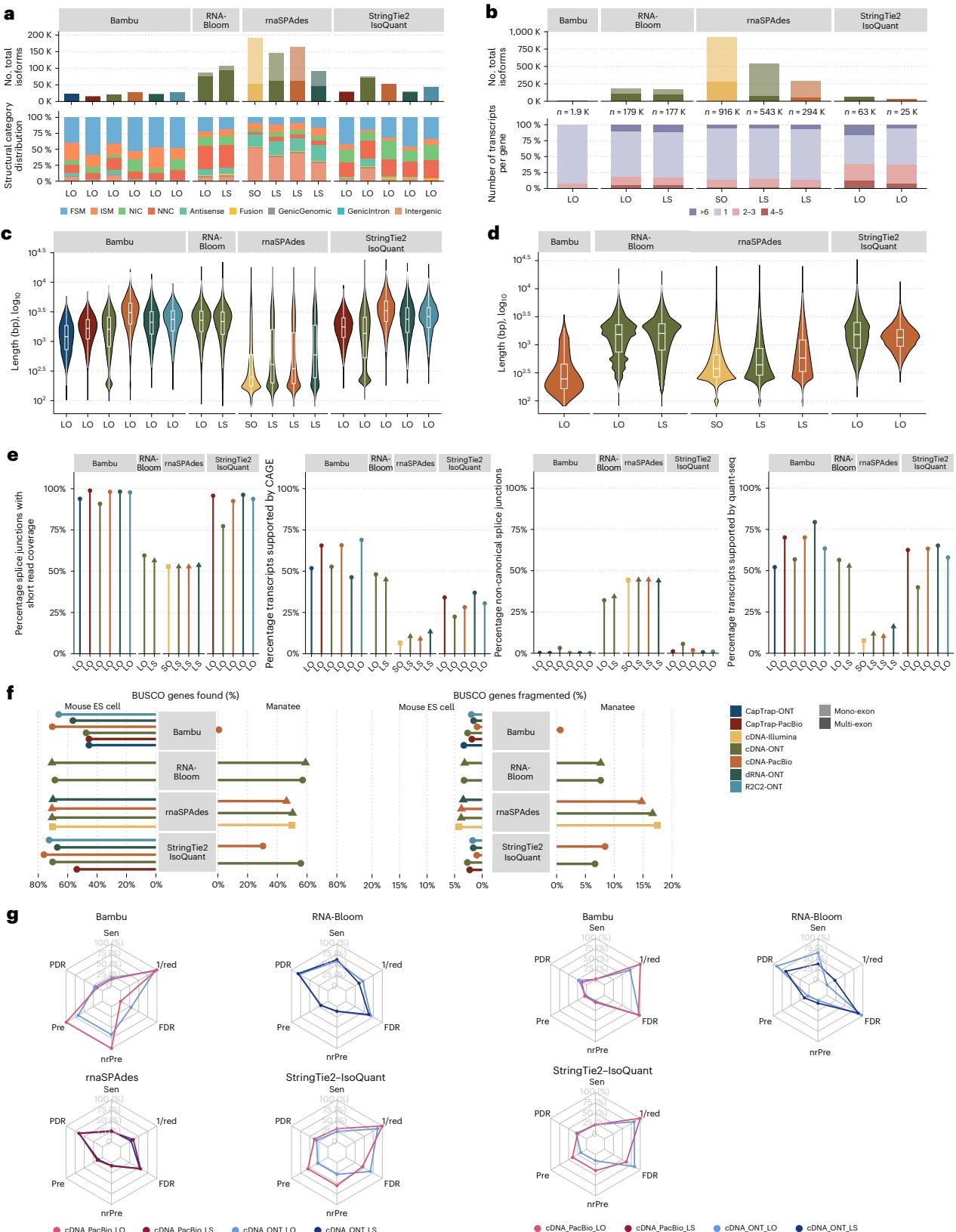

**Fig. 4 | Evaluation of transcript identification without a reference annotation for Challenge 3. a**, Number of detected transcripts and distribution of SQANTI structural categories, mouse ES cell sample. **b**, Number of detected transcripts and distribution of transcripts per loci, manatee sample. **c**, Length distribution of mouse ES cell transcripts predictions. Number of transcripts reported by each pipeline, in order, $n$ = 23,540, 15,054, 21,312, 27,215, 21,913, 27,056, 85,720, 107,832,

192,324, 144,752, 164,117, 91,833, 28,293, 75,106, 52,944, 29,458 and 44,079. **d**, Length distribution of manatee transcripts predictions. Number of transcripts reported by each pipeline, in order, $n$ = 1,911, 179,258, 176,895, 695,167, 535,845, 288,958, 63,000 and 25,643. **e**, Support by orthogonal data. **f**, BUSCO metrics. **g**, Performance metrics based on SIRVs. Sen, sensitivity; PDR, positive detection rate; Pre, precision; nrPred, non-redundant precision; SO, short only.

Finally, we evaluated tool performance across various gene/transcript sets grouped by transcript features, including abundance, isoform-exon number, length and a customized statistic $K$-value representing exon-isoform structure complexity (Fig. 3h and Supplementary Figs. 37c, 38d and 45).

For real data with multiple replicates, all tools showed reduced coefficient of variation (CV) and increased CM across six protocols-platforms with rising transcript abundances (Supplementary Figs. 37c and 38d). We further analyzed normalized MRD changes using human cDNA-PacBio simulation data (Fig. 3h). MRD scores on all tools spiked when transcript abundance was TPM ≤ 2, indicating heightened variability and errors in estimating abundance for low-expression transcripts. Moreover, tools exhibited poor performance at high $K$-values and isoform numbers, suggesting challenges in accurately quantifying more complex gene structures. Most tools performed well with isoforms with 5 to 15 exons, except for RSEM and Bambu. Notably, quantification errors were more pronounced for transcripts shorter than 1 kb, whereas tools exhibited varying performance for transcripts longer than 1 kb.

In summary, our evaluation revealed variable tool performance based on gene/transcript features, posing challenges in accurately quantifying low-expression and complex transcripts. Notably, tools exhibited performance differences across diverse data scenarios (Fig. 3f and Extended Data Figs. 6 and 7). Overall, RSEM outperformed long-read-based tools in different protocols-platforms and metrics (Fig. 3b–e). IsoQuant, FLAIR and Bambu stood out among long-read-based tools (Fig. 3f and Extended Data Fig. 7). IsoTools excelled in cDNA-PacBio data, whereas NanoSim performed best in SIRV transcript quantification. Generally, cDNA-Illumina showed top overall performance in different protocols-platforms, ranking among the top three in all metrics, except RE (Fig. 3g). Meanwhile, cDNA-PacBio, cDNA-ONT, CapTrap-ONT and CapTrap-PacBio demonstrated consistently good performance across various scenarios, surpassing dRNA-ONT and R2C2-ONT.

## Evaluation of de novo transcript detection

We assessed long-read methods for transcript identification without a reference in two scenarios: high-quality genome assembly and data (mouse ES cell sample) and limited genomic information on a field experiment (manatee leukocyte sample). Additionally, the manatee sample had excess SIRV spike-ins, representing a challenging dataset. A draft manatee genome was assembled using Nanopore and Illumina sequencing (Supplementary Fig. 46) and provided to submitters, but no genome annotation was allowed in Challenge 3 analyses. Matched short-read RNA-seq data were available to all submitters.

Four tools (Bambu, StringTie2+IsoQuant, RNA-Bloom[26] and rnaSPAdes[27]) submitted transcriptome predictions for both samples (Fig. 1f). Although overall transcript mapping rates were high (Supplementary Fig. 47), the number of detected transcripts varied, ranging from approximately 20,000 to 150,000 in mouse ES cells and from around 2,000 to 500,000 in the manatee sample (Fig. 4a,b and

Supplementary Data 12 and 13). rnaSPAdes predicted the largest number of transcripts and the highest fraction of noncoding sequences, followed by RNA-Bloom. Conversely, Bambu predicted the fewest transcripts (Fig. 4a). In the mouse sample, most detected transcripts were novel (Fig. 4a), contrasting with IsoQuant and Bambu Challenge 1 results using the reference annotation, highlighting the impact of annotation on predictions (Fig. 2a and Extended Data Fig. 8). Structural category analysis was not possible for the manatee sample, but examination of transcript counts per locus revealed variance among methods and data types. Bambu, rnaSPAdes and RNA-Bloom predicted a single transcript for most loci, whereas StringTie2+IsoQuant, especially with cDNA-ONT data, predicted two or more transcripts for nearly half of the loci (Fig. 4b).

In the absence of a reference annotation, Bambu, StringTie2+IsoQuant and RNA-Bloom predicted transcript models mainly between 1 kb and 3 kb. Bambu and StringTie2+IsoQuant reported many short transcripts in the mouse ES cell cDNA-ONT dataset (Fig. 4c), likely influenced by shorter reads (Fig. 1c), with Bambu showing shorter transcripts in the manatee cDNA-PacBio dataset (Fig. 4d). rnaSPAdes generated numerous short transcripts, affecting overall length distributions (Fig. 4c,d).

For mouse ES cell transcripts, a link was observed between the number of predicted transcripts and their orthogonal data support. rnaSPAdes, with the most predictions, had the least support from Illumina, CAGE and QuantSeq datasets and a high percentage of non-canonical splice junctions. Conversely, Bambu had fewer predictions but higher orthogonal support. RNA-Bloom showed moderate support and many non-canonical junctions. StringTie2+IsoQuant's transcripts had good junction quality but low CAGE support (Fig. 4e).

Most transcripts identified by Bambu and StringTie2+IsoQuant in the mouse ES cell sample were protein-coding, except in CapTrap-ONT and cDNA-ONT datasets, where about 25% were noncoding, possibly due to the higher number of reads in these datasets. In the manatee sample, a lower percentage of transcripts were predicted as coding, with about 70% for IsoQuant and Bambu and less than 20% for rnaSPAdes (Supplementary Fig. 48).

BUSCO[28] (benchmarking sets of universal single-copy orthologs), a database of highly conserved genes, was used to assess transcriptome completeness, showing good performance across most tools despite the observed differences in protein-coding transcript rates. In the mouse ES cell sample, rnaSPAdes and RNA-Bloom detected over 60% of complete BUSCO genes, whereas Bambu reached this only with cDNA-PacBio and R2C2-ONT data. In the manatee sample, IsoQuant and RNA-Bloom had the highest BUSCO completeness (~50%) on Nanopore datasets, with rnaSPAdes at around 30% and Bambu performing poorly. Incomplete BUSCO genes were generally fewer in the mouse ES cells than in the manatee, with rnaSPAdes showing the highest ratio of incompleteness in the manatee sample (Fig. 4f).

SIRV spike-in analysis showed notable tool and sample variations (Fig. 4g). RNA-Bloom detected SIRVs in the mouse ES sample with about 70% sensitivity but had low precision and a high false discovery rate (FDR). rnaSPAdes exhibited low sensitivity and a high positive

**Fig. 5 | Experimental validation of known and novel isoforms. a,** Schematic for the experimental validation pipeline. QC, quality control **b,** Example of a consistently detected NIC isoform (detected in over half of all LRGASP pipeline submissions), which was successfully validated by targeted PCR. The primer set amplifies a new event of exon skipping (NIC). Only transcripts above ~5 CPM and any part of the GENCODE Basic annotation are shown. **c,** Example of a successfully validated new terminal exon, with ONT amplicon reads shown in the IGV track (PacBio produces similar results). **d,** Recovery rates for GENCODE-annotated isoforms that are reference matched (known), novel and rejected. **e,** Recovery rates for consistently versus rarely detected isoforms for known and novel isoforms. **f,** Recovery rates between isoforms that are more frequently identified in ONT versus PacBio pipelines. **g–i,** Relationship between

estimated transcript abundances (calculated as the sum of reads across all WTC11 sequencing samples) and validation success for GENCODE (**g**), consistent versus rare (**h**) and platform-preferential (**i**) isoforms. NV, not validated; V, validated. The number of transcripts in each category is shown in **d–f. j,** Fraction of validated transcripts as a function of the number of WTC11 samples in which supportive reads were observed. **k,** Example of two de novo isoforms in manatee validated through isoform-specific PCR amplification. Purple corresponds to the designed primers, orange to the possible amplification product associated with one isoform and black to the predicted isoforms. **l,** PCR validation results for manatee isoforms for seven target genes. Blue corresponds to supported transcripts and red to unsupported transcripts. The figure was partially created with BioRender.com.

detection rate, suggesting incomplete transcript model detections. RNA-Bloom and rnaSPAdes often predicted multiple models for the same SIRV (low1/redundancy values). Conversely, StringTie2+IsoQuant

and Bambu showed lower sensitivity (~25%) but better precision and FDR control, particularly in cDNA-PacBio data. In the manatee sample, where SIRVs were abundant, performance dropped across all tools,

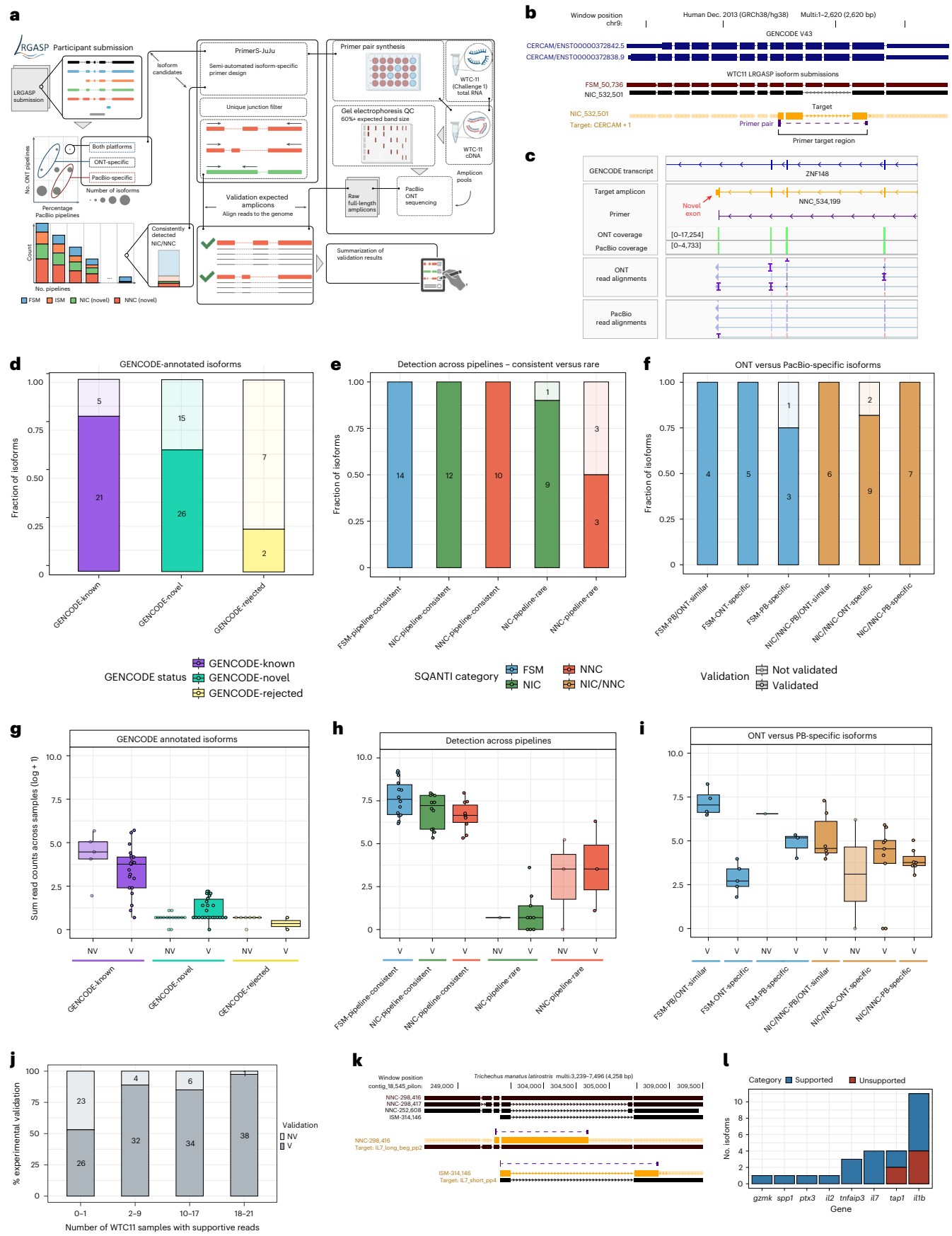

with Bambu failing to recover spiked RNAs. SQANTI3 analysis against annotated SIRV data indicated a majority of SIRV reads were FSM and most SIRVs had at least one reference-match read in both manatee datasets (Supplementary Fig. 49) suggesting that data quality was not a limiting factor.

Transcript detection without reference annotation proved challenging. Notably, transcripts with higher coverage, as in our SIRV spike-ins in the manatee sample, led to poorer performance for all tools, suggesting that accurate detection from highly expressed genes may be problematic. Bambu and IsoQuant had moderate to good precision but low sensitivity, RNA-Bloom had high sensitivity but low precision and rnaSPAdes generated fragmented, short transcripts with high FDR.

### Experimental validation of transcript predictions

To experimentally validate isoforms, we targeted isoform-specific regions for PCR amplification followed by gel electrophoresis and sequencing of pooled amplicons via ONT and PacBio sequencing (Fig. 5a). We prioritized validation of three comparison groups: (1) GENCODE-annotated (known or novel) based on LRGASP data and annotation; (2) consistently identified by >50% of pipelines or rarely by 1–2 pipelines; and (3) preferentially identified in ONT or PacBio libraries. From these comparison groups, we designed primers for 178 target regions, in which the length of the amplified region ranged from 120 to 4,406-bp long, with a median of 488 bp and 25–75 interquartile range of 305 to 795 bp (Fig. 5a). Examples of a validated exon-skipping event (NIC) and a novel terminal exon (NNC) are shown in Fig. 5b,c.

To evaluate GENCODE-annotated isoforms, we compared groups of randomly selected isoforms that were (1) annotated (GENCODE-known, $n = 26$); (2) novel and confirmed through manual annotation (GENCODE-novel, $n = 41$); and (3) unsupported isoforms that were investigated but did not pass rigorous manual curation (GENCODE-rejected $n = 9$). As expected, we found a high validation rate for GENCODE-known, 81% (Fig. 5d). Of the GENCODE-known isoforms, we found that 5 of the 28 targets failed to validate despite orthogonal support[29]; therefore, we speculate that they failed due to suboptimal primer or PCR conditions. GENCODE-novel isoforms validated at a slightly lower validation rate (63%) compared to GENCODE-known. Further review confirmed that GENCODE-novel isoforms that failed to validate tended to be lower in abundance compared to their successfully validated counterparts (Fig. 5g). Only two of nine GENCODE-rejected isoforms were amplified, which were later confirmed to be mismapped due to tandem repeats.

A large number of novel isoforms were detected in this study (for example, 279,791 new isoforms in WTC11; Fig. 1b). We found that 743 novel isoforms were detected consistently, but a vast majority or 242,125 isoforms, were rarely detected and found in only one or two of the pipelines. We obtained a 100% validation rate for consistently detected new isoforms (Fig. 5e). For isoforms with low reproducibility across pipelines, we found a surprisingly high validation rate of 90% and 50% for NIC and NNC isoforms, respectively. Abundance correlated with validation rate (Fig. 5h), as found for the GENCODE validation set.

Last, we determined the validation rates of known and novel isoforms in common or preferentially detected in the cDNA-ONT or cDNA-PacBio experiments. For example, an isoform detected in more than 50% of ONT pipelines but less than 50% of PacBio pipelines would be considered ONT-preferential and vice versa. We found that all known and new isoforms found frequently across both platforms were validated (Fig. 5f,i) and most validated isoforms were identified by amplicon sequencing on both ONT and PacBio. We acknowledge that this validation set is a relatively small sample size, which limits drawing general conclusions on validation rates for platform-preferential isoforms.

Validation experiments using long-read transcript models suggest high accuracy for novel isoform predictions, even if not consistently detected across pipelines and platforms. Validation success seems linked to the isoform detection frequency, measured by either the number of samples (combinations of library preparation and sequencing technology) detecting the isoform (Fig. 5j) or the total read counts supporting it (Extended Data Fig. 9).

To validate long-read-based isoform discovery without a reference annotation, we focused on the manatee dataset. Challenge 3 had fewer submissions than Challenge 1; therefore, we established a goal of not explicitly comparing pipelines but rather assessing the ability of the long-read RNA-seq datasets to return accurate transcript isoform annotation.

Seven genes related to immune pathways and their respective isoforms were manually selected based on visualization on a custom UCSC Genome Browser track. We designed 22 primers that could potentially amplify 26 transcript predictions. The length of the amplified region ranged from 78 to 2,633-bp long, with a median of 1,038 bp and a 25–75 interquartile range of 379 to 1,379 bps. Validation of targets was confirmed by PacBio sequencing of the amplicons (Fig. 5k). For the five genes with few isoforms, all isoforms were validated. For the two genes for which many isoform models were predicted with more variability across participants, approximately half of the targets were validated (Fig. 5l and Supplementary Fig. 50).

Overall, we find a greater variability of 'field collected' non-model organism long-read data compared to the human or mouse datasets. Though our sample population was small, we found that isoforms predicted by many pipelines tended to be validated.

### Final recommendations

Based on the results of LRGASP, the Consortium recommends the following suggestions to improve the analysis of transcriptomes using long reads:

1. For transcript identification, longer and more accurate sequences are preferable to having more reads. Therefore, the cDNA-PacBio and R2C2-ONT datasets are the best options. If the goal is quantification, especially on a well-annotated reference, higher throughput sequencing, such as with cDNA-ONT and CapTrap-ONT, are the best choice.
2. When choosing a bioinformatics tool, it is crucial to consider the study's objective:
   a. If the goal is to identify a sample-specific transcriptome in a well-annotated organism when only minimal new transcripts are expected, Bambu, IsoQuant and FLAIR are the most effective.
   b. If the aim is to detect lowly expressed or rare transcripts, use a tool that allows novelty and includes orthogonal data. Mandalorion and FLAIR, combined with short reads, are among the best performers, with a good balance of sensitivity and precision. For identifying a conservative set of highly supported novel transcripts, LyRic is an effective tool. Experimental validation of rare transcripts is recommended.
3. If quantification is essential, IsoQuant, FLAIR and Bambu are the best options and can perform comparably to short-read tools.
4. To create a reference for genome annotation, we recommend using high-quality data, including replicates, using extensive orthogonal data, imposing a transcript-level filter and using transcripts identified from more than one analysis tool.

## Discussion

This LRGASP study revealed that increased read quantity does not always lead to more accurate transcripts, emphasizing the importance of read quality and length. ONT sequencing of cDNA and CapTrap libraries produced many reads, whereas cDNA-PacBio and R2C2-ONT gave the most accurate ones. The choice of analysis tool notably influenced results, with some favoring known transcripts and others more sensitive to novel ones. Different approaches varied in their handling of RNA degradation and library preparation artifacts.

Performance differed based on ground-truth data, with tools excelling in known transcripts but varying in simulated and manual annotations. Challenge 2 highlighted key factors affecting transcript quantification tool performance, including annotation accuracy and read length distributions. De novo annotation remains challenging, with inconsistent results in Challenge 3 indicating a need for further tool development. Experimental validation demonstrated long-read methods' effectiveness in uncovering transcriptome complexity in non-model organisms.

Our benchmark has limitations. LRGASP did not evaluate long-read mapping methods and all tools used minimap2 (ref.[30]) as the aligner, leading to mapping errors identified by the GENCODE manual curation. Participants submitted only one set of predictions per dataset, configuring their methods as they saw fit. Furthermore, to facilitate participation, submitters were allowed to choose which data modalities they contributed to, and many chose a subset, resulting in unbalanced data when evaluating methods. We tried to consider this in our analyses, but balanced participation would have been more useful. Several prominent long-read analysis tools[31–34] did not participate in LRGASP or were introduced later. To address this, we have made LRGASP datasets and evaluation strategies available on OpenEBench[35] for ongoing tool benchmarking and development. This platform facilitates the creation of new lrRNA-seq analysis tools. Finally, as data continue to evolve, conclusions may need revision as sequencing technologies improve.

Overall, LRGASP demonstrates the power of community collaboration in assessing evolving technologies. Its insights pave the way for further advancements in long-read RNA-seq analysis, ultimately unlocking a deeper understanding of gene expression and regulation.

## Online content

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

## Methods

### Computational evaluation of transcript isoform detection and quantification

**Evaluation of transcript isoform detection for Challenge 1.** Four sets of transcripts were used for the evaluation of transcript calls made on human and mouse lrRNA-seq data:

1. Lexogen SIRV-Set 4 (ref. [36]) (SIRV-Set 3 plus 15 new long SIRVs with sizes ranging from 4 to 12 kb).
2. Comprehensive GENCODE annotation: human v.39, mouse v.M28. GENCODE human v.38 and v.M27 were available at the time of the LRGASP data release and new versions of GENCODE were released after the close of LRGASP submissions.
3. A set of transcripts from a subset of undisclosed genes that were manually annotated by GENCODE. These transcripts are considered high-quality models derived from LRGASP data.
4. Simulated data for both Nanopore (NanoSim) and PacBio (IsoSeqSim) reads.

The rationale for including these different types of transcript data is that each set creates a different evaluation opportunity but also has limitations. For example, SIRVs and simulated data provide a clear ground truth that allows the calculation of standard performance metrics such as sensitivity, precision or FDR. Evaluation of SIRVs can identify potential limitations of both library preparation and sequencing, but the SIRVs themselves represent a dataset of limited complexity and the SIRV annotation is known to submitters. Higher complexity can be generated when simulating long reads based on actual sample data; however, read simulation algorithms only capture some potential biases of the sequencing technologies (for example, error profiles) and not of the library preparation protocols. In any case, both types of data approximate but do not fully recapitulate real-world datasets. Evaluation against the GENCODE annotation[37] represents this real dataset scenario, although in this case, the ground truth is not entirely known. This limitation was partially mitigated by the identification of a subset of GENCODE transcript models that were manually annotated by GENCODE annotators and by follow-up experimental validation for a small set of transcripts using semi-quantitative PCR with reverse transcription (RT–PCR). In this way, although an exhaustive validation of the real data is not possible, estimates of the methods' performances can be inferred. By putting together evaluation results obtained with all these different benchmarking datasets, insights can be gained on the performance of the library preparation, sequencing and analysis approaches both in absolute and relative terms.

The evaluation of the transcript models was guided by the use of SQANTI3 categories (Box [1]), implemented in the SQANTI3 software. It incorporated additional definitions and performance metrics to provide a comprehensive framework for transcript model assessment. The evaluation considers the accuracy of the transcript models both at splice junctions and at 3′ and 5′ transcript ends. It took into account external sources of evidence such as CAGE and QuantSeq data, poly(A) annotation and support by Illumina reads. The evaluation script was provided to participants at the time of data release.

Given the LRGASP definitions, evaluation metrics were specified for each type of data type.

*SIRVs.* To evaluate SIRVs, we extracted from each submission all transcript models that were associated with SIRV sequences after SQANTI3 analysis. This includes FSM and ISM isoforms of SIRVs and NIC, NNC, antisense and fusion transcripts mapping to SIRV loci. The metrics for SIRV evaluation are shown in Supplementary Table 9.

*Simulated data.* The simulated data contained both transcript models based on the current GENCODE annotation and several simulated novel transcripts that will result in valid NIC and NNC annotations. Transcript models generated from simulated data were analyzed by

SQANTI3, providing a GTF file that includes all simulated transcripts (GENCODE and novel) and excludes all transcripts for which reads were not simulated. The evaluation metrics for simulated data are shown in Supplementary Table 10.

*Comprehensive GENCODE annotation.* Submitted transcript models were analyzed with SQANTI3 using the newly released GENCODE annotation (v.39 for human and M28 for mouse) and different metrics were obtained for FSM, ISM, NIC, NNC and 'other' models (Supplementary Table 8).

*High-confidence transcripts derived from LRGASP data.* Finally, a set of manually curated transcript models was used to estimate the sensitivity and precision on real data. Metrics that were applied in this transcript set are TP, PTP and FN, sensitivity, positive detection rate, redundancy and %LRC (Supplementary Table 11).

*Analysis of transcript model identification across pipelines.* We evaluated the characteristics of the transcripts detected as a function of the experimental factors of the LRGASP study, for example, sequencing platform or library protocol. To do that, we compared detected transcripts across pipelines at the level of UICs, allowing for variability in the 3′ and 5′ definitions, and annotated the pipelines that detected each UIC. The location of BED files of the UICs consolidated models can be found in the Data Availability section.

Transcript models were visualized in the UCSC Genome Browser[38] using the Track Hub[39] facility. The track hub displayed consolidated transcript models from the submissions with metadata, color-coding and filtering by attributes. This allowed us to efficiently explore the significant quality of LRGASP results in the genomic context.

**Evaluation of transcript isoform quantification for Challenge 2.** We evaluated transcript isoform quantification performance with four data scenarios (real data with multiple replicates, cell mixing experiment, SIRV-Set 4 data and simulation data). We designed nine metrics for performance assessment, both with and without known ground truth (Box [1] and Fig. [3a]). The participants of Challenge 2 were able to run these evaluations by submitting their quantification results at the website at https://lrrna-seq-quantification.org/, which generates an interactive report in HTML and PDF formats (see Data Availability and Code Availability).

*Ground truth is available.* We evaluated how close the estimations and the ground-truth values are by three metrics as follows:

denote $\hat{\Theta} = (\hat{\theta}_1, \hat{\theta}_2, \cdots, \hat{\theta}_I)^T$ and $\Theta = (\theta_1, \theta_2, \cdots, \theta_I)^T$ as the estimation and ground truth of the abundance of transcript isoforms in a sample, respectively. Here, we use TPM as the unit of transcript abundance. Then, four metrics can be calculated by the following formulas:

SCC: this evaluates the monotonic relationship between the estimation and the ground truth, which is based on the rank for transcript isoform abundance. It is calculated by

$$SCC_{\Theta,\hat{\Theta}} = \frac{\text{cov}\left(rg_\Theta, rg_{\hat{\Theta}}\right)}{S_{rg_\Theta} \cdot S_{rg_{\hat{\Theta}}}},$$

where $rg_\Theta$ and $rg_{\hat{\Theta}}$ are the ranks of $\Theta$ and $\hat{\Theta}$, respectively and $\text{cov}\left(rg_\Theta, rg_{\hat{\Theta}}\right)$ is the covariance of the corresponding ranks and $S_{rg_\Theta}$ and $S_{rg_{\hat{\Theta}}}$ are the sample s.d. of $rg_\Theta$ and $rg_{\hat{\Theta}}$, respectively.

MRD: this is the median of the relative difference of abundance estimates among all transcript isoforms within a sample and is calculated by

$$MRD = \text{median}\left\{ \frac{|\theta_i - \hat{\theta}_i|}{\theta_i}, (i = 1, 2, \cdots, I) \right\}.$$

A small MRD value indicates a good performance of abundance estimation.

NRMSE: this provides a measure of the extent to which the one-to-one relationship deviates from a linear pattern. It can be calculated by

$$\text{NRMSE} = \frac{\sqrt{\frac{1}{I}\sum_{i=1}^{I}\left(\theta_i - \hat{\theta}_i\right)^2}}{s_\Theta},$$

where $s_\Theta$ is the sample s.d. of $\Theta$. A good performance of abundance estimation should have a small value of NRMSE.

In the case of LRGASP, the above metrics were calculated using the cell mixing experiment, simulation data and SIRVs.

*Ground truth is unavailable.* For multiple replicates under different conditions without the ground truth, we evaluated a quantification method by the 'goodness' of its statistical properties, including irreproducibility, consistency and RE that is also calculated for single-sample data.

IM: this statistic characterizes the average CV of abundance estimates among different replicates (Supplementary Fig. 37), which is calculated by

$$\text{IM} = \sqrt{\frac{1}{IG}\sum_{i=1}^{I}\sum_{g=1}^{G} CV_{ig}^2}$$

Here, $CV_{ig}$ is the CV of $\log\left(\hat{\theta}_{igr}+1\right)$ $(r=1,2,\cdots,R)$, which is calculated by

$$CV_{ig} = \frac{s_{ig}}{u_{ig}},$$

where $s_{ig}$ and $u_{ig}$ are the sample s.d. and mean of abundance estimates, which are calculated by

$$s_{ig} = \sqrt{\frac{1}{R}\sum_{r=1}^{R}\left(\log\left(\hat{\theta}_{igr}+1\right)-u_{ig}\right)^2},$$

$$u_{ig} = \frac{1}{R}\sum_{r=1}^{R}\log\left(\hat{\theta}_{igr}+1\right).$$

By plotting $CV_{ig}$ versus average abundance $u_{ig}$, we examined how the coefficient of variation changes with respect to the abundance and the ACVC was calculated as a secondary statistic. With a small value of irreproducibility and ACVC scores, the method has high reproducibility.

Consistency: a good quantification method consistently characterizes abundance patterns in different replicates. Here, we propose a CM $C(\alpha)$ to examine the similarity of abundance profiles between mutual pairs of replicates (Supplementary Fig. 38), which is defined as:

$$C(\alpha) = \frac{1}{IG \cdot C_R^2}\sum_{i=1}^{I}\sum_{g=1}^{G}\sum_{1 \leq r_1 < r_2 \leq R} P\left(\left\{\log\left(\hat{\theta}_{igr_1}+1\right) < \alpha, \log\left(\hat{\theta}_{igr_2}+1\right) < \alpha\right\}\right.$$

$$\left. or \left\{\log\left(\hat{\theta}_{igr_1}+1\right) \geq \alpha, \log\left(\hat{\theta}_{igr_2}+1\right) \geq \alpha\right\}\right),$$

where $\alpha$ is a customized threshold defining whether a transcript is expressed or not.

We plotted the abundance threshold $\alpha$ versus CM $C(\alpha)$ to evaluate how $C(\alpha)$ changes with respect to the abundance threshold. The ACC can be used as the second metric to characterize the degree of similarity of transcript expression. With a large value of consistency and ACC scores, the method has a higher similarity of abundance estimates among multiple replicates.

RE: a good quantification method should have a high resolution of abundance values. For a given sample, an RE statistic characterizes the resolution of abundance estimation (Supplementary Fig. 39):

$$\text{RE} = -\sum_{m=1}^{M} P_m \ln(P_m), \text{ where } P_m = \frac{n_m}{\sum_{j=1}^{M} n_j}$$

Here, the abundance estimates are binned into $M$ groups, where $n_m$ represents the number of transcript isoforms with the abundance estimate $\hat{\Theta} \in [m \cdot \alpha, (m+1) \cdot \alpha]$ and $\alpha = \max(\hat{\Theta})/M$. RE = 0 if all transcript isoforms have the same estimated abundance values, while it obtains a large value when the estimates are uniformly distributed among $M$ groups.

Evaluation with respect to multiple transcript features: different transcript features, such as exon-isoform structure and the true abundance level could influence quantification performance. Thus, we also evaluated the quantification performance for different sets of genes/transcripts grouped by transcript features, including a number of isoforms, number of exons, ground-truth abundance values and a customized statistic $K$-value representing the complexity of exon-isoform structures.

$K$-value: most methods for transcript isoform quantification assign sequencing coverage to isoforms; therefore, the exon-isoform structure of a gene is a key factor influencing quantification accuracy. Here, we used a statistic $K$-value (Supplementary Fig. 45; H.L., manuscript in preparation) to measure the complexity of exon-isoform structures for each gene. Suppose a gene of interest has $I$ transcript isoforms and $E$ exons and $A = (a_{ie}), (i = 1,2,\cdots,I; e = 1,2,\cdots,E)$ is the exon-isoform binary matrix, where

$$a_{ie} = 1, \text{ if the isoform i includes the exon e, } 0, \text{ otherwise.}$$

$K$-value is the condition number of the exon-isoform binary matrix $A$, which is calculated by

$$K \text{ value} = \frac{\sigma_{\max}(A)}{\sigma_{\min}(A)},$$

where $\sigma_{\max}(A)$ and $\sigma_{\min}(A)$ are the maximum and minimum singular values of the matrix $A$, respectively.

**Evaluation of de novo transcript isoform detection without a high-quality genome for Challenge 3.** Challenge 3 evaluated the applicability of lrRNA-seq for de novo delineation of transcriptomes in non-model organisms to assess the capacity of technologies and analysis pipelines for both defining accurate transcript models and for correctly identifying the complexity of expressed transcripts at genomic loci when genome information is limited.

The challenge includes three types of datasets. The mouse ES cell transcriptome data (Supplementary Table 2) was used to request the reconstruction of mouse transcripts without making use of the available genome or transcriptome resources for this species. Models were compared to the true annotations set with the same parameters as in Challenge 1. As FASTA rather than GTF files were submitted in Challenge 3, we used the same mapper, minimap2, for all the submissions to transform sequence information into a genome annotation file.

While this dataset allows for a quantitative evaluation of transcript predictions in Challenge 3, it might deliver unrealistic results if analysis pipelines were somehow biased by information derived from previous knowledge of the mouse genome. To avoid this problem, a second dataset was used corresponding to the Florida manatee's whole-blood transcriptome (*Trichechus manatus latirostris*). An Illumina draft genome of this organism exists (GCF_000243295.1) and the LRGASP consortium has generated a long-read genome assembly to support transcript predictions for this species (GCA_030013775.1).

Additionally, Illumina RNA-seq data were generated from manatee blood samples. As no curated gene models exist for the manatee, Challenge 1 metrics cannot be applied. Instead, the evaluation of Challenge 3 submissions involved a comparative assessment of the reconstructed transcriptomes and experimental validation (see 'Experimental validation of transcript models' section). For computational assessment, the following parameters were calculated:

a. Total number of transcripts
b. Mapping rate of transcripts to the draft genomes (for pipelines not using genome data)
c. Length of the transcript models
d. Number of mono- and multi-exon transcripts
e. Percentage of junctions with Illumina coverage
f. Percentage of transcripts with Illumina coverage at all junctions
g. Percentage junctions and transcripts with non-canonical splicing
h. Percentage of transcripts with predicted coding potential
i. Predicted RT-switching incidence
j. Predicted intra-priming
k. Number of transcripts/loci
l. BUSCO[28] analysis:

   i. Number of complete BUSCO genes detected by a single transcript (complete single-copy)
   ii. Number of complete BUSCO genes detected by multiple transcripts (completely duplicated)
   iii. Number of fragmented BUSCO genes detected (fragmented)
   iv. Number of BUSCO genes not detected (missing)

The BUSCO analysis used the eutherian BUSCO (lineage *eutheria_odb10*) gene set. We do not expect a BUSCO-complete transcriptome recovery as only one tissue or cell type per organism was sequenced. We expect that good-performing pipelines obtain longer transcripts, well supported by Illumina data, with a high mapping rate to the draft genomes, most of them coding and with a higher number of complete BUSCO genes.

Finally, as both the mouse and manatee datasets contained spiked-in SIRVs, they were used as the third dataset to compute performance for Challenge 3 using the same type of metrics as described for Challenge 1.

**GENCODE manual annotation**

Expert GENCODE human annotators sought to establish the baseline for annotating genuine alternative isoforms using the long transcriptomic data generated by the LRGASP consortium at selected loci. An exhaustive and fully manual investigation of all aligned reads from the sequence data generated by the LRGASP consortium was undertaken at selected loci, and all isoforms passing GENCODE annotation criteria that were present were captured as annotated transcript models.

GENCODE expert human annotation has very high sensitivity and specificity as every read can be individually considered on its merits for use in supporting a transcript model that is subsequently included in the annotation set. Consequently, the fully human annotation process where every read is manually reviewed and every transcript model built manually is time-consuming. The speed of the process limits the number of loci that could be considered for the LRGASP project, with 50 human and 50 mouse loci being selected.

The selection of loci was random within constraints based on the properties of the locus, identification of aligned reads in all libraries sequenced (where possible), the number of aligned reads and an indication of the presence of valid isoforms at the locus.

Properties required of the locus to be considered for annotation included the presence of multiple exons and compactness. Loci with very long introns were excluded. While criteria required that (where possible) all libraries had at least one aligned read at the locus, a maximum number of aligned reads also had to be added. This additional

requirement was necessary for two reasons: first, to allow the manual consideration of every read at the locus to determine whether it could support an isoform, and second, because the large number of reads generated from the LRGASP sequencing experiments exposed bugs affecting the consistent display of transcripts when loaded into the Otter/Zmap[40] tools used for manual annotation, raising the possibility of erroneous exclusion of reads and transcripts that failed to display properly.

Preliminary analysis was conducted to identify plausible alternative splicing in aligned reads using Tmerge[5] at most permissive settings to create a set of putative transcripts from the LRGASP long transcriptomic data. The introns of this set of putative transcripts were assessed using recount3 (ref. [29]) data and those putative transcripts where all introns were supported by at least two RNA-seq reads from the GTEx[41] dataset captured by recount3 analysis were considered to be alternatively spliced for locus selection. The transcript models generated at this stage were only used to select loci for manual annotation and were not directly included in the GENCODE annotation.

A long list of human loci fitting the criteria for annotation was then compared to an equivalent list derived from a similar analysis in mouse and candidate loci were defined. Human and mouse shortlists were manually reviewed to confirm that the selection criteria were met. Genuine alternative splicing events were maximized, and, where possible, orthologous loci were selected for humans and mice.

Expert human annotation was carried out independently for each library prep method. Independence was defined as not using reads from one library prep to support reads in another. For example, where a longer or higher-quality read from one library prep method could support the interpretation of a truncated or low-quality read in another library, it was not used to support the extension of a transcript model supported by the shorter or lower-quality read.

Effectively, this necessitated 12 independent sets of annotation, 6 in humans and 6 in mice, to support complete flexibility in downstream analysis.

While annotation was performed independently for each library, orthogonal data external to LRGASP were used to support the interpretation of the long transcriptomic data. Specifically, recount3 intron data were used to support the interpretation of splice sites and Fantom CAGE[42] data, the definition of TSS and thereby the 5' completeness of a transcript. External long transcriptomic datasets were not used to support this annotation in any way.

All transcript models passing standard Ensembl-GENCODE manual annotation criteria for splicing and supported by at least one long transcriptomic read were annotated as transcript models. GENCODE annotation criteria require that introns are canonical (GT-AG, GC-AG or AT-AC with evidence of U12 splicing) or those non-canonical introns are supported by evidence of evolutionary conservation or constraint of the splice site. Read data were required to align such that a canonical intron could be unambiguously resolved, which generally requires that there is no equally plausible alignment of the read that could give a 'non-canonical intron', for example, where sequence aligns equally well at the putative donor and acceptor splice site but can be forced into a canonical splice site by the initial alignment method or where a read has an indel near a splice site that leads to an error in its initial alignment. Where necessary, annotators could realign the read to the genomic sequence using various methods, including the Exonerate pairwise alignment software and the Dotter[43] dot-plot tool. Introns identified by spanning RNA-seq reads by the recount3 project were used as orthogonal data to support the interpretation of splice sites.

Transcript models were extended to the full length of the homology between the read (or reads) supporting a transcript model and the genome sequence. The 5' transcript ends were not modified (clipped or extended) based on annotation already present at the locus before LRGASP (including the MANE Select[44] transcript). While Fantom CAGE data were used to identify TSSs at loci in both humans and mice,

CAGE data were not used to modify the TSSs. Similarly, at the 3′ end, transcripts were extended to the full length of the homology between the read and the genome. Where a poly(A) site was identified, the transcript model was not extended further unless another, longer read with the same intron chain was identified. In addition to the annotation of alternative isoforms, expert human annotators also defined sets of polyadenylation sites and signals at every locus. Again, these were annotated independently per library prep based on the presence of a poly(A) tail on one or more reads aligned to the locus and a poly(A) site hexamer (AATAAA or ATTAAA) within 50 bases upstream of the poly(A) site. Multiple poly(A) sites and their corresponding poly(A) signals could be annotated at any locus; however, only one poly(A) site was annotated per poly(A) signal.

In total, 635 transcript models and 641 poly(A) sites were annotated in humans. The mean number of isoforms annotated per locus per library ranged from 2.58 (cDNA-PacBio) to 1.28 (CapTrap-PacBio) in humans and from 2.5 (cDNA-PacBio) to 1.38 (dRNA-ONT) in mice. The mean number poly(A) sites annotated per locus per library ranged from 2.42 (cDNA-PacBio) to 1.12 (dRNA-ONT) in humans and 2.26 (CapTrap-ONT) to 1.2 (dRNA-ONT) in mice. The loci with the most annotated alternative splicing were ANGPT1 (cDNA-PacBio library) RERG ZADH2 (CapTrap-ONT) with six isoforms. ZADH2 also had the most poly(A) sites, with 12 reported in three libraries (cDNA-PacBio, cDNA-ONT and CapTrap-ONT). For mice, nine isoforms were annotated at Rcan1 (cDNA-PacBio), whereas Gan displayed the most diversity in polyadenylation with ten poly(A) sites annotated (cDNA-PacBio).

Expert human annotators reviewed transcript models that were included in the annotation set but failed to validate by RT−PCR and also, a set of reads was rejected as supporting valid transcript models by the first pass annotation. In both cases, the initial annotation of the transcript model was supported by review (Supplementary Data 14).

### Experimental validation of transcript models
**Gene selection process in WTC11 cells for Challenge 1.** To semi-systematically select isoforms for comparison in the validation experiments, we binned isoforms based on the frequency by which they were detected in certain pipeline parameters. Isoform test groups were defined based on their presence across various pipelines and library preparations. In general, during the isoform selection stage, we prioritized isoforms with expression higher than 10 TPM, wherever possible, and isoforms that contained distinguishable sequence regions. During the primer design process, we considered as 'present' all isoforms in the GENCODE annotation and all isoform models submitted by all participants.

We chose groups of novel isoforms (NIC or NNC) that were preferentially detected by pipelines using ONT versus PacBio platform, using the cDNA preparations, and a control group of isoforms frequently detected by all pipelines across both platforms. Additionally, we chose known (FSM) isoforms for all three of these groups.

**Targeted PCR validation of isoforms in WTC11.** cDNA was synthesized from replicate 1 and replicate 3 of the WTC11 total RNA and used as a template for two sets of PCR reactions, respectively. After targeted PCR, amplicons were analyzed via agarose electrophoresis and sizes were estimated. We found that at least 60% of all targets produced a single band corresponding to the expected amplicon size, indicating a moderately high success rate. All amplicons were pooled and sequenced with ONT minION (R10.4.1) and PacBio Sequel II. Amplicon reads were aligned to the genome as well as expected target sequences (a subset of the test transcript) and all targets with at least one gapless (<2 bp), high identity long-read alignment was considered validated.

**Analysis of the amplicon reads to determine the support of targets.** Using minimap2 (v.2.24-r1122), we aligned the RT−PCR sequences to the human genome assembly, with the targeted transcripts serving as

junction specifications. We aligned the expected amplicon sequences extracted from the genome to control for difficult-to-align cases. We aligned the ONT sequences with the 'splice' minimap2 preset, while using the 'splice:hq' preset for the PacBio and control sequences.

We aligned the WTC11 RT−PCR sequences to the GRC3h38 assembly, which includes SIRVs and an EBV sequence provided to LRGASP participants (syn25683364). In addition to genomic alignments, we aligned the RT−PCR sequences to a reference composed of the predicted amplicon sequences. Furthermore, we aligned the amplicon alignments to the amplicon reference. This approach facilitated the identification of difficult-to-align amplicons to the genome and cases where alignments to different isoforms might not be detected. For ONT sequences, we used the minimap2 'map-ont' preset for amplicon reference alignments and the 'map-pb' preset for the PacBio and control alignments.

We evaluated each data class (ONT, PacBio or control) by examining the counts of supporting reads for each amplicon on the genomic and amplicon reference alignments. To validate intron chains in the genomic alignments, we ensured the read alignment had the same intron chain as the targeted amplicon. We used minimap2 to identify introns in reads, rejecting those with adjacent indels.

For the amplicon reference alignments, we evaluated two metrics: indel similarity and the maximum number of indels. The similarity metric is length-independent, while the absolute difference can distinguish subtle differences such as NAGNAG split sites. We filtered the intron chain results for only those reads with no more than two bases difference in indels.

We gathered these statistics in a table (Supplementary Data 15) and manually classified them as 'supported', 'likely' or 'unsupported'. In cases where there were low read counts or conflicted data (generally less than 50), we examined them using IGV[45].

One confounding issue when using RT−PCR for validation is that the primers crossing introns may result in the ends of some amplicons not aligning across the introns. In such cases, the ends would sometimes align into the intron with a similar sequence while the remainder of the amplicon was soft-clipped. The control alignments of the amplicons are a good indication of this issue, and targets exhibiting the unaligned end regions could be classified as supported based on other evidence.

An interesting case was the ALG6 WTC11 target. Here minimap2 forced a rare but annotated GT-AT U12 intron into a GT-AG intron and genomic deletion, leading to none of the pipelines correctly identifying the isoforms of this gene containing this intron (Supplementary Fig. 51).

To estimate the counts across WTC11 datasets for each experimentally tested transcript, long reads derived from different combinations of library preparation and sequencing methods were first mapped to hg38 using minimap2 (PacBio, minimap2 -ax splice:hq -uf−MD -t 40; ONT, minimap2 -ax splice−MD -t 30; and ONT direct RNA-seq, minimap2 -ax splice -uf -k14−MD -t 30). Primary read alignments were converted from BAM to GTF format to extract the UICs for each read. Then, read counts matching the UIC of a given transcript were summed across all WTC11 datasets. The number of WTC11 datasets where at least one read supported the corresponding UIC was computed, with R2C2 samples with and without size selection being treated independently.

**Targeted PCR validation of isoforms in manatee.** As the manatee transcriptome is not annotated, we employed ab initio gene finding and transcript annotation. Genes were predicted from the manatee genome assembled as part of this LRGASP project and the program GeneMark[46]. BUSCO[28] analysis was used to evaluate the completeness of the transcript assembly and for annotation of the proteins represented from translations of the transcript sequences, thus quantifying the coverage and completeness of the open reading frames.

We targeted a small set of relevant genes for the immune system. To validate our approach, we selected two genes for which a clear single

isoform was consistently identified across pipelines: Secreted phosphoprotein 1 (*SPP1*) and Granzyme K (*GZMK*). Secreted phosphoprotein 1 is involved in immune regulation and tumor progression[47,48] and *GZMK* modulates the proinflammatory immune cytokine response[49].

Next, we selected six genes that were present in the BUSCO database, showed variability in the number of isoforms predicted by each pipeline and had a role in the immune response: interleukin 2 (*IL2*), interleukin 7 (*IL7*), interleukin 1 β (*IL1B*), Pentraxin 3 (*PTX3*), transporter 1, ATP binding cassette subfamily B member (*TAP1*) and TNF α-induced protein 3 (*TNFAIP3*). *IL2* is a cytokine that regulates survival, proliferation and differentiation of T cells[49,50]. *IL7* is important for B and T cell development. *IL1B* is an inflammatory cytokine, an amplifier of an immune response, involved in cell proliferation, differentiation and apoptosis[51]. *PTX3* regulates complement activation of the immune system in the innate immune system and is in the same family of proteins as C-reactive protein[52]. *TNFAIP3* is a negative regulator of inflammation and immunity and a target for drug development[47]. *IL2*, *PTX3* and *TNFAIP3* had one or three isoforms predicted by pipelines using only ONT data. *IL7* had three isoforms predicted by ONT and three by PacBio pipelines. Meanwhile, *TAP1* had four isoforms predicted only by ONT pipelines. *IL1B* had 11 isoforms predicted by ONT data and one by PacBio.

For manual target selection, a similar protocol employed for Challenge 1 targets, using Primers-Juju, was used to select regions of isoforms with UJCs that could be confirmed by the generation of a PCR amplicon product. Whenever possible, the full span of the isoform, up to ~2 kb, was selected. In some regions, multiple primer sets were designed.

Aliquots of the original nine individual manatee RNA samples used in Challenge 3 were stored at −80 °C until the validation stage (Supplementary Data 16). RNA quality was re-verified using BioAnalyzer PicoChip for mRNA (Agilent). Approximately 400 ng of RNA from each manatee sample was pooled to prepare cDNA. cDNA was synthesized using Maxima H minus First Strand cDNA Synthesis kit (Thermo Scientific). Following the manufacturer's instructions, we used a combination of oligonucleotide dT and random hexamer primers for the cDNA synthesis. Controls lacking RT enzyme and controls lacking template were prepared in tandem with test samples.

*Primer selection and RT–PCR*. In the case of the manatee, two or four primer sets were designed for each gene of interest. The process used for primer design was similar to that used for Challenge 1, using a semi-automated approach (see Supplementary Methods and Supplementary Data 17 for the list of primers).

Manatee PCR was performed using KAPA HiFi HotStart Ready Mix (Roche) due to its high sensitivity and low error rate. Approximately 0.01 ng of cDNA was used as a template for individual PCR reactions. PCR protocol was also a touchdown approach with an initial annealing temperature of 70 °C for 15 s, with a reduction of this temperature 1 °C per cycle during 12 cycles. The second amplification phase was carried out for 21 cycles and 2 min of extension. When a PCR fragment larger than 1,500 pb was expected, another PCR was run for that primer set, including 25 cycles and 5 min of extension. PCR products were quantified and sized using an Agilent Bioanalyzer7000 DNA chip (Agilent).

The obtained PCR products were cleaned using a QIAquick PCR purification kit (QIAGEN). All PCR products were pooled as an equimolar pool for PacBio sequencing. We prepared the equimolar solution based on the BioAnalyzer molarity quantification for the band corresponding to the intended PCR product. Additionally, 25 µl of each PCR product that did not show a quantifiable band on the BioAnalyzer was added to the final sample equimolar pool for sequencing.

Analysis of the long-read amplicon reads was conducted in the same manner as for Challenge 1. The manatee RT–PCR sequences were aligned to the pre-submission manatee genome assembly used in LRGASP (GenBank accession no. JARVKP000000000.1). The resulting statistics were gathered (Supplementary Data 18) for manual analysis.

## Reporting summary

Further information on research design is available in the Nature Portfolio Reporting Summary linked to this article.

## Data availability

An overview and documentation about the LRGASP Consortium can be found at https://www.gencodegenes.org/pages/LRGASP/. Biological sequencing data are available from the ENCODE Portal (https://www.encodeproject.org/) and are described in the RNA-seq data matrix (Supplementary Data 1). Experimental data used in GENCODE manual evaluation: ssCAGE WTC11 (Gene Expression Omnibus (GEO): GSE185917); WTC11 QuantSeq (ENCODE: ENCSR322MWL, GEO: GSE219685); H1 QuantSeq (ENCODE: ENCSR813AOB, GEO: GSE219788); and H1-DE QuantSeq (ENCODE: ENCSR198UNH, GEO: GSE219571). Reads generated for experimental validation are available in the NCBI Sequence Read Archive: SRR24680099, manatee whole-blood RT–PCR mixed with human WTC11; GCA_030013775.1, manatee Nanopore genome assembly, BioProject PRJNA939417 (a pre-submission version of the assembly, along with SIRVs, was used in LRGASP at https://cgl.gi.ucsc.edu/data/LRGASP/data/references/lrgasp_manatee_sirv1.fasta.gz); SRR24680098, human WTC11 mixed with manatee whole-blood RT–PCR; and SRR23881262, LRGASP WTC11 experimental validation RT–PCR/ONT. Other data provided to participants, participant submissions, evaluation results and data for generating the paper figures are available from the LRGASP project at https://cgl.gi.ucsc.edu/data/LRGASP/. A UCSC Browser hub with the consolidated models and other data is also available here. LRGASP reference genomes and annotations: https://cgl.gi.ucsc.edu/data/LRGASP/data/references/. LRGASP simulation data: https://cgl.gi.ucsc.edu/data/LRGASP/data/simulation/. Participant submissions: https://cgl.gi.ucsc.edu/data/LRGASP/submissions/. Evaluation results for all challenges: https://cgl.gi.ucsc.edu/data/LRGASP/results/. Spearman correlations of TPMs for each Challenge 2 pipeline: https://cgl.gi.ucsc.edu/data/LRGASP/paper/Spearman_correlation_of_TPM_values.zip. Non-redundant genome annotations derived from the submitted annotations: https://cgl.gi.ucsc.edu/data/LRGASP/annotations/. UCSC Browser Hub with LRGASP evaluation data for human, mouse and manatee: LRGASP Hub, Hub URL. LRGASP-consolidated models description and BED files: https://cgl.gi.ucsc.edu/data/LRGASP/consolidated-models/LRGASP-consolidated-models.html. Simulation ground truth, including lists of incorrectly duplicated artificial transcripts: human simulation ground truth and mouse simulation ground truth. Data for generating Challenge 1 figures for the paper: https://cgl.gi.ucsc.edu/data/LRGASP/paper/Challenge1_Figures_Data.zip. Data for generating Challenge 2 figures for the paper: https://cgl.gi.ucsc.edu/data/LRGASP/paper/Challenge2_Figures_Data.zip. Data for generating Challenge 3 figures for the paper: https://cgl.gi.ucsc.edu/data/LRGASP/paper/Challenge3_Figures_Data.zip.

## Code availability

LRGASP-specific code is available at the GitHub LRGASP project (https://github.com/LRGASP/). LRGASP submission commands, which include documentation on submission metadata and data files: https://github.com/LRGASP/lrgasp-submissions/. Read simulation pipeline: https://github.com/LRGASP/lrgasp-simulation/. Challenge 1 evaluation code: https://github.com/LRGASP/lrgasp-challenge-1-evaluation/. Challenge 2 evaluation code: https://github.com/LRGASP/lrgasp-challenge-2-evaluation/. Challenge 3 evaluation code: https://github.com/LRGASP/lrgasp-challenge-3-evaluation/. Code to generate Challenge 1 figures for the paper: https://github.com/LRGASP/Challenge1_Figures_Code/. Code to generate Challenge 2 figures for the paper: https://github.com/LRGASP/Challenge2_Figures_Code/. Code to generate Challenge 3 figures for the paper: https://github.com/LRGASP/Challenge3_Figures_Code/. Primers-Juju source code is available at https://github.com/diekhans/PrimerS-JuJu/ and was developed

by The University of California, Santa Cruz and El Centre de Regulació Genòmica. Code used for analysis of long-read RNA-seq data used by submitters is described in the 'Computational pipeline description from submitters' section in the Supplementary Information.

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

## Acknowledgements

We thank Lexogen, ONT and PacBio for helpful discussions. ONT provided partial support for flow cells and reagents. We thank T. Sasaki and D. Gilbert for providing the F121-9 hybrid mouse ES cells and K. M. Parsi for assistance with human H1-hES cells and H1-DE cells. We also thank M. Akeson and M. Jain for providing resources and technical advice for Nanopore sequencing. We thank J. Visser for contributing artwork that gives an overview of the LRGASP Consortium. The project is supported by the following grants: Pew Charitable Trust (A.N.B.), NIGMS R35GM138122 (A.N.B.), NHGRI R21HG011280 (A. Conesa, J.M.-T., A.M.-M., A.P. and L.F.-P.), Spanish Ministry of Science PID2023-152976NB-I00 (A. Conesa and F.J.P.), NIGMS R35GM142647 (G.M.S.), NIGMS R35GM133569 (C.V.), NHGRI U41HG007234 (J. Lagarde, M.D., R.G., S.C.-S., J.E.L., J.M.G., T.H., I. Barnes, A.E.B., J.M.M. and A.F.), NHGRI F31HG010999 (A.D.T.) and UM1 HG009443 (A. Mortazavi and B.W.),

NHGRI R01HG008759 and R01HG011469 (K.F.A., D.W. and H.L.), NHGRI R01HG007182 (I. Birol, K.M.N., S.H. and C.Y.), NHGRI UM1HG009402 (Y.S.), NHMRC Investigator Grant GNT2017257 (M.E.R.), Comunitat Valenciana Grant ACIF/2018/290 (F.J.P.), Chan Zuckerberg Initiative DAF, an advised fund of Silicon Valley Community Foundation (grant no. 2019-002443 to M.E.R.), an institutional fund from the Department of Biomedical Informatics, The Ohio State University (K.F.A., D.W. and H.L.), an institutional fund from the Department of Computational Medicine and Bioinformatics, University of Michigan (K.F.A., D.W. and H.L.), SPBU 73023672 (A.P.), AMED 22kk0305013h9903, 23kk0305024h0001 (H.K.), Wellcome Trust (WT222155/Z/20/Z) and European Molecular Biology Laboratory (A.F.). P.C. acknowledges the contribution of funds from MEXT (Ministry of Education, Culture, Sports, Science and Technology of Japan) to RIKEN. We acknowledge M. T. Walsh (University of Florida) and E. Schiller (Homosassa Springs Park) for providing archive Lorelei blood samples. We acknowledge the support of the Spanish Ministry of Science and Innovation to the EMBL partnership, Centro de Excelencia Severo Ochoa and CERCA Programme/Generalitat de Catalunya and the support of the German Federal Ministry of Education and Research with grant no. 161L0242A (M.L. and R.H.). The content of this paper is solely the responsibility of the authors and does not necessarily represent the official views of the National Institutes of Health. Any use of trade, firm or product names is for descriptive purposes only and does not imply endorsement by the US Government. The funders had no role in study design, data collection, analysis, decision to publish or preparation of the manuscript.

## Author contributions

Biosample collection and preparation was carried out by S.C.-S., B.W., M.D.M., A. Cousineau, X.R., M.E.H., R.M., Y.S. and A. Mortazavi. Library preparation and sequencing was carried out by S.C.-S., B.W., M.S.A., G.B.-G., A.K.B., J. Lagarde, C.E.L., D.A.M.A., N.G.P., R.G., B.J.W., C.V., A. Mortazavi and A.N.B. H.T. and P.C. carried out cDNA library technologies development. Data coordination and curation was carried out by F.J.P., F.R., M.D., B.W., G.B.-G., J. Lagarde, M.H.C., S.G., A.M.-M., C.M., I.A.Y., A. Mortazavi and A. Conesa. Quality control was carried out by F.R., B.W., M.D.M., G.B.-G., J. Lagarde, A. Cousineau, X.R., M.E.H., R.M., Y.S., C.V. and A. Mortazavi. Evaluation of Challenge 1 was carried out by F.J.P., J.E.L., J.M.G.M., S.C.-G., J.M.F.-G., C.H.-F., L.K., T.L., J.M.-T., J.M.M., D.R., E.S., A.F., A. Conesa and A.N.B. Evaluation of Challenge 2 was carried out by D.W., G.B.-G., H.L., B.J.W., K.F.A. and A.N.B. Evaluation of Challenge 3 was carried out by F.J.P., S.C.-G., J.M.F.-G., C.H.-F., T.L., C.M., A.P., D.R., E.S., A. Conesa and A.N.B. Validation was carried out by F.J.P., M.D., S.C.-S., M.D., M.J.M., N.D., L.F.-P., N.G.-R., E.R., B.S.-J., L.S., M.L.S., H.T., P.C., N.D.D., M.E.H., G.M.S., A. Mortazavi, A. Conesa and A.N.B. GENCODE benchmarks were carried out by J.E.L., J.M.G.M., T.H., I. Barnes, A.E.B., J.M.M., M.S., A.F., M.M.-T. and A. Conesa. Challenge and submission logistics were carried out by F.J.P., F.R., M.D., J. Lagarde, A.D.T., A. Mortazavi, A. Conesa and A.N.B. Simulation was carried out by F.J.P., F.R., A.D.P. and A. Conesa. LRGASP Challenge Participant/Submitter was carried out by J. Lagarde, A.D.P., I. Birol, H.B., A.M.B., Y.C., M.R.M.D., C.F., J.G., S.H., R.H., H.K., J. Lee, J.-L.L., M.L., A. Mortazavi, A. Mikheenko, D.M., K.M.N., M.P., M.E.R., A.D.S., A.D.T., Y.W., C.W., B.Y.W., H.U.T. and C.Y. Writing was carried out by F.J.P., D.W., F.R., M.D., S.C.-S., B.W., M.D.M., M.A., A.K.B., J. Lagarde, C.E.L., A.D.P., L.F.-P., M.E.H., C.V., A.F., K.F.A., G.M.S., A. Mortazavi, A. Conesa and A.N.B. with input from all co-authors. M.S.A., G.B.-G., A.K.B., J.M.G.M., T.H., J. Lagarde, C.E.L., H.L., M.J.M., D.A.M.A. and A.D.P. contributed equally to this work. C.V., A.F., K.F.A., G.M.S., A. Mortazavi, A. Conesa and A.N.B. jointly supervised the work. More specifically, quality control and R2C2 sequencing was supervised by C.V. GENCODE benchmarks were supervised by A.F. Challenge 2 results were supervised by K.F.A. Validation was supervised by G.M.S. Obtaining human and mouse samples and PacBio sequencing

was supervised by A. Mortazavi. Obtaining manatee samples and sequencing and Challenges 1 and 3 were supervised by A. Conesa. Submission logistics and ONT cDNA and dRNA sequencing were supervised by A.N.B. A. Mortazavi, A. Conesa and A.N.B. co-led the overall study.

## Competing interests

The design of the project was discussed with ONT, PacBio and Lexogen. ONT provided partial support for flow cells and reagents. H.U.T. and A. Conesa have, in the past, presented at events organized by PacBio and have received reimbursement or support for travel, accommodation and conference fees. H.U.T. has also spoken at local ONT events during the duration of this project and received food. Unrelated to this project, the laboratory of H.U.T. has purchased reagents from Illumina, PacBio and ONT at discounted prices. S.C.-S., A.N.B. and J.G. have received reimbursement for travel, accommodation and conference fees to speak at events organized by ONT. A.N.B. is a consultant for Remix Therapeutics. A. Conesa is the founder of Biobam Bioinformatics. The other authors declare no competing interests.

## Additional information

**Extended data** is available for this paper at https://doi.org/10.1038/s41592-024-02298-3.

**Correspondence and requests for materials** should be addressed to Christopher Vollmers, Adam Frankish, Kin Fai Au, Gloria M. Sheynkman, Ali Mortazavi, Ana Conesa or Angela N. Brooks.

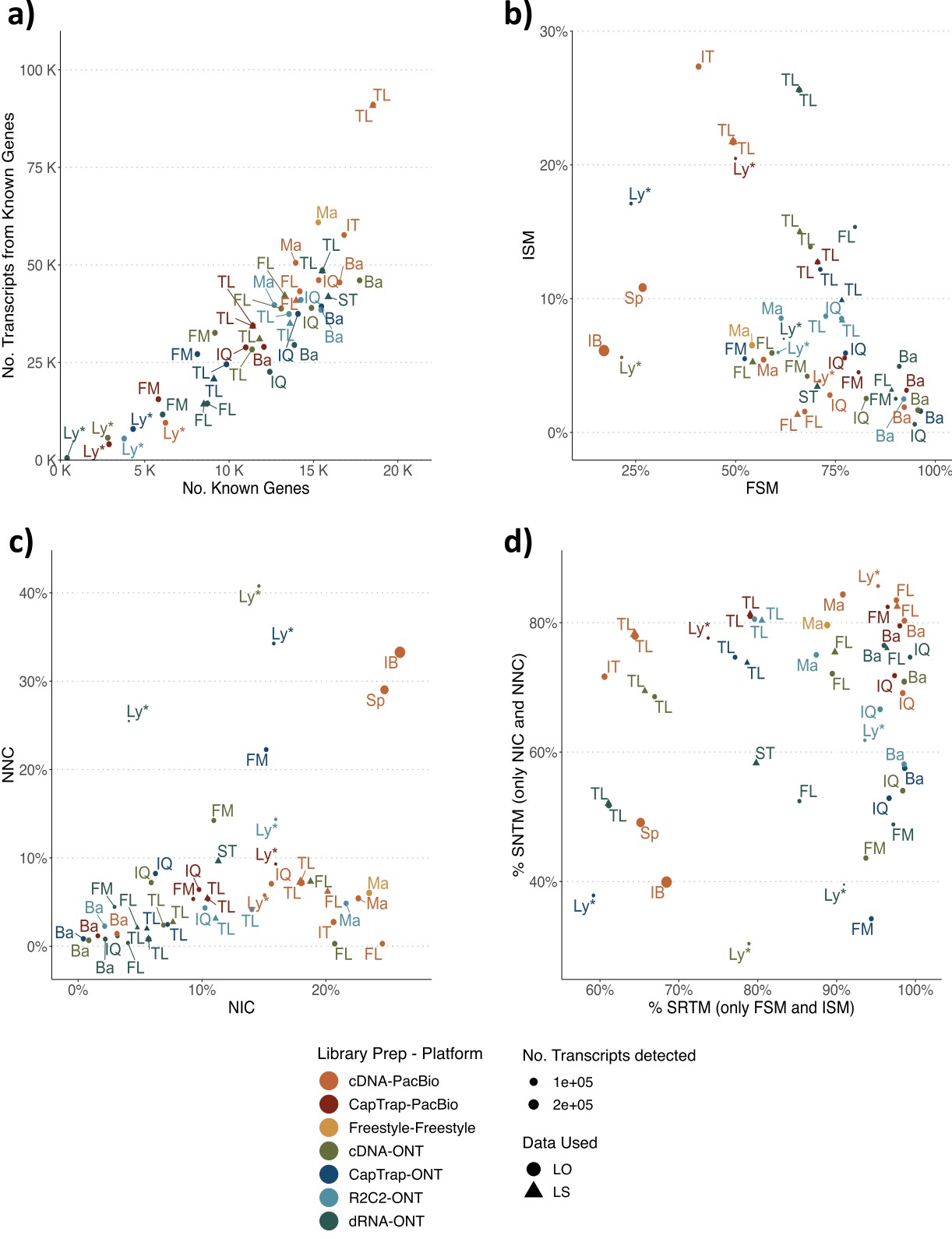

**Extended Data Fig. 1 | SQANTI3 classifications of LRGASP submissions on the WTC11 dataset. a)** Comparison of the number of known genes to transcripts in those genes for the WTC11 dataset. **b)** Percentage of FSM (Full Splice Match) vs ISM (Incomplete Splice Match). **c)** Percentage of NIC (Novel In Catalog) vs NNC

(Novel Not in Catalog). **d)** Percentage of known and novel transcripts with full support at junctions and end positions. Ba: Bambu, FM: FLAMES, FL: FLAIR, IQ: IsoQuant, IT: IsoTools, IB: Iso_IB, Ly: LyRic, Ma: Mandalorion, TL: TALON-LAPA, Sp: Spectra, ST: StringTie2.

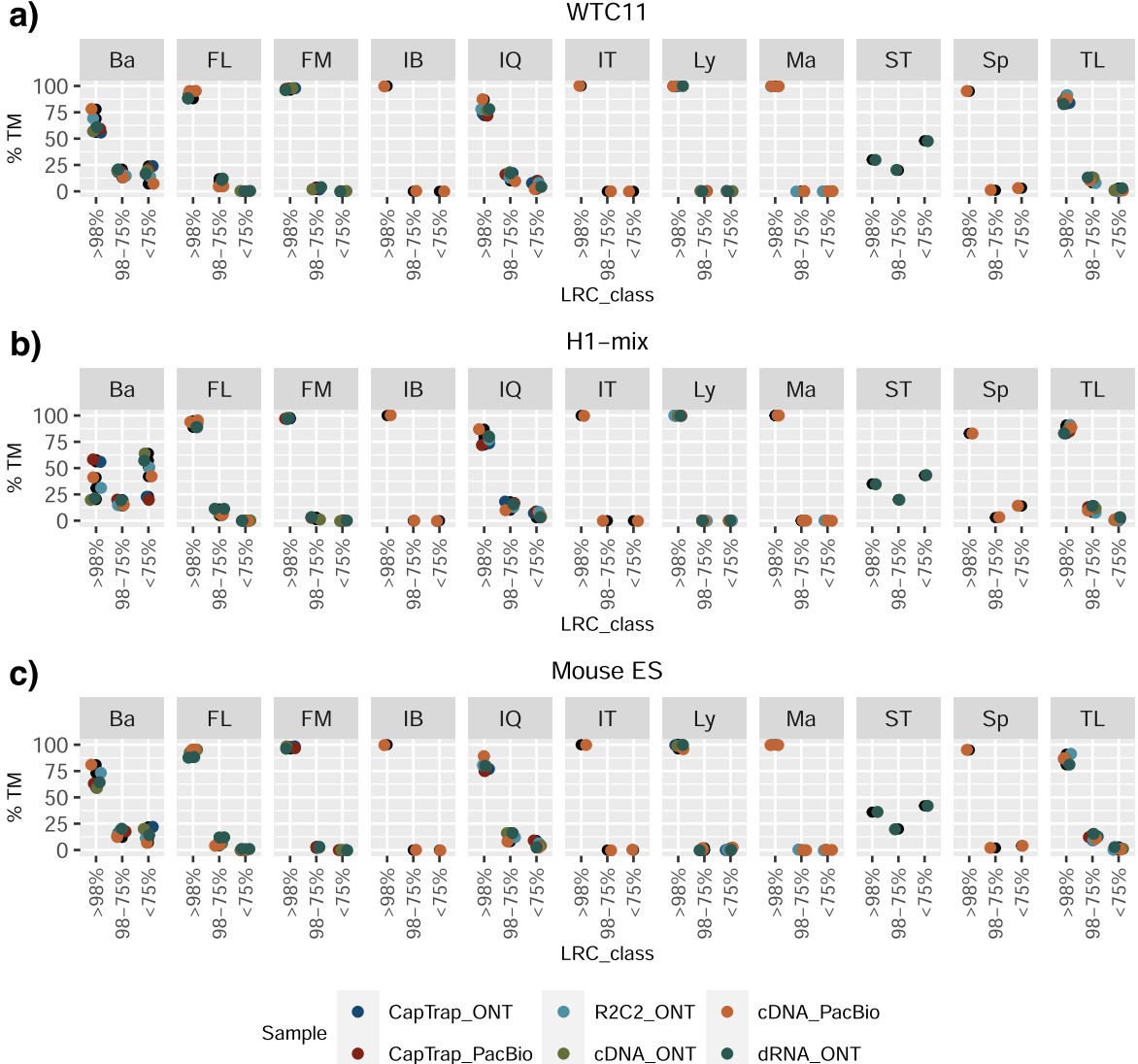

**Extended Data Fig. 2 | Percentage of transcript models with different ranges of sequence coverage by long reads. a)** WTC11. **b)** H1-mix. **c)** Mouse ES. Ba: Bambu, FM: FLAMES, FL: FLAIR, IQ: IsoQuant, IT: IsoTools, IB: Iso_IB, Ly: LyRic, Ma: Mandalorion, TL: TALON-LAPA, Sp: Spectra, ST: StringTie2.

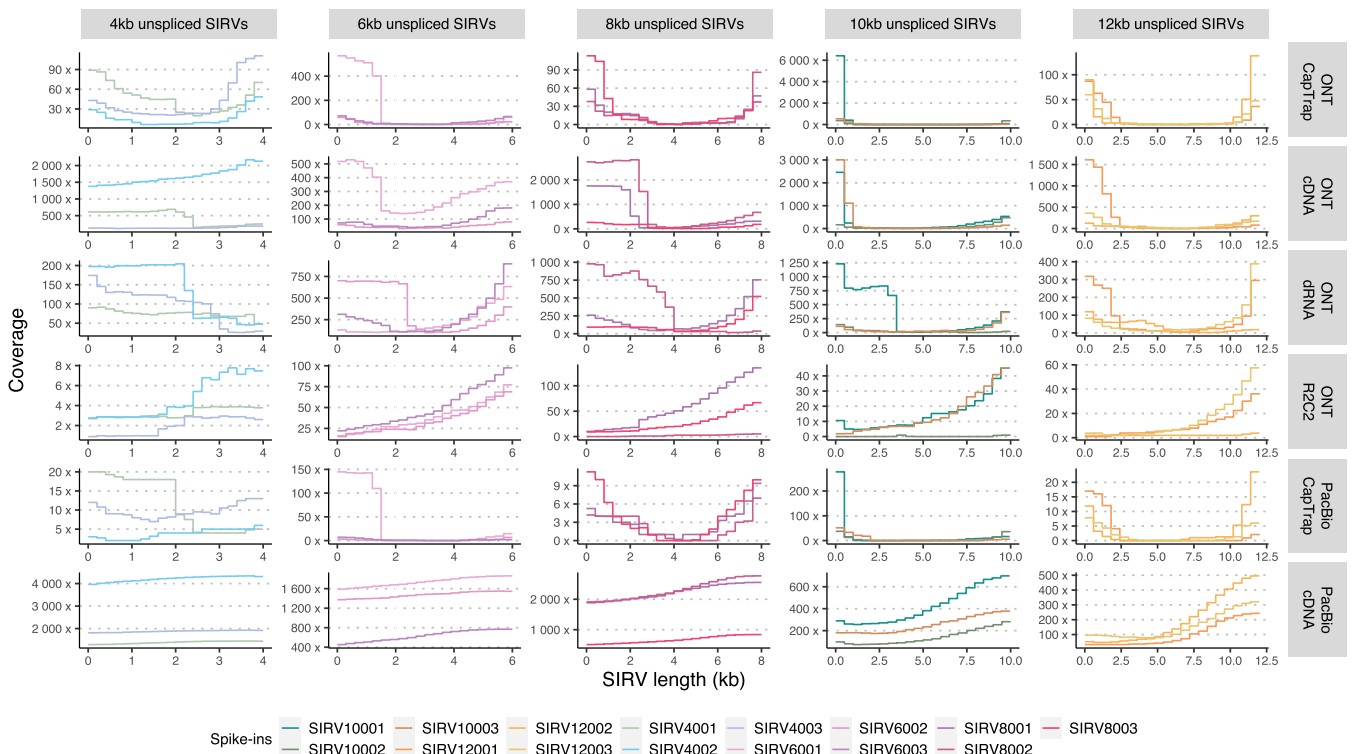

**Extended Data Fig. 3 | Positional coverage of long unspliced SIRV transcript sequences by long reads for each sample type.** The coverage of bases of long unspliced SIRV transcript by long reads for each sample type, grouped by sequence length range.

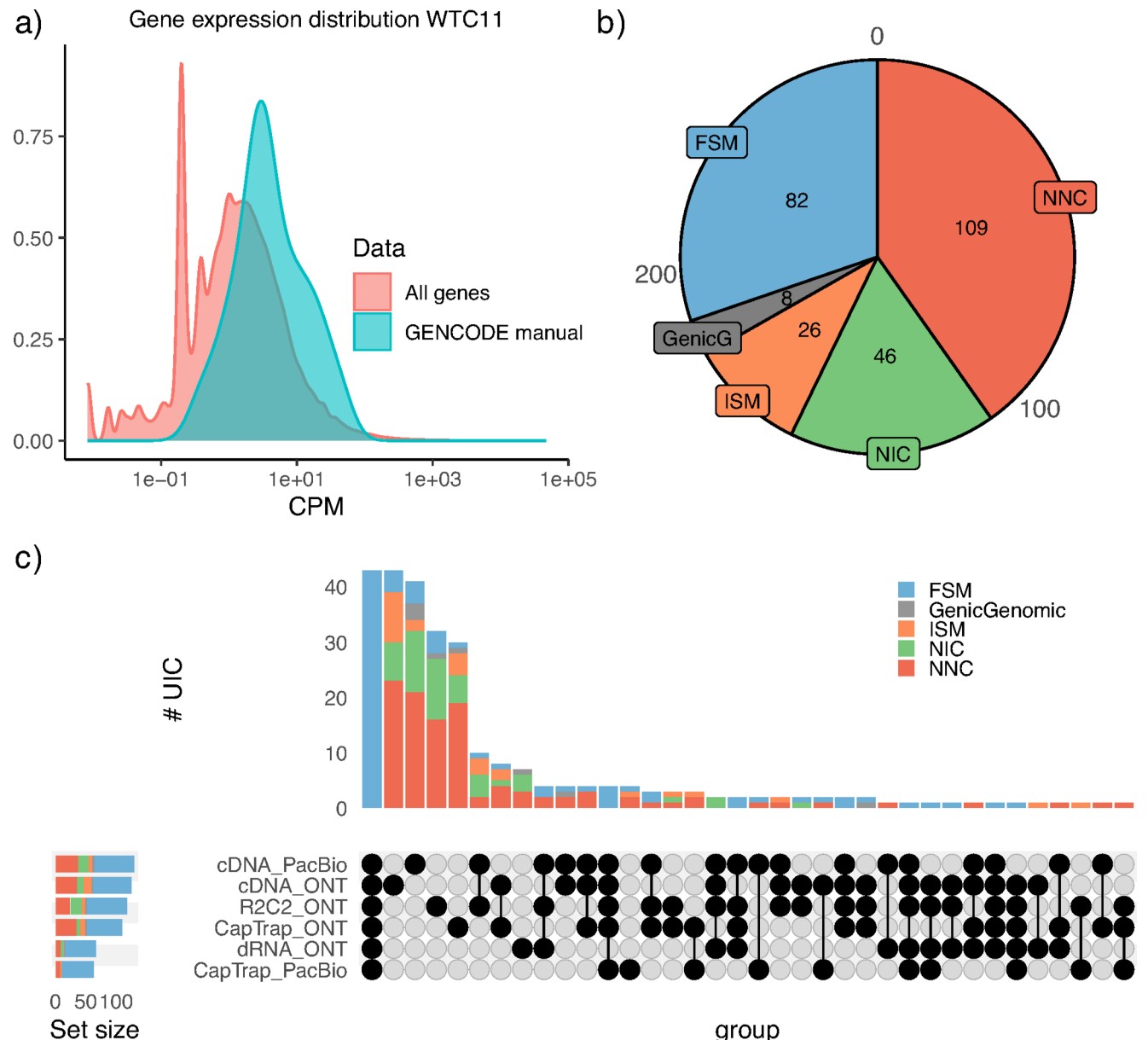

**Extended Data Fig. 4 | Properties of GENCODE manually annotated loci for WTC11 sample. a**) Distribution of gene expression. **b**) Distribution of SQANTI categories. **c**) Intersection of Unique Intron Chains (UIC) among experimental protocols.

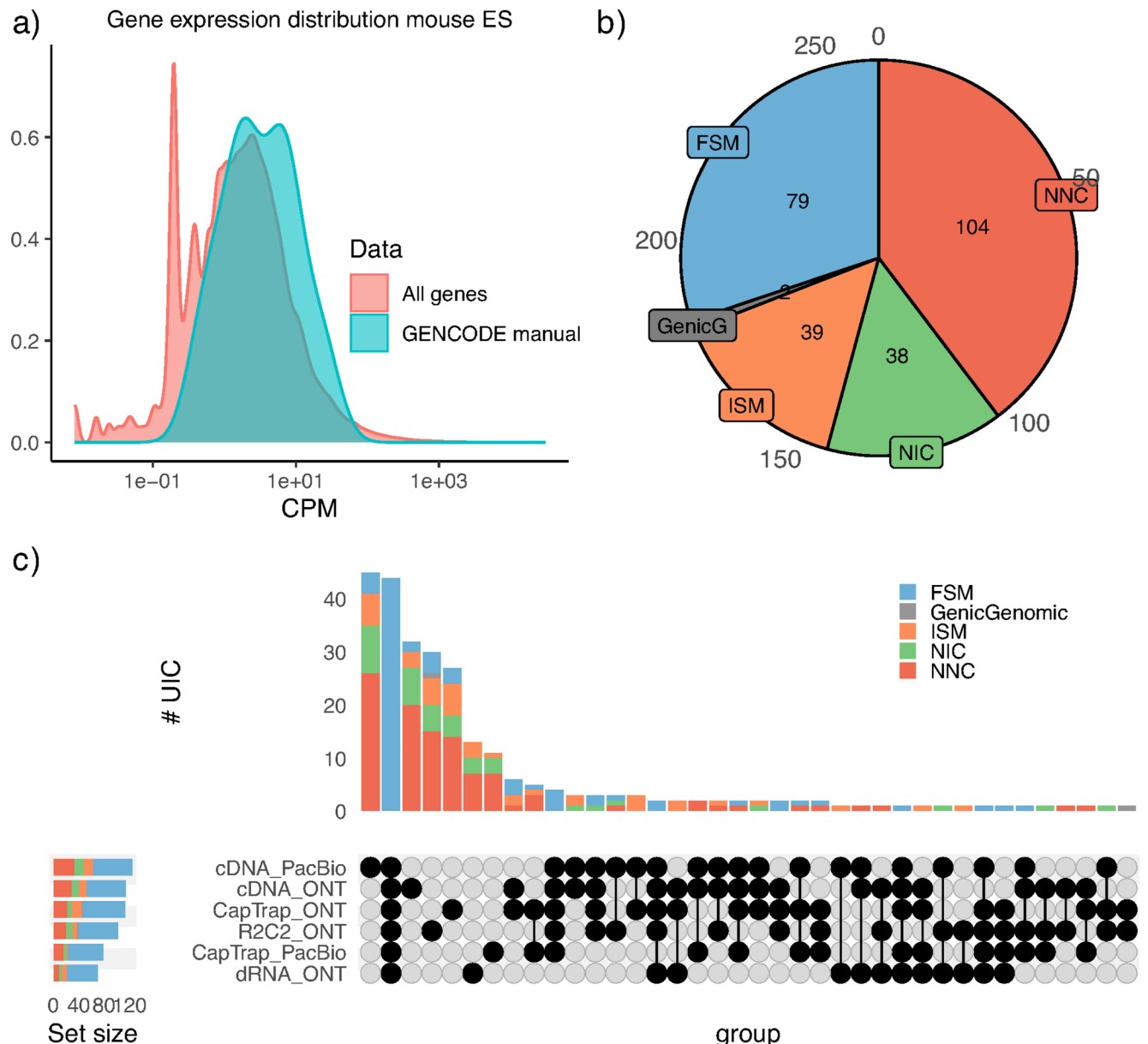

**Extended Data Fig. 5 | Properties of GENCODE manually annotated loci for mouse ES sample. a**) Distribution of gene expression. **b**) Distribution of SQANTI categories. **c**) Intersection of Unique Intron Chains (UIC) among experimental protocols.

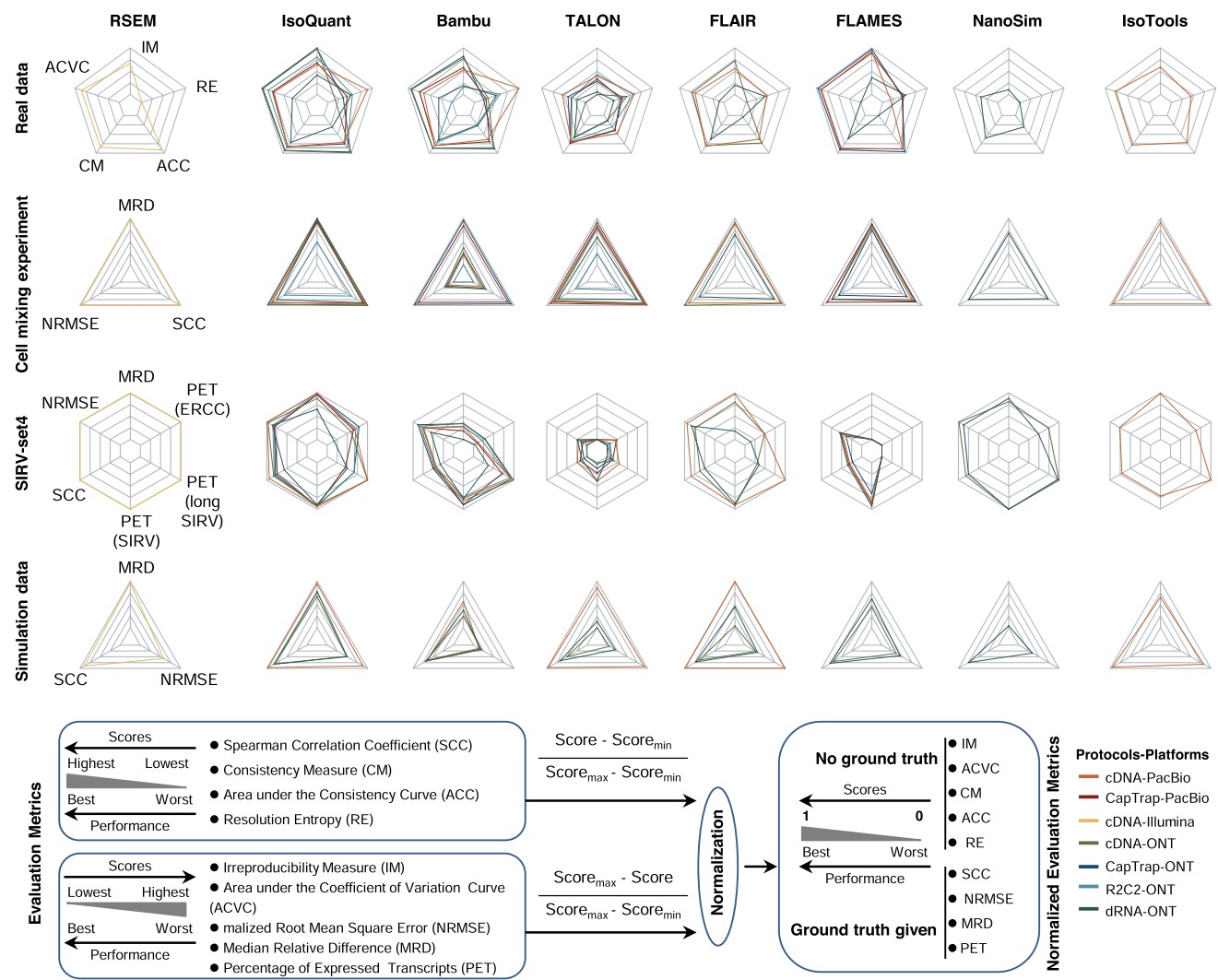

**Extended Data Fig. 6 | Overall evaluation results of eight quantification tools.** Evaluation results from seven protocols-platforms on four data scenarios: real data with multiple replicates, cell mixing experiment, SIRV-set 4 data, and simulation data. To display the evaluation results more effectively, we normalized all metrics to 0–1 range: 0 corresponds to the worst performance, and 1 corresponds to the best performance.

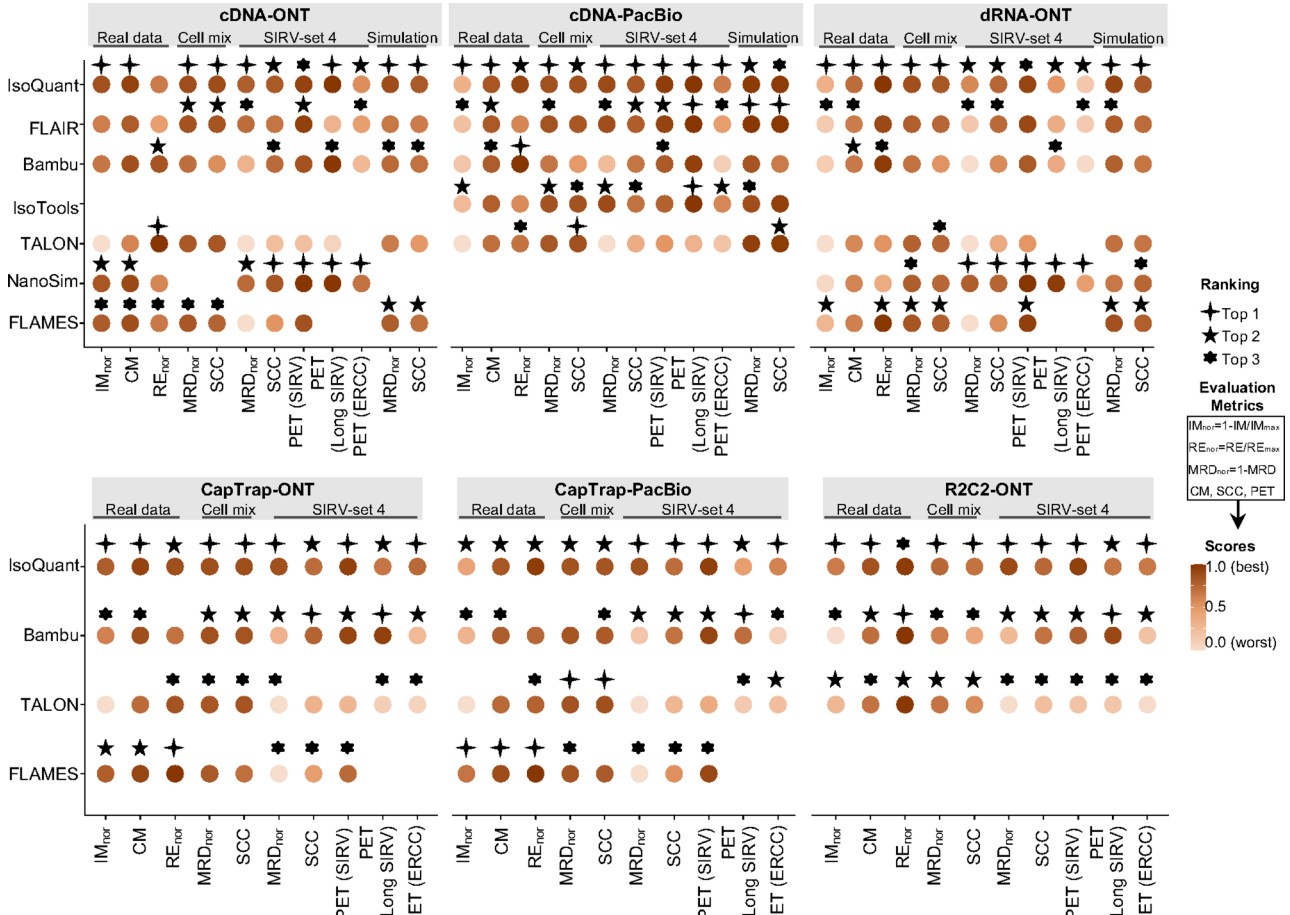

**Extended Data Fig. 7 | Top three performance on quantification tools.**
Quantification results under six different protocols-platforms for each metric.
Here, quantification tools showcase scores under six different protocols-platforms across various evaluation metrics, with the top three performers highlighted for each metric. Blank spaces denote instances where the tool or protocols-platforms did not have participants submitting the corresponding quantitative results.

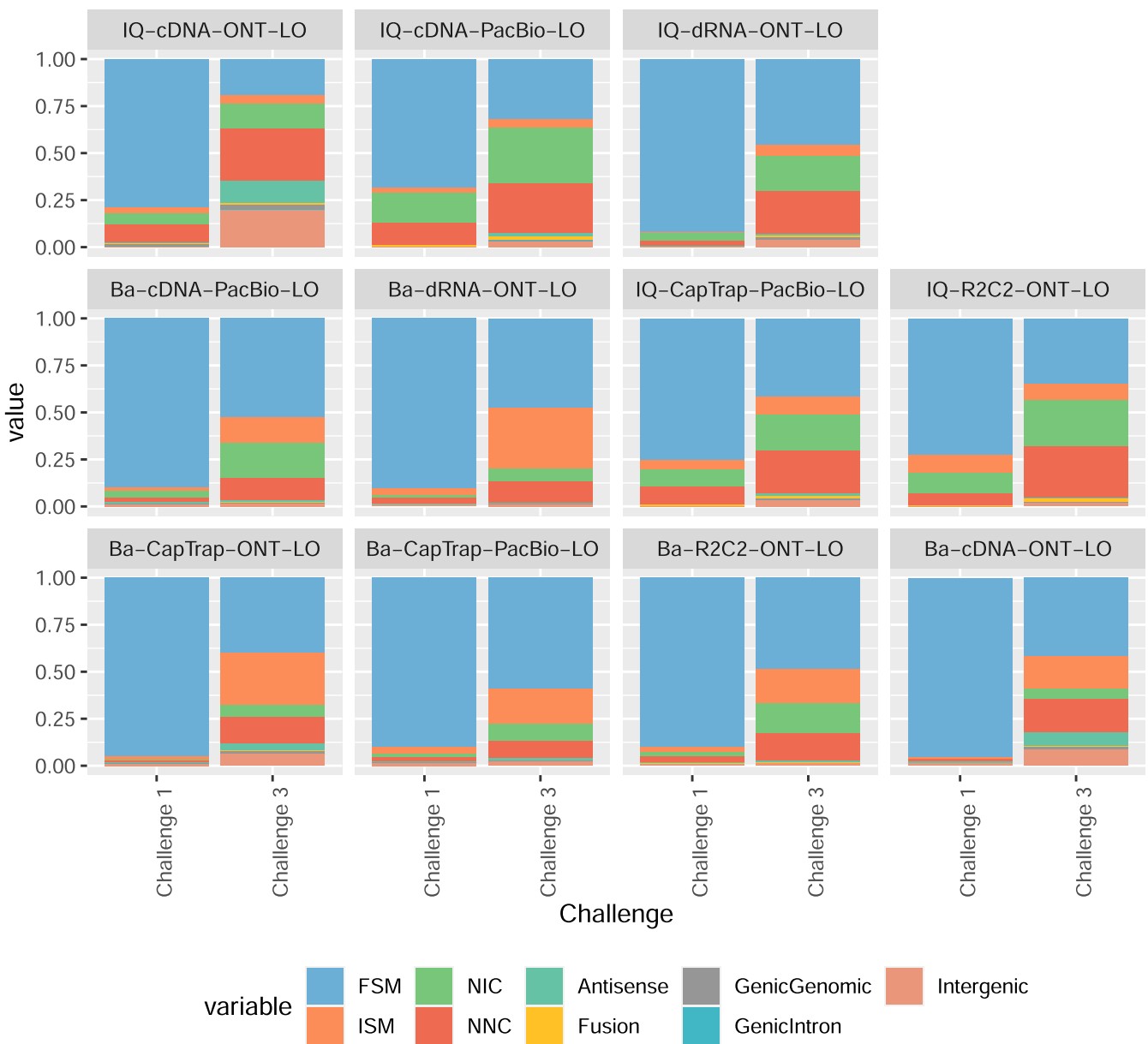

**Extended Data Fig. 8 | SQANTI category classification of transcript models.** Results on transcript models detected by the same tools in Challenge 1 predictions using the reference annotation, and Challenge 3 predictions did not. Ba = Bambu, IQ = StringTie2/IsoQuant.

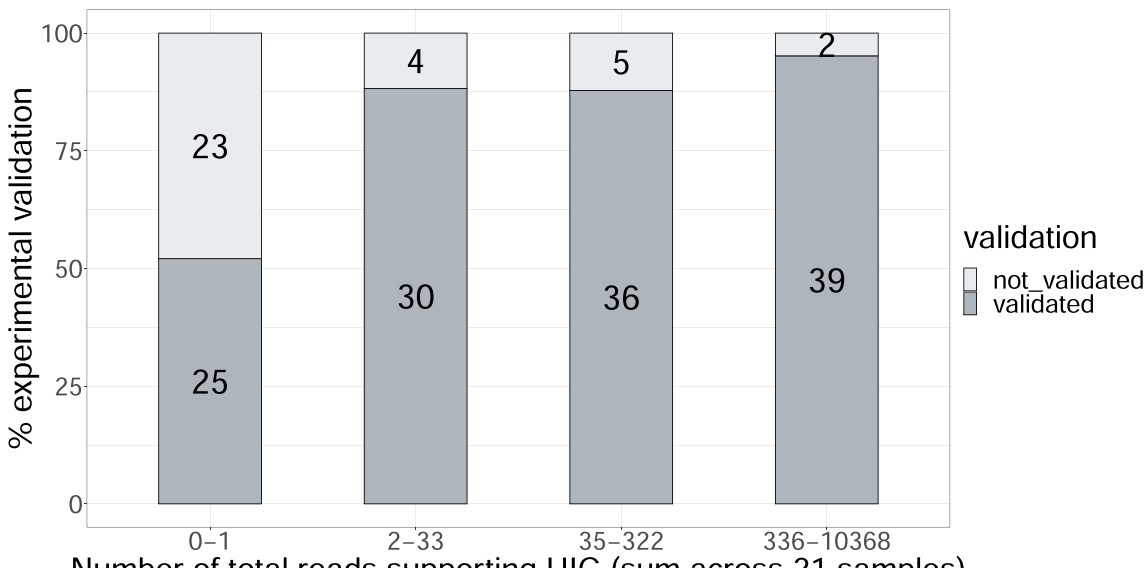

**Extended Data Fig. 9 | Fraction of experimentally validated WTC11 transcripts.** Experimental validation of WTC11 transcripts as a function of the total numbers of long reads that were observed across the 21 library preparations (for example, PacBio cDNA, ONT cDNA, PacBio CapTrap).

# Reporting Summary

## Statistics

For all statistical analyses, confirm that the following items are present in the figure legend, table legend, main text, or Methods section.

| n/a | Confirmed | |
|---|---|---|
| ☐ | ☒ | The exact sample size (*n*) for each experimental group/condition, given as a discrete number and unit of measurement |
| ☐ | ☒ | A statement on whether measurements were taken from distinct samples or whether the same sample was measured repeatedly |
| ☐ | ☒ | The statistical test(s) used AND whether they are one- or two-sided<br>*Only common tests should be described solely by name; describe more complex techniques in the Methods section.* |
| ☒ | ☐ | A description of all covariates tested |
| ☒ | ☐ | A description of any assumptions or corrections, such as tests of normality and adjustment for multiple comparisons |
| ☐ | ☒ | A full description of the statistical parameters including central tendency (e.g. means) or other basic estimates (e.g. regression coefficient) AND variation (e.g. standard deviation) or associated estimates of uncertainty (e.g. confidence intervals) |
| ☒ | ☐ | For null hypothesis testing, the test statistic (e.g. *F*, *t*, *r*) with confidence intervals, effect sizes, degrees of freedom and *P* value noted<br>*Give P values as exact values whenever suitable.* |
| ☒ | ☐ | For Bayesian analysis, information on the choice of priors and Markov chain Monte Carlo settings |
| ☒ | ☐ | For hierarchical and complex designs, identification of the appropriate level for tests and full reporting of outcomes |
| ☒ | ☐ | Estimates of effect sizes (e.g. Cohen's *d*, Pearson's *r*), indicating how they were calculated |

*Our web collection on statistics for biologists contains articles on many of the points above.*

## Software and code

Policy information about availability of computer code

| Data collection | Details on software used for sequencing and basecalling are detailed in the Methods Section and Supplementary data and code availability section. |
|---|---|
| Data analysis | Details on software and code used in analysis are detailed in the "Computational pipeline description from submitters" and "Data and code availability" sections in the Supplementary Information. |

For manuscripts utilizing custom algorithms or software that are central to the research but not yet described in published literature, software must be made available to editors and reviewers. We strongly encourage code deposition in a community repository (e.g. GitHub). See the Nature Portfolio guidelines for submitting code & software for further information.

## Data

Policy information about availability of data

All manuscripts must include a data availability statement. This statement should provide the following information, where applicable:
- Accession codes, unique identifiers, or web links for publicly available datasets
- A description of any restrictions on data availability
- For clinical datasets or third party data, please ensure that the statement adheres to our policy

*Provide your data availability statement here.*

## Research involving human participants, their data, or biological material

Policy information about studies with human participants or human data. See also policy information about sex, gender (identity/presentation), and sexual orientation and race, ethnicity and racism.

| | |
|---|---|
| Reporting on sex and gender | N/A |
| Reporting on race, ethnicity, or other socially relevant groupings | N/A |
| Population characteristics | N/A |
| Recruitment | N/A |
| Ethics oversight | N/A |

Note that full information on the approval of the study protocol must also be provided in the manuscript.

# Field-specific reporting

Please select the one below that is the best fit for your research. If you are not sure, read the appropriate sections before making your selection.

☒ Life sciences          ☐ Behavioural & social sciences          ☐ Ecological, evolutionary & environmental sciences

For a reference copy of the document with all sections, see nature.com/documents/nr-reporting-summary-flat.pdf

# Life sciences study design

All studies must disclose on these points even when the disclosure is negative.

| | |
|---|---|
| Sample size | RNA from each human and mouse sample was obtained and sequenced in biological triplicate. This is a minimum standard in the RNA-seq field. |
| Data exclusions | No data exclusions |
| Replication | All data and code are made publicly available. |
| Randomization | Randomization is not relevant to our study as this did not involve any experiments |
| Blinding | For benchmarking computational tools, multiple benchmarks were blinded or unknown to submitters upon submission. |

# Reporting for specific materials, systems and methods

We require information from authors about some types of materials, experimental systems and methods used in many studies. Here, indicate whether each material, system or method listed is relevant to your study. If you are not sure if a list item applies to your research, read the appropriate section before selecting a response.

### Materials & experimental systems

| n/a | Involved in the study |
|---|---|
| ☒ ☐ | Antibodies |
| ☐ ☒ | Eukaryotic cell lines |
| ☒ ☐ | Palaeontology and archaeology |
| ☒ ☐ | Animals and other organisms |
| ☒ ☐ | Clinical data |
| ☒ ☐ | Dual use research of concern |
| ☒ ☐ | Plants |

### Methods

| n/a | Involved in the study |
|---|---|
| ☒ ☐ | ChIP-seq |
| ☒ ☐ | Flow cytometry |
| ☒ ☐ | MRI-based neuroimaging |

## Eukaryotic cell lines

Policy information about cell lines and Sex and Gender in Research

| | |
|---|---|
| Cell line source(s) | ENCODE cell lines WTC11 (Gladstone Stem Cell Core; cat. # human iPSC line: WTC), H1 (WiCell; cat# WA01), and a Mouse ES |

| Cell line source(s) | F129-1 (S129 Sv/Jae X Cast) cell line (4D Nucleome Consortium, 4DN Biosource ID: 4DNSRMG5APUM) were used. |
|---|---|
| Authentication | Short tandem repeat authentication was not performed for cell lines. |
| Mycoplasma contamination | WTC11 and H1 cell lines were routinely tested for mycoplasma and none as detected. The mouse ES cell line was tested with MycoAlert PLUS (Lonza # LT07-710) and none detected |
| Commonly misidentified lines (See ICLAC register) | None |

## Plants

| Seed stocks | *Report on the source of all seed stocks or other plant material used. If applicable, state the seed stock centre and catalogue number. If plant specimens were collected from the field, describe the collection location, date and sampling procedures.* |
|---|---|
| Novel plant genotypes | *Describe the methods by which all novel plant genotypes were produced. This includes those generated by transgenic approaches, gene editing, chemical/radiation-based mutagenesis and hybridization. For transgenic lines, describe the transformation method, the number of independent lines analyzed and the generation upon which experiments were performed. For gene-edited lines, describe the editor used, the endogenous sequence targeted for editing, the targeting guide RNA sequence (if applicable) and how the editor was applied.* |
| Authentication | *Describe any authentication procedures for each seed stock used or novel genotype generated. Describe any experiments used to assess the effect of a mutation and, where applicable, how potential secondary effects (e.g. second site T-DNA insertions, mosiacism, off-target gene editing) were examined.* |

