## [Peer Review File · Nature Methods]

Peer Review Information

Manuscript Title: Systematic assessment of long-read RNA-seq methods for transcript identification and quantification

Corresponding author name(s): Christopher Vollmers, Adam Frankish, Kin Fai Au, Gloria M. Sheynkman, Ali Mortazavi, Ana Conesa, Angela N. Brooks

Editorial Notes:

Reviewer Comments & Decisions:

Decision Letter, initial version:
--

12th Oct 2021

Dear Angela,

Your Registered Report Stage 1 proposal for an Analysis, "Systematic assessment of long-read RNA-seq methods for transcript identification and quantification", has now been seen by 4 reviewers. As you will see from their comments below, although the reviewers find your work of considerable potential interest, they have raised a number of suggestions and concerns. We are very interested in the possibility of proceeding further with your submission at Nature Methods, but would like to consider your response to these concerns before we reach a final decision on acceptance in principle and Stage 2 submission.

I have discussed the reviewer reports in detail with our chief editor and other members of the Nature Methods team, with a view to 1) identifying key priorities that should be addressed in revision, and 2) overruling reviewer requests that we think fall beyond the scope of the study. Please see the attached file which contains the reviewer reports annotated with our editorial recommendations. We invite you to revise your Stage 1 proposal to address these concerns.

- * include a point-by-point response to the reviewers and to any editorial suggestions
- * please underline/highlight any additions to the text or areas with other significant changes to facilitate review of the revised manuscript

* include a cover letter with the following information

- an anticipated timeline for completing the study (should the study be accepted in principle)
- a statement confirming that, should the study be accepted in principle, you agree to share your raw data and conform to our other open science requirements (detailed below)
- a statement confirming that, following acceptance in principle, you agree to register the Stage 1 proposal in a recognized repository (either publicly or under private embargo), until submission of the Stage 2 manuscript (please note we can assist you in uploading your Stage 1 proposal in our Figshare space)
- a statement confirming that if you later decide to withdraw your manuscript from consideration at Nature Methods, you agree to the journal publishing a brief note about the withdrawn proposal

Please resubmit all the necessary files electronically by using the link below to access your home page:

[REDACTED]

We hope to receive your revised proposal within 4 weeks. If you cannot send it within this time, please let us know. In this event, we will still be happy to reconsider your paper at a later date so long as nothing similar has been accepted for publication at Nature Methods or published elsewhere.

ORCID

Nature Methods is committed to improving transparency in authorship. As part of our efforts in this direction, we are now requesting that all authors identified as 'corresponding author' on published papers create and link their Open Researcher and Contributor Identifier (ORCID) with their account on the Manuscript Tracking System (MTS), prior to acceptance. This applies to primary research papers only. ORCID helps the scientific community achieve unambiguous attribution of all scholarly contributions. You can create and link your ORCID from the home page of the MTS by clicking on 'Modify my Springer Nature account'. For more information please visit please visit www.springernature.com/orcid.

Author feedback:

Registered Report is a new article type being trialed at Nature Methods. We sincerely welcome any author feedback about their experience with this article type and workflow.

Best regards,
Lei

Lei Tang, Ph.D.
Senior Editor
Nature Methods

Reviewers' Comments:

Reviewer #1:

Remarks to the Author:

The authors present a well-defined, systematic approach for the evaluation of pipelines in identifying and quantifying transcripts using long read sequencing data. This community evaluation effort is very timely and invaluable for a broad research community.

Major comments:

First, I think the data set they planned to share with participants is quite limited, which makes it insufficient to capture the diversity and dynamics of transcriptomes. Specifically, there are only three human cell lines (WTC-11, H1, and H1-DE), two of which are close in their developmental stages such that their transcriptomic profiles might be similar. The human body has numerous tissue and cell types—it is well known that different tissues have different genes expressed at substantially varying levels. Also, different isoforms of genes are expressed in different tissues. It may be possible that some pipelines working on one tissue may be less effective in identifying and accurately quantifying transcripts on a different tissue owing to these dynamic transcriptomic profiles. I expect there will be many new methods to be developed for many years to come so the proposed evaluation data may need to be more comprehensive to be truly useful.

Second, I suggest that the authors check the plausibility of their data and metrics by developing their own preliminary pipeline. I don't mean that they have to devote a lot of time to doing so, but they can develop a simple pipeline and test it on the data they plan to provide, which would enable them to verify and possibly improve their suggested evaluation approach. They may not need to make their pipeline available until the close of the competition so that participants may not be biased.

Third, the metrics described in Tables 2 to 5 and statistical measures are well-defined and comprehensive. However, it occurred to me that the authors may want to include some metrics specific to long, error-prone reads (ONT and PacBio) compared to short, accurate reads (Illumina). For example, long reads (especially directly sequenced mRNAs) can match the entirety of a transcript—it may be good to know how many long reads match and support transcripts, with a greater number of reads so matched as an indicator of better pipeline performance. Along the same lines, due to high sequencing error rates, long reads may map to multiple transcripts or the wrong transcript, so the authors may want to include a few metrics on mapping accuracy of long reads using simulation or SIRV-Set reads. These raw numbers on long read alignments may be useful information in addition to the gene expression estimations that the authors suggested. It may be possible that some pipelines may have better alignment ability than others while having worse transcript assembly and quantification ability.

Fourth, the success of this assessment effort would partially depend on securing the funding. I am wondering if the authors could further describe how they plan to secure funding and who would

collect, evaluate, and experimentally validate results from potentially many participants.

Minor comments:

It is not clear whether participants can use both genomes and gene annotations or only genomes for challenges 1 and 2.

As to challenge 2, explicit mention of a transcript expression unit, e.g. TPM, might be clarifying.

The total number of targets, 84 in Table 6, is inconsistent with the number 96 in the text. "We plan to select 96 targets from human WTC-11 cells and 96 targets from the mouse 129/Casteneus cells."

Line 271: there may be a typo in the equation. The denominator has a term, θ , which refers to group number 2, not 1.

Reviewer #2:

Remarks to the Author:

The authors describe the LRGASP Consortium.

Modelled after previous similar consortia GASP, EGASP and RGASP, this project aims to evaluate the long-read sequencing technologies and computational methods to recover transcriptomes and their abundances using a variety of conditions and sequencing methods.

The consortium proposes to evaluate this at three levels.

Level 1 will assess the reconstruction of full-length transcripts in a sample from a well-curated genome, i.e. human and mouse. Level 2 will evaluate the quantification of transcript abundances. Level 3 will assess the reconstruction of full length transcripts from samples without a high-quality reference genome. These three levels of evaluation are set as challenges (challenges 1, 2, and 3) to participants. Participants can submit one more predictions to each of the challenges.

For challenge 1, the pipeline SQANTI3 will be used to obtain performance metrics. These will be computed on the basis of the SIRV-Set 4 spike-ins, simulated data, and a set of manually curated models from GENCODE (undisclosed to participants). The evaluation metrics are based on the values calculated by the SQANTI3 pipeline and are thoroughly described for each of the benchmarking datasets (SIRVs, simulated reads, etc...).

The metrics for the evaluation against SIRVs include

Precision = $RM / SIRV_transcripts$

RM is defined as "FSM transcript with 5' and 3' ends within 50 nts of the transcription start site (TSS)/transcription termination site (TTS) annotation", where FSM is "Transcripts matching a reference transcript at all splice junctions"

The problem with this and possibly other metrics is that there may be that some methods may generate more RMs than SIRVs, which will produce Precisions > 1. This can happen when different predicted transcripts match the same SIRV with equal score/identity and cannot be separated. If this occurs in this or any other cases, what is the plan of the authors in this case? Will they select one at random? Will they select the one predicted with highest abundance?

The behaviour of this and the other parameters would be best clarified if the authors could also present preliminary runs with some sample data to show that the parameters are well defined and useful.

In the table describing the metrics for the comparisons against the simulated data, the authors use Partial True Positive (PTP), which seems to be defined as partial good matches (ISM or FSM_non_RM). Further below, the False Discovery Rate is defined as $(FP + PTP) / (TP + PTP + PTP)$. However, I would not expect the PTP to appear in the numerator. If this is correct, please explain.

The metrics for the comparisons against Gencode (undisclosed release) are described differently (Table 5). In fact, it is not clear what values will be calculated, and what the "X" mean. This table could benefit from additional explanation.

Finally, the metrics for the "curated transcript models" are just enumerated, but without any explanation as before. The authors should describe how the TP, PTP, etc... will be calculated. If these definitions are the same as in the tables above, they should indicate so.

These descriptions would largely benefit if the authors could present some preliminary runs on sample data to show that the long list of metrics are well-behaved and useful.

For Challenge 2, there is a number of different metrics proposed. These are defined with formulae. However, the formulae and their descriptions are not very clear. Some of the symbols are not defined. Moreover, for many symbols, a similar index is used to describe dimensions in completely different spaces, e.g. ijk are used to describe isoforms $i=1,2,\dots,I$, samples $j=1,2$, and tissues $k=1,2,\dots,K$. A more clear notation would use e.g. the i, j, k , letter group for isoforms, the a, b, c , letter group for tissues, and α, β , etc... for conditions.

In any case, all these definitions considerably obscure what is exactly measured for the evaluation and what aspect it covers. A table with somewhat simplified definitions will help to follow what is going to be measured.

As mentioned above, some preliminary results with these proposed metrics for Challenge 2 illustrating that they are sound and useful would provide support for their inclusion in this study.

The evaluation metrics for the Challenge 3 can be split into two levels.

For the mouse data, the prediction of transcripts without the reference genome or transcriptome will be analysed using the metrics from Challenge 1. This implies that the transcript models will be mapped to the genome, but the authors do not say this explicitly. They should also indicate what method they will use to map the mouse transcript models to the mouse genome.

This also raises the potential issue of errors introduced by the mapping tool. Even if the transcript models have 100% identity with the reference genome, it is not clear that every tool will produce the correct alignment. If the identity is $<100\%$, as it is most likely when using long reads, any mapping tool will make mistakes. Using the same mapping tool for all methods will make this evaluation comparable. The authors should clarify further their approach in this case.

The evaluation metrics for the data from manatee seem to be less developed. The authors mention

that 454 data will be used, but do not appear to say how.

The authors mention using BUSCO. They should probably specify with BUSCO set they will use (e.g. mammalian). Also, they should specify what they mean with BUSCO completeness. Analysis with BUSCOs can provide the proportion of predictions with a BUSCO match, and the proportion of BUSCOs found. Both are relevant in this context.

Regarding the experimental validation, the authors propose to evaluate the predicted sequences as well as their abundances. These aspects are less clearly defined, but may require further clarity and possibly a more structured plan. It would be interesting also if the authors could provide preliminary data showing that the proposed approaches are indeed valid for the proposed aims.

For the sequences, there is a clear plan based on targeted sequencing with Sanger / ONT / or PacBio. For quantification is less clear what the best procedure will be.

The authors propose the use of NRSeq. However, this is based on long-read sequencing, which will also include errors, and its not 100% efficient, so the experiments may not produce as many reads as used in the predictions. It would be useful if the authors could show some preliminary data supporting that this strategy is valid for the proposed validation.

As before, the experimental validation for Challenge 3 is less clear. It is more a list of possibilities, but it is not clearly defined what will be tested. Further clarification of what aspects will be covered would be useful.

In summary, the study is well-developed and clearly defined regarding the evaluation of transcripts using a reference genome (Challenge 1), but it is still not clearly defined for Challenge 2 and very poorly developed for Challenge 3. The authors should significantly improve the clarity of the text, tables, descriptions, and formulae to make it more homogenous and easily accessible to other people interested in using and interpreting the proposed metrics. The experimental validation proposed is a very interesting aspect, but it is not well defined in some cases. Providing some preliminary evidence that the proposed metrics are useful and sound for the evaluations to be carried out would significantly improve the manuscript and potentially will simplify some aspects that may turn out to be not as useful as suggested.

Reviewer #3:

Remarks to the Author:

The manuscript presented here outlines the proposed data collection and analysis competition for the LRGASP. The purpose of this is to evaluate long read sequencing technologies for transcriptome sequencing. This is a very timely project as these technologies are beginning to gain momentum for this purpose and we are highly likely to see much broader uptake of long read sequencing in transcriptomics in the near future. With these technological advances it is very important to understand the strengths and weaknesses of the technologies and the analysis methods for this application. This proposal addresses these problems specifically for transcript identification and quantification. It uses established definitions of transcripts in well studied organism and also addresses the use of long reads in non-model organisms which is likely to be a significant application of these technologies.

The plan is that data is generated for different samples and different technologies. This has already been done. Then the analysis aspect is performed through challenges where the community is asked to submit their analysis of the data. It is organised into 3 research challenges and entrants can submit analysis approaches for one part or multiple parts of the challenges. The results will then be compared by the research team. It is clear that the methods for comparing the analysis results have been thought through thoroughly and this is not an easy thing to do. This is a strength of this proposal.

My biggest concern with this manuscript is that it relies on the community to enter the analysis challenges. How do we ensure that the best methods or methods that have already been proposed or ones that are still in development are entered? Do the authors of analysis methods need to submit their own analysis or will some of the team be able to perform already proposed analysis methods on the data to make sure it is entered into the challenge? I feel that there should be a survey of the literature to make sure that proposed methods for these tasks are included in the challenges. If the set of most commonly used methods were not submitted by the community this manuscript would not turn out to be of the highest utility and standard. Can you ensure this is achieved?

Reviewer #4:

Remarks to the Author:

The manuscript describes the design and logistics of LRGASP, a community competition designed to assess the strengths and challenges of the long-read RNA-seq technologies.

While the long-read DNA-seq grew explosively over the last few years, the long-read RNA-seq appears to be lagging, in part owing to the lack of careful quality assessment studies. The LRGASP aims to close this gap, hopefully resulting in much broader adoption of long-read technologies for transcriptomic analyses.

I am impressed with the magnitude and thoughtfulness of the LRGASP design and applaud the Authors for taking on such an important community project. Since the competition is already underway, my comments and suggestions below mostly speak to the analyses and presentation of the results. However, it appears that the Authors are open to running the pipelines themselves: "We expect to re-run analysis pipelines for well-performing submissions to help ensure reproducibility." I believe it will be beneficial to do that not just for the sake of reproducibility but also to perform analyses for the publicly available datasets that are not present in the competition, which I will point out in several of my comments.

1. Most of the biological samples in this study are cell lines. While this simplifies both the data generation, it may also reduce the generalizability of the results to the more complex biological samples (e.g., tissues). It would be illuminating if the Authors ran the pipelines for a few more complex long-read datasets and compared their performance to the cell lines.

2. SQANTI3 is the primary analysis tool for all the results. The original SQANTI publication was in 2018, and it describes a much earlier version. It also describes the SQANTI application to the PacBio but not ONT datasets. Was the new version expanded to include ONT support? The Authors need a very detailed Methods section describing SQANTI3 (or even a companion paper).

3. A large number of challenges, datasets, input data combinations, and metrics may become overwhelming for the readers. Tables are probably not the best way to represent the competition setup. I suggest replacing them with flowcharts and cartoons, illustrating the input data flow, output

results, and metrics.

4. The simulations are an essential part of any benchmarking data. The simulation strategy needs to be described in detail and thoroughly justified. How close are the simulations to the actual data? Which idiosyncrasies of the long-read technologies can (or cannot) be appropriately simulated and benchmarked?

5. One of the significant problems in competitions like RGASP is overfitting the models to maximize the known performance metrics. While some of the “truth” is hidden from the competitors, all the datasets and evaluation scripts are available to participants, which allows for parameter overfitting for a specific dataset and metric. How are the Authors planning to detect and combat the overfitting? If the Authors could run the pipelines themselves on several “hidden” datasets, it would help assess the overfitting and allow for more fair and generalizable comparisons.

6. Most evaluations discussed in the present manuscript deal with comparing analysis pipelines. However, one of the stated (and very important) goals of the LRGASP is to compare the long-read technologies, namely PacBio and ONT, as well comparison of long- and short-read (Illumina) technologies. How are the Authors planning to do it?

7. Experimental validation is extremely interesting and will probably be the most exciting part of this work. It’s a bit disappointing that long-range novel transcript validation via targeted amplification/re-sequencing only involves a small number of target transcripts, which would not permit to confidently assess error rates for detection of novel transcripts.

OPEN SCIENCE REQUIREMENTS

REPORTING SUMMARY AND EDITORIAL POLICY CHECKLISTS

Please note that these forms are dynamic ‘smart pdfs’ and must therefore be downloaded and completed in Adobe Reader. We will then flatten them for ease of use by the reviewers. If you would

like to reference the guidance text as you complete the template, please access these flattened versions at <http://www.nature.com/authors/policies/availability.html>.

DATA AVAILABILITY

All novel DNA and RNA sequencing data, protein sequences, genetic polymorphisms, linked genotype and phenotype data, gene expression data, macromolecular structures, and proteomics data must be deposited in a publicly accessible database, and accession codes and associated hyperlinks must be provided in the "Data Availability" section.

CODE AVAILABILITY

Please include a "Code Availability" subsection in the Online Methods which details how your custom code is made available. Only in rare cases (where code is not central to the main conclusions of the paper) is the statement "available upon request" allowed (and reasons should be specified).

Reviewer Comments	Editor recommendations
Reviewer #1 Remarks to the Author: The authors present a well-defined, systematic approach for the evaluation of pipelines in identifying and quantifying transcripts using long read sequencing data. This community evaluation effort is very timely and invaluable for a broad research community. Major comments: First, I think the data set they planned to share with participants is quite limited, which makes it insufficient to capture the diversity and dynamics of transcriptomes. Specifically, there are only three human cell lines (WTC-11, H1, and H1-DE), two of which are close in their developmental stages such that their transcriptomic profiles might be similar. The human body has numerous tissue and cell types—it is well known that different tissues have different genes expressed at substantially varying levels. Also, different isoforms of genes are expressed in different tissues. It may be possible that some pipelines working on one tissue may be less effective in identifying and accurately quantifying transcripts on a different tissue owing to these dynamic transcriptomic profiles. I expect there will be many new methods to be developed for many years to come so the proposed evaluation data may need to be more comprehensive to be truly useful. Second, I suggest that the authors check the plausibility of their data and metrics by developing their own preliminary pipeline. I don't mean that they have to devote a lot of time to doing so, but they can develop a simple pipeline	We think that the referee raises important concerns regarding the evaluation design, and think most of the comments should be addressed in the revision. Note, three refs have different suggestions on how to improve the metrics evaluating the tools submitted to the Challenges, such as including metrics specific to long, error-prone reads. Please take these concerns into account and try to design a pilot experiment to verify the evaluation pipeline. We understand it may be impossible to meet all of their specific recommendations, though. Note that we do not expect the authors to extend the datasets to additional human tissue samples.

and test it on the data they plan to provide, which would enable them to verify and possibly improve their suggested evaluation approach. They may not need to make their pipeline available until the close of the competition so that participants may not be biased.

Third, the metrics described in Tables 2 to 5 and statistical measures are well-defined and comprehensive. However, it occurred to me that the authors may want to include some metrics specific to long, error-prone reads (ONT and PacBio) compared to short, accurate reads (Illumina). For example, long reads (especially directly sequenced mRNAs) can match the entirety of a transcript—it may be good to know how many long reads match and support transcripts, with a greater number of reads so matched as an indicator of better pipeline performance. Along the same lines, due to high sequencing error rates, long reads may map to multiple transcripts or the wrong transcript, so the authors may want to include a few metrics on mapping accuracy of long reads using simulation or SIRV-Set reads. These raw numbers on long read alignments may be useful information in addition to the gene expression estimations that the authors suggested. It may be possible that some pipelines may have better alignment ability than others while having worse transcript assembly and quantification ability.

Fourth, the success of this assessment effort would partially depend on securing the funding. I am wondering if the authors could further describe how they plan to secure funding and who would collect, evaluate, and experimentally validate results from potentially many participants.

Minor comments:

It is not clear whether participants can use both genomes and gene annotations or only genomes for challenges 1 and 2. As to challenge 2, explicit mention of a transcript expression unit, e.g. TPM, might be clarifying. The total number of targets, 84 in Table 6, is inconsistent with the number 96 in the text. “We plan to select 96 targets from human WTC-11 cells and 96 targets from the mouse 129/Casteneus cells.” Line 271: there may be a typo in the equation. The denominator has a term, θ, which refers to group number 2, not 1	
Reviewer #2 Remarks to the Author: The authors describe the LRGASP Consortium. Modelled after previous similar consortia GASP, EGASP and RGASP, this project aims to evaluate the long-read sequencing technologies and computational methods to recover transcriptomes and their abundances using a variety of conditions and sequencing methods. The consortium proposes to evaluate this at three levels. Level 1 will assess the reconstruction of full-length transcripts in a sample from a well-curated genome, i.e. human and mouse. Level 2 will evaluate the quantification of transcript abundances. Level 3 will assess the reconstruction of full length transcripts from samples without a high-quality reference genome. These three levels of evaluation are set as challenges (challenges 1, 2, and 3) to participants. Participants can submit one more predictions to each of the challenges.	We think the concerns about Challenge 2 and Challenge 3 should be addressed and agree with the referee that preliminary runs are needed to show that the parameters are appropriately defined. As mentioned before, the authors should address the concerns about evaluation metrics. The authors should provide a more detailed plan for the experimental validation.

For challenge 1, the pipeline SQANTI3 will be used to obtain performance metrics. These will be computed on the basis of the SIRV-Set 4 spike-ins, simulated data, and a set of manually curated models from GENCODE (undisclosed to participants). The evaluation metrics are based on the values calculated by the SQANTI3 pipeline and are thoroughly described for each of the benchmarking datasets (SIRVs, simulated reads, etc...).

The metrics for the evaluation against SIRVs include

$\text{Precision} = \text{RM} / \text{SIRV_transcripts}$

RM is defined as “FSM transcript with 5’ and 3’ ends within 50 nts of the transcription start site (TSS)/transcription termination site (TTS) annotation”, where FSM is “Transcripts matching a reference transcript at all splice junctions”

The problem with this and possibly other metrics is that there may be that some methods may generate more RMs than SIRVs, which will produce Precisions > 1. This can happen when different predicted transcripts match the same SIRV with equal score/identity and cannot be separated. If this occurs in this or any other cases, what is the plan of the authors in this case? Will they select one at random? Will they select the one predicted with highest abundance?

The behaviour of this and the other parameters would be best clarified if the authors could also present preliminary runs with some sample data to show that the parameters are well defined and useful.

In the table describing the metrics for the comparisons against the simulated data, the authors use Partial True Positive (PTP), which

seems to be defined as partial good matches (ISM or FSM_non_RM). Further below, the False Discovery Rate is defined as $(FP + PTP) / (TP + PTP + PTP)$. However, I would not expect the PTP to appear in the numerator. If this is correct, please explain.

The metrics for the comparisons against Gencode (undisclosed release) are described differently (Table 5). In fact, it is not clear what values will be calculated, and what the “X” mean. This table could benefit from additional explanation.

Finally, the metrics for the “curated transcript models” are just enumerated, but without any explanation as before. The authors should describe how the TP, PTP, etc... will be calculated. If these definitions are the same as in the tables above, they should indicate so.

These descriptions would largely benefit if the authors could present some preliminary runs on sample data to show that the long list of metrics are well-behaved and useful.

For Challenge 2, there is a number of different metrics proposed. These are defined with formulae. However, the formulae and their descriptions are not very clear. Some of the symbols are not defined. Moreover, for many symbols, a similar index is used to describe dimensions in completely different spaces, e.g. ijk are used to describe isoforms $i=1,2,\dots,I$, samples $j=1,2$, and tissues $k=1,2,\dots,K$. A more clear notation would use e.g. the i, j, k , letter group for isoforms, the $a, b, c\dots$ letter group for tissues, and α, β , etc... for conditions.

In any case, all these definitions considerably obscure what is exactly measured for the evaluation and what aspect it covers. A table with

somewhat simplified definitions will help to follow what is going to be measured.

As mentioned above, some preliminary results with these proposed metrics for Challenge 2 illustrating that they are sound are useful would provide support for their inclusion in this study.

The evaluation metrics for the Challenge 3 can be split into two levels.

For the mouse data, the prediction of transcripts without the reference genome or transcriptome will be analysed using the metrics from Challenge 1. This implies that the transcript models will be mapped to the genome, but the authors do not say this explicitly. They should also indicate what method they will use to map the mouse transcript models to the mouse genome.

This also raises the potential issue of errors introduced by the mapping tool. Even if the transcript models have 100% identity with the reference genome, it is not clear that every tool will produce the correct alignment. If the identity is <100%, as it is most likely when using long reads, any mapping tool will make mistakes. Using the same mapping tool for all methods will make this evaluation comparable. The authors should clarify further their approach in this case.

The evaluation metrics for the data from manatee seem to be less developed. The authors mention that 454 data will be used, but do not appear to say how.

The authors mention using BUSCO. They should probably specify with BUSCO set they will use (e.g. mammalian). Also, they should specify what they mean with BUSCO completeness. Analysis with BUSCOs can provide the proportion of predictions with a BUSCO match, and the proportion of BUSCOs found. Both are relevant in this context.

Regarding the experimental validation, the authors propose to evaluate the predicted sequences as well as their abundances. These aspects are less clearly defined, but may require further clarity and possibly a more structured plan. It would be interesting also if the authors could provide preliminary data showing that the proposed approaches are indeed valid for the proposed aims.

For the sequences, there is a clear plan based on targeted sequencing with Sanger / ONT / or PacBio. For quantification is less clear what the best procedure will be.

The authors propose the use of NRSeq. However, this is based on long-read sequencing, which will also include errors, and its not 100% efficient, so the experiments may not produce as many reads as used in the predictions. It would be useful if the authors could show some preliminary data supporting that this strategy is valid for the proposed validation.

As before, the experimental validation for Challenge 3 is less clear. It is more a list of possibilities, but it is not clearly defined what will be tested. Further clarification of what aspects will be covered would be useful.

In summary, the study is well-developed and clearly defined regarding the evaluation of transcripts using a reference genome (Challenge 1), but it is still not clearly defined for Challenge 2 and very poorly developed for Challenge 3. The authors should significantly improve the clarity of the text, tables, descriptions, and formulae to make it more homogenous and easily accessible to other people interested in using and interpreting the proposed metrics. The experimental validation proposed is a very

interesting aspect, but it is not well defined in some cases. Providing some preliminary evidence that the proposed metrics are useful and sound for the evaluations to be carried out would significantly improve the manuscript and potentially will simplify some aspects that may turn out to be not as useful as suggested.	
Reviewer #3 Remarks to the Author: The manuscript presented here outlines the proposed data collection and analysis competition for the LRGASP. The purpose of this is to evaluate long read sequencing technologies for transcriptome sequencing. This is a very timely project as these technologies are beginning to gain momentum for this purpose and we are highly likely to see much broader uptake of long read sequencing in transcriptomics in the near future. With these technological advances it is very important to understand the strengths and weaknesses of the technologies and the analysis methods for this application. This proposal addresses these problems specifically for transcript identification and quantification. It uses established definitions of transcripts in well studied organism and also addresses the use of long reads in non-model organisms which is likely to be a significant application of these technologies. The plan is that data is generated for different samples and different technologies. This has already been done. Then the analysis aspect is performed through challenges where the community is asked to submit their analysis of the data. It is organised into 3 research challenges and entrants can submit analysis approaches for one part or multiple parts of the challenges. The results will then be compared by the research team. It is clear that the methods for comparing the analysis results have been thought	This reviewer raises a question about how to ensure that the widely used methods or methods under development are entered in the Challenges. We are satisfied with your response to our email, but please do include tables showing which methods were entered into the competition as part of your revision. Please also describe your plans for running short-read methods in your revision. We expect that you discuss the potential limitation of the challenges as well as mention the popular tools that did not participate the challenges.

through thoroughly and this is not an easy thing to do. This is a strength of this proposal. My biggest concern with this manuscript is that it relies on the community to enter the analysis challenges. How do we ensure that the best methods or methods that have already been proposed or ones that are still in development are entered? Do the authors of analysis methods need to submit their own analysis or will some of the team be able to perform already proposed analysis methods on the data to make sure it is entered into the challenge? I feel that there should be a survey of the literature to make sure that proposed methods for these task are included in the challenges. If the set of most commonly used methods were not submitted by the community this manuscript would not turn out to be of the highest utility and standard. Can you ensure this is achieved?	
Reviewer #4: Remarks to the Author: The manuscript describes the design and logistics of LRGASP, a community competition designed to assess the strengths and challenges of the long-read RNA-seq technologies. While the long-read DNA-seq grew explosively over the last few years, the long-read RNA-seq appears to be lagging, in part owing to the lack of careful quality assessment studies. The LRGASP aims to close this gap, hopefully resulting in much broader adoption of long-read technologies for transcriptomic analyses. I am impressed with the magnitude and thoughtfulness of the LRGASP design and applaud the Authors for taking on such an important community project. Since the competition is already underway, my comments and suggestions below mostly speak to the analyses and presentation of the results. However, it appears that the Authors are open to running the	We expect the authors to address the technical concerns and think addressing these concerns will strengthen the paper. There is one suggestion we find out of scope: we do not expect the authors to include additional complex biological samples. Please let us know if it is a challenge to increase the number of target transcripts for experimental validation.

pipelines themselves: “We expect to re-run analysis pipelines for well-performing submissions to help ensure reproducibility.” I believe it will be beneficial to do that not just for the sake of reproducibility but also to perform analyses for the publicly available datasets that are not present in the competition, which I will point out in several of my comments.

1. Most of the biological samples in this study are cell lines. While this simplifies both the data generation, it may also reduce the generalizability of the results to the more complex biological samples (e.g., tissues). It would be illuminating if the Authors ran the pipelines for a few more complex long-read datasets and compared their performance to the cell lines.

2. SQANTI3 is the primary analysis tool for all the results. The original SQANTI publication was in 2018, and it describes a much earlier version. It also describes the SQANTI application to the PacBio but not ONT datasets. Was the new version expanded to include ONT support? The Authors need a very detailed Methods section describing SQANTI3 (or even a companion paper).

3. A large number of challenges, datasets, input data combinations, and metrics may become overwhelming for the readers. Tables are probably not the best way to represent the competition setup. I suggest replacing them with flowcharts and cartoons, illustrating the input data flow, output results, and metrics.

4. The simulations are an essential part of any benchmarking data. The simulation strategy needs to be described in detail and thoroughly justified. How close are the simulations to the actual data? Which idiosyncrasies of the long-

read technologies can (or cannot) be appropriately simulated and benchmarked?

5. One of the significant problems in competitions like RGASP is overfitting the models to maximize the known performance metrics. While some of the “truth” is hidden from the competitors, all the datasets and evaluation scripts are available to participants, which allows for parameter overfitting for a specific dataset and metric. How are the Authors planning to detect and combat the overfitting? If the Authors could run the pipelines themselves on several “hidden” datasets, it would help assess the overfitting and allow for more fair and generalizable comparisons.

6. Most evaluations discussed in the present manuscript deal with comparing analysis pipelines. However, one of the stated (and very important) goals of the LRGASP is to compare the long-read technologies, namely PacBio and ONT, as well comparison of long- and short-read (Illumina) technologies. How are the Authors planning to do it?

7. Experimental validation is extremely interesting and will probably be the most exciting part of this work. It’s a bit disappointing that long-range novel transcript validation via targeted amplification/re-sequencing only involves a small number of target transcripts, which would not permit to confidently assessment error rates for detection of novel transcripts.

Author Rebuttal to Initial comments

We thank the reviewers for the overall positive feedback and helpful suggestions. In particular, we have included new Supplemental Figures to diagram the overall study and have provided a demonstration of our evaluation metrics based on transcript analysis of previously published GM12878 long-read transcriptome data with a variety of tools. This was meant to demonstrate the utility of our evaluation metrics, but not to evaluate the tools or sequencing platforms, which will be included in the final version of our study. We also include many points of clarification.

Reviewers' Comments:

Reviewer #1:

Remarks to the Author:

The authors present a well-defined, systematic approach for the evaluation of pipelines in identifying and quantifying transcripts using long read sequencing data. This community evaluation effort is very timely and invaluable for a broad research community.

We thank the reviewer for this positive feedback.

Major comments:

Q.1.1. First, I think the data set they planned to share with participants is quite limited, which makes it insufficient to capture the diversity and dynamics of transcriptomes. Specifically, there are only three human cell lines (WTC-11, H1, and H1-DE), two of which are close in their developmental stages such that their transcriptomic profiles might be similar. The human body has numerous tissue and cell types—it is well known that different tissues have different genes expressed at substantially varying levels. Also, different isoforms of genes are expressed in different tissues. It may be possible that some pipelines working on one tissue may be less effective in identifying and accurately quantifying transcripts on a different tissue owing to these dynamic transcriptomic profiles. I expect there will be many new methods to be developed for many years to come so the proposed evaluation data may need to be more comprehensive to be truly useful.

R.1.1. Thank you for this important remark. Challenge 1 uses three different cell lines, WTC11, H1-mix and a mouse ES sample; Challenge 2 uses H1-mix, H1 and H1-DE, while Challenge 3 uses the mouse data and a manatee blood sample. We include three different sequencing platforms, four library preparation methods and allow for the utilization of long only, long and short reads, or a combination with additional data (free-style). For Challenge 1 alone this results into 16 different combinations of experimental factors that will be evaluated. Importantly, the utilization of cell-lines allowed us to have enough uniform biological sample material to carry out this great diversity of experimental protocols. We understand that the sample representation is limited, however, we do believe that our experimental design will provide a comprehensive assessment of the contribution of both experimental and computational methods to transcriptome analysis using long reads. The fact that several cell lines must be evaluated for each analysis pipeline gives us a first insight on the robustness of the proposed methods for transcriptome characterization. We believe this is a first step towards benchmarking lrRNA-seq methods that will reveal the challenges the technology faces to accurately describe transcriptome composition and abundance. Our results on Pilot Data suggest a great deal of diversity among analysis pipelines, even with more homogeneous cell lines (see new figures 4, 5 and 6). We believe that an exhaustive analysis, as delivered by LRGASP, of “simple” cell types is a necessary start in the benchmarking lrRNA-seq methods that will provide insightful results to a follow up with more complex sample types. Importantly, the LRGASP evaluation metrics and comparison scripts are made available to the community. Moreover, our benchmarking and experimental validation approaches will be presented. Therefore other studies will be able to build on LRGASP results and resources for further benchmarking of lrRNA-seq technology on new data and methods.

Q.1.2. Second, I suggest that the authors check the plausibility of their data and metrics by developing their own preliminary pipeline. I don't mean that they have to devote a lot of time to doing so, but they can develop a simple pipeline and test it on the data they plan to provide, which would enable them to verify and possibly improve their suggested evaluation approach. They may not need to make their pipeline available until the close of the competition so that participants may not be biased.

R.1.2. Following this and other reviewers' suggestions, we now provide example evaluations for all three Challenges and include it in a new Pilot Data section of this Registered Report. For Challenges 1 and 2, we used published ONT dRNA and PacBio cDNA data from GM12878 and for Challenge 3 we used subsamples of the actual manatee data. We used several published computational pipelines and no attempts for optimizations were made. The Consortium actually performed these mock runs before launching the competition to verify that LRGASP metrics were informative and would capture differences between lrRNA-seq methods. We have now re-run these analysis to include them in the Registered Report as a Pilot Data section. We believe that the Pilot Data section demonstrates the

usefulness of the LRGASP metrics to comparatively describe the output of lrrna-seq pipelines and, more importantly, reveal great differences among them, highlighting the significance of the LRGASP challenge.

Q.1.3. Third, the metrics described in Tables 2 to 5 and statistical measures are well-defined and comprehensive. However, it occurred to me that the authors may want to include some metrics specific to long, error-prone reads (ONT and PacBio) compared to short, accurate reads (Illumina). For example, long reads (especially directly sequenced mRNAs) can match the entirety of a transcript—it may be good to know how many long reads match and support transcripts, with a greater number of reads so matched as an indicator of better pipeline performance.

R.1.3. This is an interesting point that we have already considered in our challenge, but did not realize was not clear from our registered report document. For each pipeline we asked submitters to include a file that indicates the reads that support each of the proposed transcript models. This information is used in several ways. First, as the reviewer suggested, we incorporate the Long Read Coverage (LRC) metric, that indicates the % of the transcript model sequence length supported by at least one long read. Our preliminary analysis of Pilot Data suggests that some pipelines may “complete” transcript models based on reference annotation information rather than using the actual long reads data (Figure below).

Figure. Relationship between transcript model sequence and supporting reads. The graph shows the distance between the position of the reported Transcription Start Site (TSS, a) and Transcription Termination Site (TSS, b) of the inferred transcript models (TM) and the closest mapping position of the corresponding supporting reads reported by the submitter. A distance equal to 0 indicates the start or end of the TM matches the start or end position of at least one supporting reads. Deviations from 0 indicate that the TM TSS and TTS is not well supported by the long reads.

Additionally, apart from looking at the number of reads supporting each transcript model, we will evaluate the consistency on transcript detection across pipelines, platforms and library preparations as a function of the expression value (here roughly defined as the median count-per-million for those pipelines that report the transcript). Our preliminary analysis of Pilot Data suggests that highly expressed transcripts are more frequently detected by different pipelines than those reported by only a few pipelines, but also that differences exist between PacBio and Nanopore.

Q.1.4. Along the same lines, due to high sequencing error rates, long reads may map to multiple transcripts or the wrong transcript, so the authors may want to include a few metrics on mapping accuracy of long reads using simulation or SIRV-Set reads. These raw numbers on long read alignments may be useful information in addition to the gene expression estimations that the authors suggested. It may be possible that some pipelines may have better

alignment ability than others while having worse transcript assembly and quantification ability.

R.1.4. The LRGASP challenge is about transcript identification and quantification. We are, in principle, not directly evaluating the mapping strategy implemented by each pipeline. Any mapping error should be reflected either in the identification of transcripts or in their quantification. Moreover, not every analysis pipeline starts by mapping. In some cases a clustering step is first applied and the resulting transcript models are mapped to the genome.

However, as we are requesting from submitters the list of reads supporting each transcript model, we can evaluate if reads are assigned to multiple transcripts, which is a proxy for multimapping. We have investigated this in our Pilot Data and found that, while some pipelines will assign reads to multiple transcripts, most of them only report one transcript per gene. We will incorporate this metric in the assessment and evaluate its possible relationship with the assessment of quantification accuracy. Thank you for the suggestion.

Additionally, upon reviewing documentation provided by submitters to the LRGASP challenge (submissions closed on October 8, 2021), all submitters used minimap2 as the alignment program; therefore, we expect differences between pipelines will not be derived from the underlying alignment algorithm.

Q.1.5. Fourth, the success of this assessment effort would partially depend on securing the funding. I am wondering if the authors could further describe how they plan to secure funding and who would collect, evaluate, and experimentally validate results from potentially many participants.

R.1.5. Funding has been secured from a variety of sources for the LRGASP effort. Each of the senior PIs (Vollmers, Sheynkman, Frankish, Au, Conesa, Mortazavi, Brooks) provided funding support through available start-up, allowable grants where LRGASP was already within the scope (e.g. ENCODE (PI, Mortazavi)). Moreover, in-kind contributions were done by the University of Florida Interdisciplinary Center for Biotechnology Research (UF ICBR) for library preparation and sequencing of a number of manatee samples. We also received partial support for nanopore sequencing reagents and flow cells from Oxford Nanopore. Computational evaluation of submission is done by members of the Conesa, Fai, GENCODE (Diekhans, Frankish) labs, who have already secured the personnel time for this work. Finally,

experimental validation is funded by ENCODE (providing CAGE and polyA-seq data), Dr. Sheynkman's start-up (human and mouse evaluations) and Dr. Hunter (manatee evaluation) lab.

Minor comments:

Q.1.6. It is not clear whether participants can use both genomes and gene annotations or only genomes for challenges 1 and 2.

R.1.6. Both genome and genome annotation can be used for Challenge 1 and 2. This is specified in the Challenge instructions

Q.1.7. As to challenge 2, explicit mention of a transcript expression unit, e.g. TPM, might be clarifying.

R.1.7. In the revised manuscript, we add a definition in the section "*Challenge 2 Evaluation: Transcript isoform quantification*" to clarify that Transcripts Per Million (TPM) is used as the unit of transcript abundance.

Q.1.8. The total number of targets, 84 in Table 6, is inconsistent with the number 96 in the text. "We plan to select 96 targets from human WTC-11 cells and 96 targets from the mouse 129/Castaneus cells."

R.1.8. The numbers in the text are incorrect, and we have corrected the numbers in the main text to 84.

Q.1.9. Line 271: there may be a typo in the equation. The denominator has a term, θ , which refers to group number 2, not 1.

R.1.9. We corrected this typo in the revised manuscript.

Reviewer #2:

Remarks to the Author:

The authors describe the LRGASP Consortium.

Modelled after previous similar consortia GASP, EGASP and RGASP, this project aims to evaluate the long-read sequencing technologies and computational methods to recover transcriptomes and their abundances using a variety of conditions and sequencing methods.

The consortium proposes to evaluate this at three levels.

Level 1 will assess the reconstruction of full-length transcripts in a sample from a well-curated genome, i.e. human and mouse. Level 2 will evaluate the quantification of transcript abundances. Level 3 will assess the reconstruction of full length transcripts from samples without a high-quality reference genome. These three levels of evaluation are set as challenges (challenges 1, 2, and 3) to participants. Participants can submit one more predictions to each of the challenges.

For challenge 1, the pipeline SQANTI3 will be used to obtain performance metrics. These will be computed on the basis of the SIRV-Set 4 spike-ins, simulated data, and a set of manually curated models from GENCODE (undisclosed to participants). The evaluation metrics are based on the values calculated by the SQANTI3 pipeline and are thoroughly described for each of the benchmarking datasets (SIRVs, simulated reads, etc...).

The metrics for the evaluation against SIRVs include

Precision = $RM / SIRV_transcripts$

RM is defined as “ FSM transcript with 5’ and 3’ ends within 50 nts of

the transcription start site (TSS)/transcription termination site (TTS) annotation”, where FSM is “Transcripts matching a reference transcript at all splice junctions”

Q.2.1. The problem with this and possibly other metrics is that there may be that some methods may generate more RMs than SIRVs, which will produce Precisions > 1. This can happen when different predicted transcripts match the same SIRV with equal score/identity and cannot be separated. If this

occurs in this or any other cases, what is the plan of the authors in this case? Will they select one at random? Will they select the one predicted with highest abundance?

R.2.1. Thank you for the comment. We actually accounted for this in one of our metrics and, here, we will additionally clarify. We introduced the *non-redundant precision* metric to account for this problem, where the numerator is the number of SIRVs identified with at least one RM and the denominator is the number of detected SIRVs. By comparing *precision* with *non-redundant precision*, we have an estimate of this issue. Generally, to account for the possibility of multiple FSMs matching to the same reference transcript we use the redundancy metric, available at SQANTI3. This gives the average number of FSM per transcript model. Pipelines that return only one FSM (or ISM) per transcript models have redundancy = 1. Pipelines that allow multiple FSM per transcript model (i.e. by modelling different 3' or 5' ends) have redundancy levels greater than 1. We actually see that redundancy level is one of the major differences across pipelines. The reviewer can check on these metrics in the Challenge 1 Pilot Data report provided as a supplementary file.

Q.2.2. The behaviour of this and the other parameters would be best clarified if the authors could also present preliminary runs with some sample data to show that the parameters are well defined and useful.

R.2.2. Following this and other reviewers' suggestions we are now providing example results for all three Challenges using Pilot Data. This is described above in R.1.2.

Q.2.3. In the table describing the metrics for the comparisons against the simulated data, the authors use Partial True Positive (PTP), which seems to be defined as partial good matches (ISM or FSM_non_RM). Further below, the False Discovery Rate is defined as $(FP + PTP) / (TP + PTP + PTP)$. However, I would not expect the PTP to appear in the numerator. If this is correct, please explain.

R.2.3. Thank you for pointing this out. It is debatable if a PTP is a good match or not. In the case of SIRVs, we considered this as a partial match of a true transcript. However, ISM or FSM_non_RM could also be considered novel transcripts. In this case a PTP would be a false new transcript. When computing the False Discovery Rate we considered that PTP are false novel discoveries and therefore included in the numerator. However, to account for the interpretation of PTP having a value of truth we have now included the metric False Detection Rate, (FDeR) metric as $FP / (TP + TF + PTP)$, in which PTP is not

included in the numerator. We hope in this way to be able to capture these differences in the partial matching of annotated transcripts.

Q.2.4. The metrics for the comparisons against Gencode (undisclosed release) are described differently (Table 5). In fact, it is not clear what values will be calculated, and what the “X” mean. This table could benefit from additional explanation.

R.2.4. We apologize for the confusion. The description of the metrics used in Table 5 is given in Table 2, where we describe LRGASP metrics, which are used to evaluate different types of data. The X simply refers to that metric will be used for the indicated structural category, as not all metrics make sense for all structural categories. We have added calls in Table 5 to clarify this.

Q.2.5. Finally, the metrics for the “curated transcript models” are just enumerated, but without any explanation as before. The authors should describe how the TP, PTP, etc... will be calculated. If these definitions are the same as in the tables above, they should indicate so.

R.2.5. We apologize for this. Table 6 is now included with the definition of metrics for the curated transcript models.

Q.2.6. These descriptions would largely benefit if the authors could present some preliminary runs on sample data to show that the long list of metrics are well-behaved and useful.

R.2.6. See R.1.2.

Q.2.7. For Challenge 2, there is a number of different metrics proposed. These are defined with formulae. However, the formulae and their descriptions are not very clear. Some of the symbols are not defined. Moreover, for many symbols, a similar index is used to describe dimensions in completely different spaces, e.g. ijk are used to describe isoforms $i=1,2,\dots,l$, samples $j=1,2$, and tissues $k=1,2,\dots,K$

A more clear notation would use e.g. the i, j, k , letter group for isoforms, the a, b, c, \dots letter group for tissues, and α, β , etc... for conditions.

R.2.7. In order to more clearly describe the relevant metrics in Challenge 2, we revised the symbols. Here, letter i ($i=1,2,\dots,I$) is used to represent isoforms, letter g ($g=1,2,\dots,G$) is used to represent different groups (i.e., conditions or tissues), and letter r ($r=1,2,\dots,R$) is used to represent different replicates. In addition, we add a description in the section “*Multiple replicates under two different conditions*” to explain these symbols more clearly.

Q.2.8. In any case, all these definitions considerably obscure what is exactly measured for the evaluation and what aspect it covers. A table with somewhat simplified definitions will help to follow what is going to be measured.

R.2.8. We add a table (please see **Table 8** in the revision) to explain the metrics for performance evaluation in Challenge 2. In the meanwhile, we also add some flowcharts and cartoons (please see **Figs. 3, 6a-6b**, and **Supplementary Figs. 1-4**) to more visually explain the relevant metrics in Challenge 2.

Q.2.9. As mentioned above, some preliminary results with these proposed metrics for Challenge 2 illustrating that they are sound are useful would provide support for their inclusion in this study.

R.2.9. Based on the GM12878 Pilot Data, we have now included preliminary results of the proposed metrics (e.g., *Irreproducibility* and *Consistency* for real data, *SCC*, *MRD*, *NRMSE* and *ARR* for SIRV data) for Challenge 2 (**Figure 6**). We show that these metrics are able to distinguish differences between pipelines and can highlight pipelines that perform well on multiple metrics (e.g. Pipeline 2).

Q.2.10. The evaluation metrics for the Challenge 3 can be split into two levels.

For the mouse data, the prediction of transcripts without the reference genome or transcriptome will be analysed using the metrics from Challenge 1. This implies that the transcript models will be mapped to the genome, but the authors do not say this explicitly. They should also indicate what method they will use to map the mouse transcript models to the mouse genome.

R.2.10. Thank you for this remark. Since we have the metadata for each Challenge 1 submission, we can use the same mapper when the same pipeline was applied for the mouse data in Challenge 1 and 3 (or

equivalent). However, for comparing between pipelines within Challenge 3 we will use a common mapper. We will use minimap2 as this is widely used by the community and is the only mapper that was used by submitters to the LRGASP challenges.

We have added a sentence at the Challenge 3 section indicating this “As fasta rather than gtf files are provided in Challenge 3, we will use, when possible, the same mappers as those provided in Challenge 1 for equivalent pipelines or minimap2 otherwise”.

Q.2.11. This also raises the potential issue of errors introduced by the mapping tool. Even if the transcript models have 100% identity with the reference genome, it is not clear that every tool will produce the correct alignment. If the identity is <100%, as it is most likely when using long reads, any mapping tool will make mistakes. Using the same mapping tool for all methods will make this evaluation comparable. The authors should clarify further their approach in this case.

R.2.11. Please see our response R.2.10.

Q.2.12. The evaluation metrics for the data from manatee seem to be less developed. The authors mention that 454 data will be used, but do not appear to say how.

The authors mention using BUSCO. They should probably specify with BUSCO set they will use (e.g. mammalian). Also, they should specify what they mean with BUSCO completeness. Analysis with BUSCOs can provide the proportion of predictions with a BUSCO match, and the proportion of BUSCOs found. Both are relevant in this context.

R.2.12. We apologize for the limited information on the manatee data evaluation. We have expanded now this description, and include new metrics that complement the proposed evaluations in the first version of the manuscript. In particular, 454 data will be used to evaluate junction chaining. While we do not expect that 454 reads will cover all splice sites of long reads transcripts (454 expected read length: ~400-500bp), they are more likely to span several junctions. This will be a comparative analysis across pipelines as no statement of absolute 454 read coverage of junction chaining can be made.

As for BUSCO analysis, we return the number of BUSCO genes that are found as complete sequences, as incomplete sequences and that are duplicated (more than one transcript model matches one BUSCO gene). The BUSCO_duplicated indicates which BUSCO genes have more than one matching transcript

and the BUSCO outputs allows to identify the duplication level of each BUSCO match, which is a metric similar to the redundancy level computed in Challenge 1. We denote this now as BUSCO redundancy level. Moreover, this information also reveals the proportion of transcript models with a BUSCO hit. We have now added to the manuscript the details of the BUSCO analysis. Thank you for the suggestion.

Q.2.13. Regarding the experimental validation, the authors propose to evaluate the predicted sequences as well as their abundances. These aspects are less clearly defined, but may require further clarity and possibly a more structured plan. It would be interesting also if the authors could provide preliminary data showing that the proposed approaches are indeed valid for the proposed aims.

R.2.13. The reviewer is correct that we did not present a highly structured plan for the experimental validation. We intentionally built in flexibility of the experimental design, because we did not have the results from the participant submissions at hand to understand the best procedure for selection of targets. However, we have now included more details of the validation strategies in the main text, additional summary tables, and additional supplementary figures (**Supplementary Figs. 13**). Additional details in response below (Q.2.14).

Currently, we are analyzing the participant submissions, and are compiling statistics regarding the presence and uniqueness of certain isoforms. We will dynamically visualize and filter the isoforms through a modified version of the UCSC browser. The gene selection process is semi-random, as we need to impose practical constraints such as gene length, abundance, and other properties. The entire selection process will be described in the final manuscript.

Q.2.14. For the sequences, there is a clear plan based on targeted sequencing with Sanger / ONT / or PacBio. For quantification is less clear what the best procedure will be.

R.2.14. Assessing isoform-level quantification is challenging due to a lack of a gold standard and a lack of orthogonal approaches that can accurately quantify full-length isoform relative and absolute abundances. For the LRGASP project, the validation of isoform quantification results will be achieved by a convergence of different approaches representing computational and experimental validation.

First, the H1/H1-DE mix design allows for validation of likely hundreds if not thousands of targets. We sequenced a mixed ratio of H1 and H1-DE samples (the ratio only known to one individual who's lab did not submit any predictions) and also each cell line individually to establish the isoforms present in only one or the other sample before mixing. In essence, the pre-mixed sample represents the "ground truth" of isoform expression before the mix. Participants provided transcript quantification for the H1-mix and then at a later time the H1 and H1-DE samples alone. Evaluations will computationally mix the H1 and H1-DE quantifications at the expected ratios and determine how close this is to the observed H1-mix quantification.

Second, participants assess abundances from simulated lrrRNA-seq datasets. In this case, variability of quantification that may be due to alignment or error profiles of the reads will be assessed.

Third, for 10-20 targets, we will employ multi-target, isoform-specific qPCR, targeting isoform-specific junctions, and constitutive regions which will help inform on full-length isoform abundance (e.g., Vandembroucke et al. 2001, Brooks et al. 2014). The design of qPCR targets will be undertaken with great care, as this datatype returns information of the abundance of short sequence regions and the full-length isoform abundance must be inferred from such data.

We have now summarized these validation plans in a new **Supplementary Figures 7-13**.

Q.2.15. The authors propose the use of NRSeq. However, this is based on long-read sequencing, which will also include errors, and its not 100% efficient, so the experiments may not produce as many reads as used in the predictions. It would be useful if the authors could show some preliminary data supporting that this strategy is valid for the proposed validation.

R.2.15. Since our initial submission, we found that the remaining amount of RNA aliquots from our initial study was not sufficient to perform NRSeq on; therefore, we have removed inclusion of this data in our experiments.

Q.2.16. As before, the experimental validation for Challenge 3 is less clear. It is more a list of possibilities, but it is not clearly defined what will be tested. Further clarification of what aspects will be covered would be useful.

R.2.16. For challenge 3 on the manatee data, we will perform a targeted loci isoform validation. We have identified 15 relevant genes to validate using a PCR approach similar to the design of Challenge 1 predictions. These genes have known isoforms in the mammalian immune system, stress response, or detoxification mechanisms such as tumor necrosis factor receptor superfamily member 6, tyrosine-protein kinase fyn, heat shock protein HSP 90-beta, heat shock 70 kDa protein, and intercellular adhesion molecule 1 (CD45), among others. Additionally, junction chains detected by the pipelines will be validated with PCR. These junction chains could be present in the genes of interest or other genes relevant to the immune system function.

In summary, the study is well-developed and clearly defined regarding the evaluation of transcripts using a reference genome (Challenge 1), but it is still not clearly defined for Challenge 2 and very poorly developed for Challenge 3. The authors should significantly improve the clarity of the text, tables, descriptions, and formulae to make it more homogenous and easily accessible to other people interested in using and interpreting the proposed metrics. The experimental validation proposed is a very interesting aspect, but it is not well defined in some cases. Providing some preliminary evidence that the proposed metrics are useful and sound for the evaluations to be carried out would significantly improve the manuscript and potentially will simplify some aspects that may turn out to be not as useful as suggested.

Thank you for the positive remarks and for the suggestions. We hope that the additional clarifications in the manuscript and the incorporation of the Pilot Data result are sufficient to address reviewer 2's concerns.

Reviewer #3:

Remarks to the Author:

The manuscript presented here outlines the proposed data collection and analysis competition for the LRGASP. The purpose of this is to evaluate long read sequencing technologies for transcriptome sequencing. This is a very timely project as these technologies are beginning to gain momentum for this purpose and we are highly likely to see much broader uptake of long read sequencing in transcriptomics in the near future. With these technological advances it is very important to understand the strengths and weaknesses of the technologies and the analysis methods for this application. This proposal

addresses these problems specifically for transcript identification and quantification. It uses established definitions of transcripts in well studied organism and also addresses the use of long reads in non-model organisms which is likely to be a significant application of these technologies.

The plan is that data is generated for different samples and different technologies. This has already been done. Then the analysis aspect is performed through challenges where the community is asked to submit their analysis of the data. It is organised into 3 research challenges and entrants can submit analysis approaches for one part or multiple parts of the challenges. The results will then be compared by the research team. It is clear that the methods for comparing the analysis results have been thought through thoroughly and this is not an easy thing to do. This is a strength of this proposal.

Q.3.1. My biggest concern with this manuscript is that it relies on the community to enter the analysis challenges. How do we ensure that the best methods or methods that have already been proposed or ones that are still in development are entered? Do the authors of analysis methods need to submit their own analysis or will some of the team be able to perform already proposed analysis methods on the data to make sure it is entered into the challenge? I feel that there should be a survey of the literature to make sure that proposed methods for these task are included in the challenges. **If the set of most commonly used methods were not submitted by the community this manuscript would not turn out to be of the highest utility and standard.** Can you ensure this is achieved?

R.3.1. Thank you for sharing your concern with us that was also a concern by the consortium. LRGASP was set up as a context where each participating group -usually tool developers- will submit their predictions for any of the three Challenges. Based on previous experiences with *GASP project, the organizers felt it was critical for the integrity and fairness of the tool comparisons to not have the internal organizers run other developers' tools. We made extensive announcements of the Challenge in relevant conferences and social media, created a distribution list for email notifications and held nine community calls (starting in September 2020) where the LRGASP project was announced, progress of the LRGASP project was presented, and questions on participation or submissions were addressed. We additionally did an internal survey of published lrrna-seq methods and directly invited authors to participate in the challenge. This does not guarantee that all or even most widely used tools will participate, and relevant pipelines might be absent of the LRGASP comparisons. In order to preserve the fairness of the submission process and the LRGASP timelines but still be able to extend our analysis to methods that did not make it to the competition we adopted a dual strategy. First, LRGASP evaluation scripts and datasets are available to the community. This allows for anyone to run LRGASP evaluations on transcriptome predictions made by any current or future software, and compare them to the participating pipelines. Second, we are collaborating with OpenEBench

(<https://openebench.bsc.es/dashboard>), the Elixir platform for life benchmarks, to set up the LRGASP challenge as one of their supported projects. This will provide an easy platform for anyone to submit transcriptome predictions using LRGASP data and readily visualize the performance of their methods compared to existing submissions. We are targeting a parallel publication of the LRGASP paper and the OpenEBench implementation. We plan to showcase the utility of OpenEBench-LRGASP by including methods that were not part of the competition in this separate publication.

Reviewer #4:

Remarks to the Author:

The manuscript describes the design and logistics of LRGASP, a community competition designed to assess the strengths and challenges of the long-read RNA-seq technologies.

While the long-read DNA-seq grew explosively over the last few years, the long-read RNA-seq appears to be lagging, in part owing to the lack of careful quality assessment studies. The LRGASP aims to close this gap, hopefully resulting in much broader adoption of long-read technologies for transcriptomic analyses.

I am impressed with the magnitude and thoughtfulness of the LRGASP design and applaud the Authors for taking on such an important community project. Since the competition is already underway, my comments and suggestions below mostly speak to the analyses and presentation of the results. However, it appears that the Authors are open to running the pipelines themselves: “We expect to re-run analysis pipelines for well-performing submissions to help ensure reproducibility.” I believe it will be beneficial to do that not just for the sake of reproducibility but also to perform analyses for the publicly available datasets that are not present in the competition, which I will point out in several of my comments.

Q.4.1. Most of the biological samples in this study are cell lines. While this simplifies both the data generation, it may also reduce the generalizability of the results to the more complex biological samples (e.g., tissues). It would be illuminating if the Authors ran the pipelines for a few more complex long-read datasets and compared their performance to the cell lines.

R.4.1. This is an important question that was also raised by reviewer 1. Please refer to R1.1 for our answer.

Q.4.2. SQANTI3 is the primary analysis tool for all the results. The original SQANTI publication was in 2018, and it describes a much earlier version. It also describes the SQANTI application to the PacBio but not ONT datasets. Was the new version expanded to include ONT support? The Authors need a very detailed Methods section describing SQANTI3 (or even a companion paper).

R.4.2. SQANTI is actually agnostic for the type of sequencing platform that produces transcript models as it simply compares a fasta (or gtf) file to a reference transcriptome. Therefore the evaluation framework can be applied to any technology and in fact many Nanopore transcriptomics papers use SQANTI (PMID: 31366910, PMID: 33937765, PMID: 33397972, doi: <https://doi.org/10.1101/478172>, to cite a few) and even Nanopore recommends SQANTI for QC of their transcriptome datasets (https://nanoporetech.com/sites/default/files/s3/literature/ResearchSpotlight_Olive_Fruit_Fly_17Jun2019_FAW_WEB_INTERACTIVE.pdf). New metrics implemented in SQANTI3 are actually described in the LRGASP paper. We second the reviewer's suggestion and are aiming on publishing SQANTI3 as a companion paper.

Q.4.3. A large number of challenges, datasets, input data combinations, and metrics may become overwhelming for the readers. Tables are probably not the best way to represent the competition setup. I suggest replacing them with flowcharts and cartoons, illustrating the input data flow, output results, and metrics.

Simplified flow charts were created as an overview of each challenge, the challenge evaluations, and validation approaches. These are provided in new Supplementary Figures 7-13. In addition, examples of specific output and data visualizations of the resulting evaluations are presented by our new Pilot Data section of a mock evaluation of published GM12878 data.

Q.4.4. The simulations are an essential part of any benchmarking data. The simulation strategy needs to be described in detail and thoroughly justified. How close are the simulations to the actual data? Which idiosyncrasies of the long-read technologies can (or cannot) be appropriately simulated and benchmarked?

R.4.4. Various tools for simulating sequencing data were developed in the past decade. The question of how well they mimic the real data remains an important one to this day. Below we provide simulated data properties and potential divergence from the real datasets.

ONT data was simulated with NanoSim in transcriptome mode (also known as Trans-NanoSim). NanoSim uses pre-trained models that take into account such read characteristics as (1) read length distribution, (2) error pattern, including homopolymer-dependent errors, and (3) unaligned sequences at reads ends typical for ONT. As claimed in the Trans-NanoSim publication, the tool simulates reads that accurately mimic real Nanopore RNA sequencing data and outperforms previously developed DeepSimulator (Hafezqorani et al., 2020). For generating the data we selected ONT cDNA and dRNA pre-trained models provided in the NanoSim package. However, manual inspection revealed that as the transcript truncation is done randomly in Trans-NanoSim, no 3'/5' bias is introduced. Thus, simulated ONT data has slightly different coverage profiles compared to the real ONT cDNA/dRNA data (see Figure below).

Figure 4.4. Normalized transcript coverage profile for simulated (red) and real (blue) for ONT cDNA (left) and dRNA (right) data.

For PacBio simulation we used IsoSeqSim (<https://github.com/yunhaowang/IsoSeqSim>), which truncates input reference transcript sequences and uniformly inserts errors according to the given probabilities. In contrast to ONT data, uniform sequencing error profile for PacBio appears to be a reasonable choice according to the previously developed tool for simulating genomic PacBio reads (Ono et al., 2013). Error rate for PacBio data was derived from one of the real datasets sequenced in this work. To create a realistic coverage profile, for read truncation in IsoSeqSim we used pre-computed Sequel II truncation probabilities provided along with the package.

We also compared error rates between real and simulated data based on read alignments obtained with minimap2 (Li, 2018) in spliced mode (Table 11 in the manuscript). As the table shows, with the exception of ONT cDNA data, error rates appear to be similar. For ONT cDNA, however, real data seems to be more accurate.

To generate data with a realistic expression profile we selected a publicly available long-read RNA dataset and estimated transcript counts by mapping reads to the reference transcriptome with minimap2 (Li, 2018), which were further provided to all simulation tools as an input. Although some reads can be potentially mapped to a wrong isoform or a paralogous gene, Hafezqorani et al., 2020 shows that this simple estimation gives realistic transcript abundances. Finally, polyA tails were attached to the 3' end of reference transcript sequences prior to running the simulation.

An additional potential difference that we see is absence of intergenic / DNA contamination reads that may appear in the real data.

As a summary, we consider that simulated data strongly resembles real RNA long-read data in terms of sequencing error patterns and transcript abundances. Although some aspects of the real data might not be fully captured, exploiting such data for tool benchmarking is useful and may provide additional insights. We have now added more details in the respective section of the manuscript.

Q.4.5 One of the significant problems in competitions like RGASP is overfitting the models to maximize the known performance metrics. While some of the “truth” is hidden from the competitors, all the datasets and evaluation scripts are available to participants, which allows for parameter overfitting for a specific dataset and metric. How are the Authors planning to detect and combat the overfitting? If the Authors could run the pipelines themselves on several “hidden” datasets, it would help assess the overfitting and allow for more fair and generalizable comparisons.

R.4.5. The reviewer raises an important issue. The consortium has worked to balance two aspects that are important for a competition such as LRGASP: transparency and fairness, versus overfitting. By providing datasets and analysis scripts, we aimed to be transparent and fair and to make sure that all participants could have access to implemented evaluation metrics before submission to minimize any undisclosed evaluation criteria that will obscure the fairness of the context. To mitigate possible overfitting, we worked with a number of factors.

1. The GENCODE release used for evaluation of Challenges 1 and 2 was not available by the submission deadline
2. The loci used for GENCODE manually-curated annotations that will be used for Challenge 1 and 2 evaluations were not disclosed before-hand
3. CAGE and PolyA-seq data used in evaluations were not available before the submission deadline
4. We required predictions in multiple real and synthetic samples, and in multiple organisms, to reduce the chances of overfitting.
5. For Challenge 2, new samples (H1 and H1-DE) had to be analyzed after the deadline, but using the same pipeline and parameters used for quantifying the H1/H1-DE-mix.

Moreover, we request all analysis pipelines are available and ready to be used to verify results. This will allow us to assess any overfitting of parameters.

Q.4.6 Most evaluations discussed in the present manuscript deal with comparing analysis pipelines. However, one of the stated (and very important) goals of the LRGASP is to compare the long-read technologies, namely PacBio and ONT, as well comparison of long- and short-read (Illumina) technologies. How are the Authors planning to do it?

R.4.6. Thank you for raising this important aspect of our evaluation. The LRGASP experimental design contains three factors: sequencing platform, library preparation protocol and bioinformatics algorithm to process data. Individual pipelines can be assessed by comparing LRGASP metrics and this has been now illustrated in Figures 4 to 6 with the pilot data. The evaluation of the experimental factors (e.g. sequencing platform) will be done by assessing transcript model properties that are associated to a majority of pipelines that use a given level in the experimental factor. Our Pilot Data suggests that there might not be a large number of transcript models that are detected exclusively by all pipelines with the same factor level (for example by all pipelines using Pacbio and by no pipelines using Nanopore). To make comparisons of transcripts identified by different platforms or computational pipelines easier, we have created a barcode that is associated with each unique transcript model, and that describes the frequency of detection as a function of the experimental factors. Note that to allow direct comparison of transcript models across pipelines we consider the same transcript those transcript models that have the same Unique Junction Chain (UJC) and allow for variability at 3' and 5' ends. The barcode includes additional information of transcript length, number of exons, expression value, and variability at 3' and 5' ends. The barcode gives us great flexibility for making queries that address questions related to experimental factors. For example, we can study expression level, exon number and transcript length of transcripts in relation to their detection by Nanopore and/or Pacbio, or if there is a particular transcript feature that is associated to transcripts exclusively found by the dRNA-seq protocol. We have added a

new section at the end of Challenge 1 evaluation where we describe how transcripts will be compared across pipelines, the definition of the UJC barcode and the identification of biases associated to LRGASP experimental factors. We provide some examples of the expected results based on our Pilot Data in Figure 7.

Q.4.7. Experimental validation is extremely interesting and will probably be the most exciting part of this work. It's a bit disappointing that long-range novel transcript validation via targeted amplification/re-sequencing only involves a small number of target transcripts, which would not permit to confidently assessment error rates for detection of novel transcripts.

We thank the reviewer for this suggestion. Indeed, experimental validation will be very interesting. The scope of the work is inevitably constrained by resource and practical limitations. First, there are relatively few, if any, orthogonal approaches that can identify and quantify full-length isoforms (and confirm sequence), even at reduced scale. Through a series of many meetings with LRGASP organizers and outside parties, we exhaustively considered available technologies and options for experimental validation.

What we present is a multi-pronged approach based on simulated data, computational validation (e.g. H1-mix experiment for quantification), and multiple PCR-based (e.g., RT-PCR but with a different RT, CAGE-seq, polyA-seq) approaches. The accuracy for novel transcript detection will not only be assessed through experimental validation, but through simulated datasets and manual annotation (GENCODE). We will target for amplification and re-sequencing 168 isoforms, with potential to increase this to 384 targets as needed.

Decision Letter, first revision:

2nd Apr 2022

Dear Angela,

Thank you once again for submitting your revised Stage 1 Registered Report, entitled "Systematic assessment of long-read RNA-seq methods for transcript identification and quantification." After consulting again with our reviewers, I am delighted to say that we can offer acceptance in principle. You may progress to Stage 2 and complete your study as approved.

As you know, a condition of acceptance-in-principle is that the authors agree to deposit their Stage 1

accepted manuscript in a repository, either publicly or under embargo until Stage 2 manuscript acceptance and publication. We are very keen to showcase our accepted-in-principle manuscripts, so that our readers, reviewers, and potential authors can gain insight into the requirements of the format as well as an idea of the types of projects that are suitable for publication as Registered Reports in Nature Methods. We have set up a space on figshare (https://springernature.figshare.com/registered-reports_nmmethods) to host all of our accepted-in-principle manuscripts, which can either be made public or kept under embargo until Stage 2 acceptance (depending on author preference). This gives you the opportunity to have your work publicly associated with Nature Methods, and of course we will be very pleased to showcase your report if you agree to share it publicly.

If you agree with posting your Stage 1 manuscript on our figshare space, we will upload it on your behalf and either set it public or place it under embargo, depending on your choice. Your protocol will be licensed under a CC BY license (Creative Commons Attribution 4.0 International License). The CC BY license allows for maximum dissemination and re-use of open access materials and is preferred by many research funding bodies. Under this license users are free to share (copy, distribute and transmit) and remix (adapt) the contribution including for commercial purposes, providing they attribute the contribution in the manner specified by the author or licensor (read full legal code: <http://creativecommons.org/licenses/by/4.0/legalcode>) Please note that any use of <https://springernature.figshare.com> will be subject to the figshare terms of use. Figshare has the right to enforce these terms and conditions where applicable. Use of third party services and sites will be subject to the relevant terms of use and will apply if we act on your behalf in this regard. Do let me know if you would like to take up this option or if you have any questions regarding the manuscript deposition requirement.

Please also note that depositing the work on our figshare space does not preclude deposition of your Stage 1 manuscript on other repositories – your manuscript can also be posted on any other public repository of your choice.

Following completion of your study, we invite you to resubmit your finalized manuscript as a Stage 2 Registered Report. We will send the Stage 2 manuscript to our reviewers for a final check, but they will be instructed that any comments on novelty and/or potential significance of the results will not factor into our final decision.

IMPORTANT: Please note that your manuscript can still be rejected for publication at Stage 2 if the Editors consider any of the following to hold:

- The authors substantially alter the rationale for the study as approved in the Stage 1 submission (please note that the Introduction should not be significantly modified from the Stage 1 manuscript).
- The authors fail to adhere closely to the approved experimental plan. (Please contact us as soon as possible for advice if at any point you need to make any changes to your experimental plan.)
- The authors' conclusions are not justified given the data obtained.
- Any post hoc (unregistered) analyses are not justified or are overly dominant in shaping the authors' conclusions.
- Our open science requirements (detailed below) are not followed.

Should authors choose to withdraw their Registered Report at any time, we will publish a Withdrawn Registration notice.

When you are ready, please use the following link to access your home page and submit your Stage 2 Registered Report:

[REDACTED]

We expect your Stage 2 Registered Report to be submitted by the date specified in your latest cover letter (i.e., July 2022). If unforeseen circumstances prevent submission by that date, please contact us as soon as possible.

Thank you again for submitting your work to Nature Methods and we look forward to receiving your Stage 2 Registered Report! Please do not hesitate to reach out to me at any time if you have questions.

Best regards,
Lei

Lei Tang, Ph.D.
Senior Editor
Nature Methods

OPEN SCIENCE REQUIREMENTS

REPORTING SUMMARY AND EDITORIAL POLICY CHECKLISTS

When submitting your Stage 2 manuscript, please include a reporting summary and editorial policy checklists. If you have any questions about these checklist, please see <http://www.nature.com/authors/policies/availability.html> or contact me.

IMAGE INTEGRITY

When submitting the revised version of your manuscript, please pay close attention to our

[href="https://www.nature.com/nature-research/editorial-policies/image-integrity">](https://www.nature.com/nature-research/editorial-policies/image-integrity)Digital Image Integrity Guidelines. and to the following points below:

DATA AVAILABILITY

All novel DNA and RNA sequencing data, protein sequences, genetic polymorphisms, linked genotype and phenotype data, gene expression data, macromolecular structures, and proteomics data must be deposited in a publicly accessible database, and accession codes and associated hyperlinks must be provided in the "Data Availability" section.

CODE AVAILABILITY

Please include a "Code Availability" subsection in the Online Methods which details how any custom code is made available. Only in rare cases (where code is not central to the main conclusions of the paper) is the statement "available upon request" allowed (and reasons should be specified).

For more information on our code sharing policy and requirements, please see:
<https://www.nature.com/nature-research/editorial-policies/reporting-standards#availability-of-computer-code>

ORCID

Nature Methods is committed to improving transparency in authorship. As part of our efforts in this direction, we are now requesting that all authors identified as 'corresponding author' on published papers create and link their Open Researcher and Contributor Identifier (ORCID) with their account on the Manuscript Tracking System (MTS), prior to acceptance. This applies to primary research papers only. ORCID helps the scientific community achieve unambiguous attribution of all scholarly contributions. You can create and link your ORCID from the home page of the MTS by clicking on 'Modify my Springer Nature account'. For more information please visit please visit www.springernature.com/orcid.

TRANSPARENT PEER REVIEW

Please note: we allow redactions to authors' rebuttal and reviewer comments in the interest of confidentiality. If you are concerned about the release of confidential data, please let us know specifically what information you would like to have removed. Please note that we cannot incorporate redactions for any other reasons. Reviewer names will be published in the peer review files if the reviewer signed the comments to authors, or if reviewers explicitly agree to release their name. For more information, please refer to our FAQ page.

Author Rebuttal, first revision:

[There is no rebuttal letter at this stage.]

Decision Letter, second revision:

Our ref: NMETH-RR46730B

17th Oct 2023

Dear Dr. Brooks,

Thank you for submitting your stage 2 Registered Report manuscript "Systematic assessment of long-read RNA-seq methods for transcript identification and quantification" (NMETH-RR46730B). It has now been seen by one of the original referees and their comments are below. The reviewer finds that the paper adheres to the experimental and analysis plan described in the Stage 1 submission. Therefore we'll be happy in principle to publish it in Nature Methods, pending minor revisions to satisfy the referee's final requests and to comply with our editorial and formatting guidelines.

TRANSPARENT PEER REVIEW

Nature Methods offers a transparent peer review option for new original research manuscripts submitted from 17th February 2021. We encourage increased transparency in peer review by publishing the reviewer comments, author rebuttal letters and editorial decision letters if the authors agree. Such peer review material is made available as a supplementary peer review file. Please state in the cover letter 'I wish to participate in transparent peer review' if you want to opt in, or 'I do not wish to participate in transparent peer review' if you don't. Failure to state your preference will result in delays in accepting your manuscript for publication.

Please note: we allow redactions to authors' rebuttal and reviewer comments in the interest of confidentiality. If you are concerned about the release of confidential data, please let us know specifically what information you would like to have removed. Please note that we cannot incorporate redactions for any other reasons. Reviewer names will be published in the peer review files if the reviewer signed the comments to authors, or if reviewers explicitly agree to release their name. For more information, please refer to our FAQ page.

ORCID

IMPORTANT: Non-corresponding authors do not have to link their ORCIDs but are encouraged to do so. Please note that it will not be possible to add/modify ORCIDs at proof. Thus, please let your co-authors know that if they wish to have their ORCID added to the paper they must follow the procedure

described in the following link prior to acceptance:

Sincerely,

Lei

Lei Tang, Ph.D.
Senior Editor
Nature Methods

Reviewer #4 (Remarks to the Author):

The manuscript describes the LRGASP project, which aims to assess the current state of long-read RNA-seq technologies and analysis methods. The Authors generated a large number of high-quality reference datasets, collected the analyses from multiple community participants, and performed a comprehensive assessment of computational pipelines and technologies. The findings and conclusions of this effort are exciting and provocative and may be of great interest to the genomics community.

I have some concerns about the presentation of the comparison results: the presented figures are somewhat hard to follow, and the Manuscript lacks punchline figures that could concisely summarize the main findings. I appreciate that summarizing enormous amounts of data (multiple experimental datasets, pipelines, and metrics) is not a trivial task. Some specific suggestions I have are as follows:

1. Figures 1c,d,e all represent precision and sensitivity for various datatypes: SIRV, simulated, and manually annotated genes. To improve readability, It would be great if these plots were all the same type, e.g., bar charts.
2. I recommend devoting more space to discussing the simulated results.
3. There is little discussion about the fidelity of the tools on real long-read data. Figure 1a shows the overall statistics of detecting annotated and novel transcripts. I appreciate that it's hard to define sensitivity and precision for real data in the absence of ground truth. However, there are substitutes that can be used; for instance, true transcripts can be defined as those supported by short reads.
4. Figures 2b,d lump together a lot of data: different tools, datasets, and metrics, and it's practically impossible to get any helpful information from them. I recommend concentrating on just a few datasets,

with the most interesting results, both in the Figure and discussion, and leaving the rest to the supplementary materials.

5. Final recommendations are a crucially important outcome of this project. I recommend that the Authors create a set of simple plots that corroborate these recommendations. This will also effectively create a set of punchline figures mentioned above.

Author Rebuttal, second revision:

We would like to thank the reviewer for the constructive feedback and suggestions. We have addressed all comments below:

Reviewer #4 (Remarks to the Author):

The manuscript describes the LRGASP project, which aims to assess the current state of long-read RNA-seq technologies and analysis methods. The Authors generated a large number of high-quality reference datasets, collected the analyses from multiple community participants, and performed a comprehensive assessment of computational pipelines and technologies. The

findings and conclusions of this effort are exciting and provocative and may be of great interest to the genomics community.

I have some concerns about the presentation of the comparison results: the presented figures are somewhat hard to follow, and the Manuscript lacks punchline figures that could concisely summarize the main findings. I appreciate that summarizing enormous amounts of data (multiple experimental datasets, pipelines, and metrics) is not a trivial task. Some specific

suggestions I have are as follows:

1. Figures 1c,d,e all represent precision and sensitivity for various datatypes: SIRV, simulated, and manually annotated genes. To improve readability, It would be great if these plots were all the same type, e.g., bar charts.

Thank you for your remarks. We understand you refer to figures 2c,d,e that describe the results of Challenge 1 evaluation. As suggested, we have modified the plots of Figure 2e to be barplots of the same format as Figure 2c, which greatly improves readability. However, we'd like to keep the radar plot version (figure 2d) for the performance on the simulated data as more metrics were derived from these data and those are harder to fit within the barplot format. We believe this is a good compromise between readability and comprehensiveness.

2. I recommend devoting more space to discussing the simulated results.

The section that discusses the results of the simulated data addressed differences in performance for the different tools and both for the simulated Pacbio and Nanopore data. The radar plots that show performance metrics (Figure 2d) includes sensitivity, precision (both for known and novel transcripts), sensitivity for highly expressed transcripts, and redundancy (i.e. the same simulated transcript detected by two or more predicted transcripts). We acknowledge that the text discussed mostly the Sensitivity and Precision results, particularly for PacBio data. We have now extended the narrative to mention results on redundancy, sensitivity for highly expressed transcripts, and specific results for the Nanopore simulations. We hope these additions result in complete discussion of the simulated data results, while still keeping within the word limitations requested by the journal. Specifically we have included the following text in the corresponding paragraph.

“For all tools, sensitivity increased on highly expressed transcripts, and redundancy values were close to 1, except for Iso IB and Spectra, which returned a higher number of redundant predictions.”

“Exceptions were Bambu and IsoQuant, which had good precision for ONT-known simulated transcripts, and StringTie at metrics other than those related to novel transcript discovery.”

3. There is little discussion about the fidelity of the tools on real long-read data. Figure 1a shows the overall statistics of detecting annotated and novel transcripts. I appreciate that it's hard to define sensitivity and precision for real data in the absence of ground truth. However, there are substitutes that can be used; for instance, true transcripts can be defined as those supported by short reads.

Thank you for your comment. Providing performance metrics from the real dataset is indeed difficult, but we have addressed the fidelity of the tools in these data using orthogonal datasets as indicated by the reviewer. Specifically, the third paragraph of the Challenge 1 results section describes the performance of the tools against CAGE (TSS validation), Quant-seq (TTS validation) and Illumina short reads (for splice junction validation). We concluded that there is a strong tool-component in the results and that there is a relationship between the extent to which tools base predictions on the reference annotation and failure in the support by orthogonal data. We even explain this behavior by showing that for some of these tools, reported transcript models are not always 100% covered by long reads sequences, suggesting overcorrection towards the reference. We hope this is a sufficient discussion of the fidelity of the tools when analyzing real data.

4. Figures 2b,d lump together a lot of data: different tools, datasets, and metrics, and it's practically impossible to get any helpful information from them. I recommend concentrating on just a few datasets, with the most interesting results, both in the Figure and discussion, and leaving the rest to the supplementary materials.

Thank you for your comments and suggestions. We appreciate your meticulous review and would like to clarify that Figs. 2b,d mentioned in your review actually corresponds to Figs. 3b,d on Challenge 2 in the main text.

To enhance the clarity and interpretability of the evaluation data and metrics for Challenge 2, we have implemented the following optimizations:

i) Data Streamlining: We reduced redundancy by relocating evaluation metrics, including ACVC, ACC, and NRMSE, to supplementary figures (**Supplementary Figs. S1b, S2c, S4b, S5a and S6b**). This adjustment aims to simplify the main figures, providing a more focused and comprehensible view.

ii) Visual Emphasis: Positive and negative correlations between evaluation metrics and quantification accuracy are highlighted using distinct colors (red and blue) on the y-axis in **Figs. 3b-e** and in the figure legend of **Fig. 3**. This visual enhancement aims to offer readers an intuitive understanding of each tool's performance across different metrics.

iii) Summary Ranking Plot: We showed concise dot plots (**Figs. 3f,g** and **Extended Data Fig. 2**) displaying the top 3 quantification tools and protocols-platforms for each metric. This enables readers to swiftly identify superior tools and platforms for specific data scenarios, facilitating their decision-making during experimental design and analysis.

5. Final recommendations are a crucially important outcome of this project. I recommend that the Authors create a set of simple plots that corroborate these recommendations. This will also effectively create a set of punchline figures mentioned above.

Our final recommendations relate to both the technologies, analysis methods and depend on the goals of the rRNA-seq experiment. We acknowledge that progress both in technologies and algorithms is constant and may affect our conclusions although we expect that general trends, such as the relevance of sequence quality for accurate transcript isoform detection or of sequencing depth for accurate quantification will hold. Summarizing recommendations about tools and technologies as punchline figures is therefore

challenging. To address this remark, we are now including a ranking plot of the performance metrics for Challenge 1 and updated and simplified the similar plot for Challenge 2 that help summarize the main findings of the project, both for technologies as for analysis tools, to support our recommendations. We hope these will help to communicate the main conclusions of our work.

Final Decision Letter:

3rd May 2024

Dear Dr Brooks,

I am pleased to inform you that your Registered Reports, "Systematic assessment of long-read RNA-seq methods for transcript identification and quantification", has now been accepted for publication in Nature Methods. The received and accepted dates will be 2nd Aug 2021 and 3rd May 2024. This note is intended to let you know what to expect from us over the next month or so, and to let you know where to address any further questions.

Over the next few weeks, your paper will be copyedited to ensure that it conforms to Nature Methods style. Once your paper is typeset, you will receive an email with a link to choose the appropriate publishing options for your paper and our Author Services team will be in touch regarding any additional information that may be required. It is extremely important that you let us know now whether you will be difficult to contact over the next month. If this is the case, we ask that you send us the contact information (email, phone and fax) of someone who will be able to check the proofs and deal with any last-minute problems.

Please note that Nature Methods is a Transformative Journal (TJ). Authors may publish their research with us through the traditional subscription access route or make their paper immediately open access through payment of an article-processing charge (APC). Authors will not be required to make a final decision about access to their article until it has been accepted. Find out more about Transformative Journals

Authors may need to take specific actions to achieve compliance with funder and institutional open access mandates. If your research is supported by a funder that requires immediate open access (e.g. according to Plan S principles) then you should select the gold OA route, and we will direct you to the compliant route where possible. For authors selecting the subscription publication route, the journal's standard licensing terms will need to be accepted, including self-archiving policies. Those licensing terms will supersede any other terms that the author or any third party may assert apply to any version of the manuscript.

If you are active on Twitter/X, please e-mail me your and your coauthors' handles so that we may tag you when the paper is published.

Please note that you and any of your coauthors will be able to order reprints and single copies of the issue containing your article through Nature Portfolio's reprint website, which is located at

<http://www.nature.com/reprints/author-reprints.html>. If there are any questions about reprints please send an email to author-reprints@nature.com and someone will assist you.

Best regards,
Lei

Lei Tang, Ph.D.
Senior Editor
Nature Methods